# THE POWER OF SMALL INITIALIZATION IN NOISY LOW-TUBAL-RANK TENSOR RECOVERY

**Zhiyu Liu**[1,2]**, Haobo Geng**[3]**, Xudong Wang**[1,2]**, Yandong Tang**[1]**, Zhi Han**[1]**, Yao Wang**[4,*]

[1] State Key Laboratory of Robotics and Intelligent Systems, Shenyang Institute of Automation,
   Chinese Academy of Sciences, China
[2] University of Chinese Academy of Sciences, China
[3] School of Electrical and Electronic Engineering, Nanyang Technological University, Singapore
[4] The Center for Intelligent Decision-making and Machine Learning,
   School of Management, Xi'an Jiaotong University, China

## ABSTRACT

We study the problem of recovering a low-tubal-rank tensor $\boldsymbol{\mathcal{X}}_\star \in \mathbb{R}^{n \times n \times k}$ from noisy linear measurements under the t-product framework. A widely adopted strategy involves factorizing the optimization variable as $\boldsymbol{\mathcal{U}} * \boldsymbol{\mathcal{U}}^\top$, where $\boldsymbol{\mathcal{U}} \in \mathbb{R}^{n \times R \times k}$, followed by applying factorized gradient descent (FGD) to solve the resulting optimization problem. Since the tubal-rank $r$ of the underlying tensor $\boldsymbol{\mathcal{X}}_\star$ is typically unknown, this method often assumes $r < R \le n$, a regime known as over-parameterization. However, when the measurements are corrupted by some dense noise (e.g., Gaussian noise), FGD with the commonly used spectral initialization yields a recovery error that grows linearly with the over-estimated tubal-rank $R$. To address this issue, we show that using a small initialization enables FGD to achieve a nearly minimax optimal recovery error, even when the tubal-rank $R$ is significantly overestimated. Using a four-stage analytic framework, we analyze this phenomenon and establish the sharpest known error bound to date, which is independent of the overestimated tubal-rank $R$. Furthermore, we provide a theoretical guarantee showing that an easy-to-use early stopping strategy can achieve the best known result in practice. All these theoretical findings are validated through a series of simulations and real-data experiments.

## 1 INTRODUCTION

In recent years, the growing complexity and dimensionality of real-world data have highlighted the limitations of traditional vector and matrix models. As a natural generalization, tensors provide a more expressive framework to capture multi-dimensional correlations inherent in data arising from applications such as hyperspectral imaging (Han et al., 2025), dynamic video sequences (Han et al., 2024), and sensor arrays (Rajesh & Chaturvedi, 2021; Fu et al., 2025). A common trait shared across these applications is the underlying low-rank structure of the data when represented in tensor form. Leveraging this property, a wide range of inverse problems can be effectively reformulated as low-rank tensor recovery tasks. Notable examples include image inpainting (Zhang & Aeron, 2016; Gilman et al., 2022; Yang et al., 2022), compressive imaging and video representation (Wang et al., 2017; Baraniuk et al., 2017; Wang et al., 2018), background modeling from incomplete observations (Cao et al., 2016; Li et al., 2022; Peng et al., 2022), and even advanced medical imaging techniques such as computed tomography (Liu et al., 2024a). The goal of low-rank tensor recovery is to recover the target tensor $\boldsymbol{\mathcal{X}}_\star$ from a few noisy measurements: $y_i = \langle \boldsymbol{\mathcal{A}}_i, \boldsymbol{\mathcal{X}}_\star \rangle + s_i$, $i = 1, 2..., m$, where $s_i$ denotes the unknown noise. This model can be concisely represented as $\boldsymbol{y} = \mathfrak{M}(\boldsymbol{\mathcal{X}}_\star) + \boldsymbol{s}$, where $\mathfrak{M}(\boldsymbol{\mathcal{X}}_\star) = [\langle \boldsymbol{\mathcal{A}}_1, \boldsymbol{\mathcal{X}}_\star \rangle, \langle \boldsymbol{\mathcal{A}}_2, \boldsymbol{\mathcal{X}}_\star \rangle, ..., \langle \boldsymbol{\mathcal{A}}_m, \boldsymbol{\mathcal{X}}_\star \rangle]$. Since $\boldsymbol{\mathcal{X}}_\star$ is low-rank, the problem can be solved via rank minimization: $\min_{\boldsymbol{\mathcal{X}}} \texttt{rank}(\boldsymbol{\mathcal{X}}), \text{s.t.} \ ||\boldsymbol{y} - \mathfrak{M}(\boldsymbol{\mathcal{X}})||_2 \le \epsilon_s$, where $\texttt{rank}(\cdot)$ denotes the tensor rank function and $\epsilon_s$ denotes the noise level.

There are various tensor decomposition methods, such as CANDECOMP/PARAFAC decomposition (CP) (Carroll & Chang, 1970; Harshman, 1970), Tucker decomposition (Tucker, 1966), Tensor

---
*Corresponding author: *yao.s.wang@gmail.com*

Singular Value Decomposition (t-SVD)(Kilmer & Martin, 2011), Tensor Train (Oseledets, 2011), and Tensor Ring (Zhao et al., 2016), each leading to different definitions of tensor rank. In this work, we adopt the t-SVD along with its associated tubal-rank (Kilmer et al., 2013). We adopt t-SVD due to its use of circular convolution along the third dimension via the t-product, enabling it to capture frequency-domain structures effectively (Wu et al., 2024). This capability makes it particularly powerful for handling multi-dimensional data such as images and videos (He et al., 2024; Wu & Fan, 2024; Wu et al., 2025; Liu et al., 2023; Han et al., 2023). Furthermore, t-SVD guarantees an optimal low-rank approximation, in a manner directly analogous to the Eckart–Young theorem for matrices (Eckart & Young, 1936). Under the t-SVD framework, since problem (2) is NP-hard, a common approach is to relax the tubal-rank constraint to the tensor nuclear norm. This reformulates the original problem as a tubal tensor nuclear norm minimization. While this relaxation is theoretically sound, solving it typically requires repeated t-SVD computations, which become increasingly expensive as the tensor dimensions grow.

To address this issue, a more recent and popular approach is to adopt the tensor Burer–Monteiro (BM) factorization, a higher-order extension of the matrix Burer–Monteiro method (Burer & Monteiro, 2003). This technique represents the large tensor as the t-product of two smaller factor tensors, thereby transforming the original problem into an optimization over the two factors, often minimizing an objective of the form[1]

$$\min_{\mathcal{U} \in \mathbb{R}^{n \times R \times k}} f(\mathcal{U}) = \frac{1}{4m} \left\| y - \mathfrak{M}(\mathcal{U} * \mathcal{U}^\top) \right\|^2, \quad \mathfrak{M}(\cdot) : \mathbb{R}^{n \times n \times k} \to \mathbb{R}^m, \tag{1}$$

where $*$ denotes the tensor-tensor product. Factorized Gradient Descent and its variants can then be applied, significantly reducing computational costs (Liu et al., 2024b; Karnik et al., 2025). However, such methods typically require prior knowledge of the tubal-rank $r$ of the target tensor, which is often unavailable in practice. As a result, it is common to assume an estimated rank $R > r$, a setting often referred to as the over-parameterized or over-rank case. However, in the case of noisy low-tubal-rank tensor recovery, over-parameterization can lead to larger recovery errors. Liu et al. (2024b) showed that the recovery error in the over-parameterized setting grows linearly with the estimated tubal-rank $R$. When the tubal-rank is significantly overestimated, the error can become substantial. Furthermore, FGD suffers from a severe slowdown in convergence when the tubal-rank is overestimated. This leads to an important question: ***In noisy low-tubal-rank tensor recovery, is it possible to obtain an error bound that depends only on the true tubal-rank $r$ ?***

By investigating this question further, we find that ***with small initialization, factorized gradient descent converges linearly to a nearly minimax optimal error only relying on $r$, even when the tubal-rank is significantly overestimated.*** As shown in Figure 1, under over-parameterization, FGD with spectral initialization yields suboptimal recovery error, while FGD with small initialization achieves the same error as in the exact tubal-rank setting. However, as the algorithm continues to iterate, the error gradually increases and eventually matches that of spectral initialization. We provide a theoretical analysis of this phenomenon and derive the best-known error bound to date. Furthermore, based on early stopping and validation (Prechelt, 1998; Stone, 2018; Ding et al., 2025), we show that this error is achievable and provide corresponding theoretical guarantees.

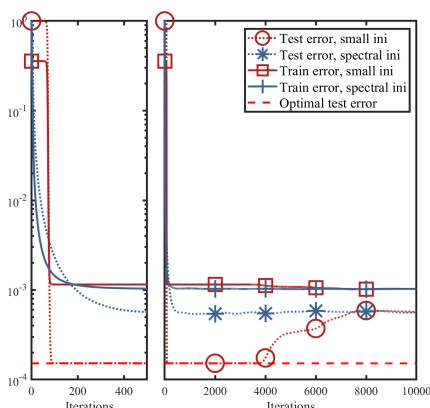

Figure 1: Comparison of training and testing errors for Problem (1) using FGD with spectral vs. small initialization. The ground-truth tensor has tubal-rank $r = 2$, overestimated rank $R = 4$, size $n = 20$, $k = 3$, $m = 5kr(2n - r)$ measurements, and noise $\sigma = 10^{-3}$. Spectral initialization follows Liu et al. (2024b), while small initialization uses a near-zero starting point. Training error is $\frac{1}{4m}||y - \mathfrak{M}(\mathcal{U} * \mathcal{U}^\top)||^2$, and testing error is $||\mathcal{U} * \mathcal{U}^\top - \mathcal{X}_\star||_F^2 / ||\mathcal{X}_\star||_F^2$. "Baseline" denotes recovery under exact rank $R = r$. Insets show early (first 500 iterations) vs. full error curves.

---

[1]As in prior work, we assume that $\mathcal{X}_\star$ is a symmetry and positive semi-definite tensor. for detailed explanation, please refer to Definition 2.

We summarize the main contributions of this paper as follows:

**Tightest error upper bound** We discover that with small initialization, FGD can achieve an error which only depends on the exact tubal-rank in noisy, over-parameterized low-tubal-rank tensor recovery. We establish global convergence and the tightest error bound for FGD that depends only on the true tubal-rank. This significantly improves upon previous results (Liu et al., 2024b). To the best of our knowledge, this is the first error bound that is independent of the overestimated tensor rank.

**Minimax lower bound and near-optimality.** We derive an information-theoretic minimax lower bound for noisy tubal-rank tensor recovery, showing that any estimator has mean square error at least $\Omega(\frac{nrk\sigma^2}{m})$. Comparing this lower bound with our upper bound demonstrates that our method is nearly optimal; the remaining gaps are only due to constant factors and dependencies on the condition number $\kappa$.

**Attainable recovery error** A validation-based early stopping method is applied to FGD to achieve the error bound without any prior information about the target tensor. We theoretically show that when the number of validation samples exceeds $\tilde{\mathcal{O}}(r^2\kappa^8)$, the validation error matches the upper bound up to constants. On both synthetic and real datasets, we demonstrate that in the over-parameterized setting, FGD (small initialization and validation-based early stopping) attains errors comparable to those achieved with the exact-rank setting, and significantly outperforms spectral and large random initializations.

## 1.1 RELATED WORKS

**Non-convex low-tubal-rank tensor recovery under t-SVD framework** Nonconvex low-tubal-rank tensor recovery methods under the t-SVD framework can be broadly categorized into two classes. The first class aims to improve recovery accuracy by replacing the tubal tensor nuclear norm with nonconvex surrogates. The second class focuses on improving computational efficiency by decomposing a large tensor into

Table 1: Comparison of several low-tubal-rank tensor recovery methods based on t-SVD. The noise vector $s$ is assumed to consist of Gaussian random variables with zero mean and variance $\sigma^2$.

| methods | rate | guarantee | error |
|---|---|---|---|
| (Zhang et al., 2020) | ✗ | ✓ | ✗ |
| (Liu et al., 2024b) | sub-linear | local | $\widetilde{\mathcal{O}}\left(\frac{nkR\sigma^2}{m}\right)$ |
| (Karnik et al., 2025) | linear | global | ✗ |
| Ours | linear | global | $\widetilde{\mathcal{O}}\left(\frac{nkr\sigma^2}{m}\right)$ |

smaller factor tensors. We first discuss the methods in the first category. These approaches are derived from the tubal tensor nuclear norm and include variants such as the t-Schatten-$p$ norm (Kong et al., 2018), weighted t-TNN (Mu et al., 2020), , partial sum of t-TNN (Jiang et al., 2020) and others (Qin et al., 2025). Other methods employ nonconvex functions such as Geman or Laplace penalties in place of the tubal tensor nuclear norm (Cai et al., 2019; Xu et al., 2019). It is worth noting that Wang et al.(Wang et al., 2021) proposed a generalized nonconvex framework that encompasses a wide range of non-convex penalty functions. However, these methods still rely on repeated t-SVD computations, which are computationally expensive, and often lack theoretical guarantees. The second category includes factorization-based methods that decompose a large tensor into two or three smaller factor tensors, followed by optimization techniques such as alternating minimization (Zhou et al., 2017; Liu et al., 2019; He & Atia, 2023; Wu et al., 2025), nonconvex tensor norms minimization (Du et al., 2021; Jiang et al., 2023b), factorized gradient descent (Liu et al., 2024b; Karnik et al., 2025), scaled gradient descent (Feng et al., 2025; Wu, 2025; Liu et al., 2025). Beyond these two main categories, there are also approaches based on randomized low-rank approximation (Qin et al., 2024) and alternating projections (Qiu et al., 2022) for solving tensor recovery problems.

**Over-parameterization in low rank tensor recovery** Factorization-based methods typically require knowledge of the tensor rank. However, the true rank is often difficult to obtain in practice. As a result, it is common to assume an estimated rank larger than the true one, a setting known as over-parameterization. In matrix sensing, it has been shown that gradient descent can still achieve the optimal solution under over-parameterization (Zhu et al., 2018; Stöger & Soltanolkotabi, 2021; Soltanolkotabi et al., 2025; Jiang et al., 2023a; Zhuo et al., 2024; Ding et al., 2025). In contrast, studies on over-parameterized settings in tensor recovery are relatively limited. Although many methods have been proposed to estimate tensor rank, these methods are computationally expensive and lack clear theoretical guarantees (Zhou & Cheung, 2019; Shi et al., 2021; Zheng et al., 2023; Zhu et al., 2025). Recently, Liu et al. (2024b) investigated low-tubal-rank tensor recovery under

over tubal-rank and established local convergence guarantees and recovery error bounds for FGD, where the error depends on the overestimated tubal-rank. Karnik et al. (2025) further proved global convergence of FGD with small initialization under over tubal-rank, in the noiseless setting. In addition, for Tucker decomposition, Luo & Zhang (2024) studied the over-parameterized setting in tensor-on-tensor regression. However, in the presence of noise, its recovery error still depends on the overestimated tensor rank. We compare our method with several closely related works, and the results are summarized in Table 1.

## 2 PRELIMINARIES

The symbols $y, \boldsymbol{y}, Y, \boldsymbol{\mathcal{Y}}$ are denoted as scalars, vectors, matrices, and tensors, respectively. Let $\boldsymbol{\mathcal{Y}} \in \mathbb{R}^{m \times n \times k}$ be a third-order tensor. We refer to its entry at position $(i, j, l)$ as $\boldsymbol{\mathcal{Y}}(i, j, l)$, and denote the $l$-th frontal slice by $\boldsymbol{Y}^{(l)} := \boldsymbol{\mathcal{Y}}(:, :, l)$, following MATLAB-style indexing. The inner product between two tensors $\boldsymbol{\mathcal{Y}}$ and $\boldsymbol{\mathcal{Z}}$ is given by $\langle \boldsymbol{\mathcal{Y}}, \boldsymbol{\mathcal{Z}} \rangle = \sum_{l=1}^{k} \langle \boldsymbol{Y}^{(l)}, \boldsymbol{Z}^{(l)} \rangle$, where each $\boldsymbol{Y}^{(l)}$ and $\boldsymbol{Z}^{(l)}$ are corresponding frontal slices.

For any tensor $\boldsymbol{\mathcal{Y}} \in \mathbb{R}^{m \times n \times k}$, its Discrete Fourier Transform along the third mode yields $\overline{\boldsymbol{\mathcal{Y}}} \in \mathbb{C}^{m \times n \times k}$. In MATLAB syntax, we have $\overline{\boldsymbol{\mathcal{Y}}} = \mathtt{fft}(\boldsymbol{\mathcal{Y}}, [\,], 3)$, and $\boldsymbol{\mathcal{Y}} = \mathtt{ifft}(\overline{\boldsymbol{\mathcal{Y}}}, [\,], 3)$. We denote $\overline{\boldsymbol{Y}} \in \mathbb{C}^{mk \times nk}$ as a block diagonal matrix of $\overline{\boldsymbol{\mathcal{Y}}}$, i.e., $\overline{\boldsymbol{Y}} = \mathtt{bdiag}(\overline{\boldsymbol{\mathcal{Y}}}) = \mathtt{diag}(\overline{\boldsymbol{Y}}^{(1)}; \overline{\boldsymbol{Y}}^{(2)}; ...; \overline{\boldsymbol{Y}}^{(k)})$.

The tensor-tensor product (t-product) of two tensors $\boldsymbol{\mathcal{Z}} \in \mathbb{R}^{m \times q \times k}$ and $\boldsymbol{\mathcal{Y}} \in \mathbb{R}^{q \times n \times k}$ is $\boldsymbol{\mathcal{Z}} * \boldsymbol{\mathcal{Y}} \in \mathbb{R}^{m \times n \times k}$, whose tubes are given $(\boldsymbol{\mathcal{Z}} * \boldsymbol{\mathcal{Y}})(i, i') = \sum_{p=1}^{q} \boldsymbol{\mathcal{Z}}(i, p, :) * \boldsymbol{\mathcal{Y}}(p, i', :)$, where $*$ denotes the circular convolution operation, i.e., $(\boldsymbol{x} * \boldsymbol{y})_i = \sum_{j=1}^{k} x_j y_{i-j(\mathrm{mod}\, k)}$.

For any tensor $\boldsymbol{\mathcal{Y}} \in \mathbb{C}^{m \times n \times k}$, its conjugate transpose $\boldsymbol{\mathcal{Y}}^{\top} \in \mathbb{C}^{n \times m \times k}$ is computed by taking the conjugate transpose of each frontal slice and reversing the order of slices 2 through $k$. The identity tensor, represented by $\boldsymbol{\mathcal{I}} \in \mathbb{R}^{n \times n \times k}$, is defined such that its first frontal slice corresponds to the $n \times n$ identity matrix, while all subsequent frontal slices are comprised entirely of zeros. This can be expressed mathematically as: $\boldsymbol{I}^{(1)} = \boldsymbol{I}_{n \times n}, \quad \boldsymbol{I}^{(l)} = 0, l = 2, 3, \ldots, k$. A tensor $\boldsymbol{\mathcal{Q}} \in \mathbb{R}^{n \times n \times k}$ is considered orthogonal if it satisfies the following condition: $\boldsymbol{\mathcal{Q}}^{\top} * \boldsymbol{\mathcal{Q}} = \boldsymbol{\mathcal{Q}} * \boldsymbol{\mathcal{Q}}^{\top} = \boldsymbol{\mathcal{I}}$.

**Theorem 1** (t-SVD (Kilmer & Martin, 2011)). *Let $\boldsymbol{\mathcal{Y}} \in \mathbb{R}^{m \times n \times k}$, then it can be factored as $\boldsymbol{\mathcal{Y}} = \boldsymbol{\mathcal{V}_Y} * \boldsymbol{\mathcal{S}_Y} * \boldsymbol{\mathcal{W}_Y}^{\top}$ where $\boldsymbol{\mathcal{V}_Y} \in \mathbb{R}^{m \times m \times k}$, $\boldsymbol{\mathcal{W}_Y} \in \mathbb{R}^{n \times n \times k}$ are orthogonal tensors, and $\boldsymbol{\mathcal{S}_Y} \in \mathbb{R}^{m \times n \times k}$ is a f-diagonal tensor, i.e., all the frontal slices of $\boldsymbol{\mathcal{S}_Y}$ are diagonal matrix.*

For $\boldsymbol{\mathcal{Y}} \in \mathbb{R}^{m \times n \times k}$, its tubal-rank as $\mathrm{rank}_t(\boldsymbol{\mathcal{Y}})$ is defined as the nonzero diagonal tubes of $\boldsymbol{\mathcal{S}_Y}$, where $\boldsymbol{\mathcal{S}_Y}$ is the f-diagonal tensor from the t-SVD of $\boldsymbol{\mathcal{Y}}$. That is $\mathrm{rank}_t(\boldsymbol{\mathcal{Y}}) := \#\{i : \boldsymbol{\mathcal{S}_Y}(i, i, :) \neq 0\}$. And its average rank is defined as $\mathrm{rank}_a(\boldsymbol{\mathcal{Y}}) = \frac{1}{k} \sum_{i}^{k} \mathrm{rank}(\overline{\boldsymbol{Y}}^{(i)})$. The condition number of a tensor $\boldsymbol{\mathcal{Y}} \in \mathbb{R}^{m \times n \times k}$ is defined as $\kappa(\boldsymbol{\mathcal{Y}}) = \frac{\sigma_1(\overline{\boldsymbol{Y}})}{\sigma_{\min}(\overline{\boldsymbol{Y}})}$, where $\overline{\boldsymbol{Y}}$ is the block diagonal matrix of tensor $\overline{\boldsymbol{\mathcal{Y}}}$ and $\sigma_1(\overline{\boldsymbol{Y}}) \geq \cdots \geq \sigma_{\min}(\overline{\boldsymbol{Y}}) > 0$ denotes the singular values of $\overline{\boldsymbol{Y}}$. For $\boldsymbol{\mathcal{Y}} \in \mathbb{R}^{m \times n \times k}$, its spectral norm is denoted as $\|\boldsymbol{\mathcal{Y}}\| := \|\mathtt{bcirc}(\boldsymbol{\mathcal{Y}})\| = \|\overline{\boldsymbol{Y}}\|$; its frobenius norm is defined as $\|\boldsymbol{\mathcal{Y}}\|_F := \sqrt{\sum_{i,j,l} \boldsymbol{\mathcal{Y}}(i, j, l)^2}$; its tubal tensor nuclear norm is defined as $\|\boldsymbol{\mathcal{Y}}\|_* := \|\overline{\boldsymbol{Y}}\|_*$(Karnik et al., 2025).

## 3 MAIN RESULTS

### 3.1 FACTORIZED GRADIENT DESCENT AND t-RIP

Firstly, we present the detailed update rule of the factorized gradient descent method for solving problem (1): $\boldsymbol{\mathcal{U}}_0 \sim \mathcal{N}(0, \frac{\alpha^2}{R})$, $\boldsymbol{\mathcal{U}}_{t+1} = \boldsymbol{\mathcal{U}}_t - \eta \cdot \frac{1}{m} \mathfrak{M}^* \left( \mathfrak{M}(\boldsymbol{\mathcal{U}}_t * \boldsymbol{\mathcal{U}}_t^{\top} - \boldsymbol{\mathcal{X}} * \boldsymbol{\mathcal{X}}^{\top}) - \boldsymbol{s} \right) * \boldsymbol{\mathcal{U}}_t$, where $\mathfrak{M}^*(\mathbf{e}) = \sum_{i=1}^{m} e_i \boldsymbol{\mathcal{A}}_i$ and $\boldsymbol{\mathcal{X}}_\star = \boldsymbol{\mathcal{X}} * \boldsymbol{\mathcal{X}}^{\top}$, $\boldsymbol{\mathcal{X}} \in \mathbb{R}^{n \times r \times k}$. A common assumption for analyzing the convergence of factorized gradient descent is the t-RIP, which is defined as follows:

**Definition 1** (t-RIP (Zhang et al., 2021)). *A linear map $\mathfrak{M} : \mathbb{R}^{n \times n \times k} \to \mathbb{R}^m$ is said to satisfy $(r, \delta)$ tensor Restricted Isometry Property (t-RIP) for $\delta \in [0, 1]$ if for any tensor $\boldsymbol{\mathcal{Y}} \in \mathbb{R}^{n \times n \times k}$ with tubal-rank $\leq r$, the following inequalities hold: $(1 - \delta)\|\boldsymbol{\mathcal{Y}}\|_F^2 \leq \|\mathfrak{M}(\boldsymbol{\mathcal{Y}})\|^2/m \leq (1 + \delta)\|\boldsymbol{\mathcal{Y}}\|_F^2$.*

The t-RIP condition has been shown to hold with high probability (Zhang et al., 2021) if $m \gtrsim rnk/\delta^2$, provided that each measurement tensor $\mathcal{A}_i$ in the operator $\mathfrak{M}$ has entries drawn independently from a sub-Gaussian distribution with zero mean and variance 1. Note that this condition has been extensively used in previous studies (Zhang et al., 2020; Liu et al., 2024b; Karnik et al., 2025), making it a natural and reasonable assumption in our setting.

We decompose the FGD update as

$$\boldsymbol{\mathcal{U}}_{t+1} = \boldsymbol{\mathcal{U}}_t - \eta(\boldsymbol{\mathcal{U}}_t * \boldsymbol{\mathcal{U}}_t^\top - \boldsymbol{\mathcal{X}}_\star) * \boldsymbol{\mathcal{U}}_t + \eta \underbrace{\left( \mathfrak{I} - \frac{\mathfrak{M}^* \mathfrak{M}}{m} \right) (\boldsymbol{\mathcal{U}}_t * \boldsymbol{\mathcal{U}}_t^\top - \boldsymbol{\mathcal{X}}_\star)}_{(a)} * \boldsymbol{\mathcal{U}}_t + \eta \cdot \underbrace{\frac{1}{m} \mathfrak{M}^*(\boldsymbol{s})}_{(b):=\boldsymbol{\mathcal{E}}} * \boldsymbol{\mathcal{U}}_t,$$

where $\mathfrak{I} : \mathbb{R}^{n \times n \times k} \to \mathbb{R}^{n \times n \times k}$ denotes the identity map. Then the t-RIP condition and tensor concentration bounds are applied to control terms (a) and (b) separately.

## 3.2 THEORETICAL GUARANTEES

We first establish theoretical guarantees for solving noisy low-tubal-rank tensor recovery via FGD with small initialization.

**Theorem 2.** *Assume the following assumptions hold: (1) the linear map $\mathfrak{M}$ satisfies $(2r + 1, \delta)$ t-RIP with $\delta \leq \frac{c}{\sqrt{k r} \kappa^4}$; (2) the step size $\eta \leq c\kappa^{-4}||\boldsymbol{\mathcal{X}}||^2$; (3) the error term $\boldsymbol{\mathcal{E}} := \frac{1}{m}\mathfrak{M}^*(\boldsymbol{s})$ satisfies $||\boldsymbol{\mathcal{E}}|| \leq c\kappa^{-2}\sigma_{\min}^2(\boldsymbol{\mathcal{X}})$; (4) each entry of the initial point $\boldsymbol{\mathcal{U}}_0$ is $i.i.d \, \mathcal{N}(0, \frac{\alpha^2}{R})$; (5) $\boldsymbol{\mathcal{X}}_\star \in \mathbb{R}^{n \times n \times k}$ is a full tubal-rank $r$ tensor. With all these assumptions, the following statements hold with probability at least $1 - ke^{-\tilde{c}R} - \max\{k(\tilde{C}\epsilon)^{R-r+1}, k\epsilon^2\}$,*

*1. When $R = r$, and the initialization scale satisfies $\alpha \lesssim \frac{\sqrt{r}\sigma_{\min}(\boldsymbol{\mathcal{X}})}{k^{23/14}(R \wedge n)\kappa^2} \left( \frac{2\kappa^2\sqrt{rn}}{\tilde{c}_3} \right)^{-10\kappa^2}$, then we have*

$$||\boldsymbol{\mathcal{U}}_{\hat{t}} * \boldsymbol{\mathcal{U}}_{\hat{t}}^\top - \boldsymbol{\mathcal{X}}_\star||_F \lesssim \sqrt{r}\kappa^2||\boldsymbol{\mathcal{E}}||, \text{ where } \hat{t} \gtrsim \frac{1}{\eta\sigma_{\min}^2(\boldsymbol{\mathcal{X}})} \ln \left( \frac{\kappa^2 r^{3/2}\sqrt{n}}{\sqrt{k}\alpha\sigma_{\min}(\boldsymbol{\mathcal{X}})} \right).$$

*2. When $r < R < 3r$, and initialization scale $\alpha$ satisfies $\alpha \lesssim \min\left\{ \frac{\sigma_{\min}(\boldsymbol{\mathcal{X}})}{(R \wedge n)\kappa^2}, \frac{\kappa^{\frac{35}{21}}||\boldsymbol{\mathcal{E}}||^{\frac{16}{21}}}{((R \wedge n) - r)^{\frac{4}{7}}||\boldsymbol{\mathcal{X}}||^{\frac{11}{21}}} \right\} \frac{r}{k^{23/14}} \left( \frac{2\kappa^2\sqrt{rn}}{\tilde{c}_3} \right)^{-10\kappa^2}$, then we have*

$$||\boldsymbol{\mathcal{U}}_{\hat{t}} * \boldsymbol{\mathcal{U}}_{\hat{t}}^\top - \boldsymbol{\mathcal{X}}_\star||_F \lesssim \sqrt{r}\kappa^2||\boldsymbol{\mathcal{E}}||, \text{ where } \hat{t} \asymp \frac{1}{\eta\sigma_{\min}^2(\boldsymbol{\mathcal{X}})} \ln \left( \frac{n^{\frac{1}{2}} r^{\frac{5}{2}}\kappa^2||\boldsymbol{\mathcal{X}}||^2}{k^2((R \wedge n) - r)\alpha^2} \right).$$

*3. When $R \geq 3r$, and the initialization scale satisfies $\alpha \lesssim \min\left\{ \frac{\sigma_{\min}(\boldsymbol{\mathcal{X}})}{(R \wedge n)\kappa^2}, \frac{\kappa^{\frac{35}{21}}||\boldsymbol{\mathcal{E}}||^{\frac{16}{21}}}{((R \wedge n) - r)^{\frac{4}{7}}||\boldsymbol{\mathcal{X}}||^{\frac{11}{21}}} \right\} \frac{1}{k^{23/14}} \left( \frac{2\kappa^2\sqrt{n}}{\tilde{c}_3\sqrt{(R \wedge n)}} \right)^{-10\kappa^2}$, then we have*

$$||\boldsymbol{\mathcal{U}}_{\hat{t}} * \boldsymbol{\mathcal{U}}_{\hat{t}}^\top - \boldsymbol{\mathcal{X}}_\star||_F \lesssim \sqrt{r}\kappa^2||\boldsymbol{\mathcal{E}}||, \text{ where } \hat{t} \asymp \frac{1}{\eta\sigma_{\min}(\boldsymbol{\mathcal{X}})^2} \ln \left( \frac{\sqrt{n}\kappa^2||\boldsymbol{\mathcal{X}}||^2}{k^2((R \wedge n) - r)(R \wedge n)\alpha^2} \right).$$

*Here, $c, \tilde{c}, \tilde{c}_3, \epsilon, \tilde{C}$ are fixed numerical constants, and we define $R \wedge n := \min\{R, n\}$, $\kappa := \kappa(\boldsymbol{\mathcal{X}})$.*

**Remark 1.** *(Recovery error) Our final recovery error is $\sqrt{r}\kappa^2||\boldsymbol{\mathcal{E}}||$, which depends only on the spectral norm of the noise term $\boldsymbol{\mathcal{E}}$, the condition number $\kappa$ of $\boldsymbol{\mathcal{X}}$, and the true tubal-rank $r$. We make no specific assumptions on the distribution of the noise, requiring only that $||\boldsymbol{\mathcal{E}}|| \leq c\kappa^{-2}\sigma_{\min}^2(\boldsymbol{\mathcal{X}})$. This makes our result potentially applicable to a wide range of noise distributions. When the noise is Gaussian noise, our bound reduces to that of (Liu et al., 2024b). However, a key difference is that our error bound depends only on the true tubal-rank $r$, whereas the bound in Liu et al. (2024b) depends on the overestimated tubal-rank $R$.*

Then we present a theorem that characterizes the minimax error in the Gaussian noise case. Theorem 3 establishes the fundamental statistical limit for low-tubal-rank tensor recovery. Specifically, for any estimation procedure, the mean squared error cannot uniformly fall below order $\Theta(nrk\sigma^2/m)$ over tensors of tubal-rank at most $r$. Furthermore, there exist parameter choices under which the error attains this order with constant probability.

**Theorem 3** (**Minimax error**). *Suppose that the linear map $\mathfrak{M}(\cdot)$ satisfies the $(r, \delta)$ t-RIP, $\boldsymbol{\mathcal{X}}_\star \in \mathbb{R}^{n \times n \times k}$ is a full tubal-rank $r$ tensor, and that $s \sim \mathcal{N}(0, \sigma^2 \boldsymbol{I})$, then any estimator $\boldsymbol{\mathcal{X}}_{est}$ obeys*

$$\sup_{\boldsymbol{\mathcal{X}}_\star} \mathbb{E} \|\boldsymbol{\mathcal{X}}_{est} - \boldsymbol{\mathcal{X}}_\star\|_F^2 \geq \frac{1}{1+\delta} \frac{nrk\sigma^2}{m}, \quad \sup_{\boldsymbol{\mathcal{X}}_\star} \mathbb{P}\left(\|\boldsymbol{\mathcal{X}}_{est} - \boldsymbol{\mathcal{X}}_\star\|_F^2 \geq \frac{nrk\sigma^2}{2m(1+\delta)}\right) \geq 1 - e^{-\frac{nrk}{16}}.$$

With the minimax error under Gaussian noise, we further show that when $s \sim \mathcal{N}(0, \sigma^2)$, FGD with small initialization converges to nearly optimal error.

**Corollary 1.** *(**Nearly minimax optimal error in Gaussian case**) Under the assumptions of Theorem 2, further assume that the entries of the noise vector $\boldsymbol{s}$ are Gaussian with zero mean and variance $\sigma^2$, and that the number of measurements satisfies $m \gtrsim nk\kappa^4 \frac{\sigma^2}{\sigma_{\min}^4(\boldsymbol{\mathcal{X}})}$. Then, with high probability, we have $\|\boldsymbol{\mathcal{U}}_{\hat{t}} * \boldsymbol{\mathcal{U}}_{\hat{t}}^\top - \boldsymbol{\mathcal{X}}_\star\|_F^2 \lesssim \frac{nkr\kappa^4 \sigma^2}{m}$, where $\hat{\boldsymbol{\mathcal{U}}}_t$ is the same as Theorem 2.*

**Remark 2.** *(**Sample complexity**) Our assumption on the number of measurements $m$ mainly comes from the t-RIP condition, which requires $m \gtrsim nkr/\delta^2$. In this work, we rely only on the $(2r + 1, \delta)$ t-RIP, without depending on the overestimated tubal-rank $R$, which is consistent with the setting in (Karnik et al., 2025). In contrast, (Liu et al., 2024b) requires the $(4R, \delta)$ t-RIP, leading to higher sample complexity as the overestimated tubal-rank $R$ increases. Note that this sampling complexity is required only for theoretical guarantees; in practice, a much smaller sample size suffices, as shown in Figure 2 (d).*

**Remark 3.** *(**Comparison with (Liu et al., 2024b)**) Both this work and (Liu et al., 2024b) employ factorized gradient descent algorithms to solve the low-tubal-rank tensor recovery problem. They are the first to apply FGD to this problem and provided convergence and recovery error analyses. However, our work differs significantly from them in several key aspects: (**1**) **Initialization:** They relies on spectral initialization to obtain a sufficiently good starting point for its theoretical analysis. In contrast, our method requires only a small random initialization to guarantee convergence. These two initialization strategies lead to entirely different analytical frameworks and theoretical results. (**2**) **Convergence rate:** In (Liu et al., 2024b), the convergence rate under over-parameterization is sublinear, whereas our analysis shows that the convergence rate remains linear even in the over-parameterized regime. (**3**) **Recovery error:** Their recovery error depends on the over-parameterized tubal rank $R$, while ours depends only on the true tubal rank $r$. (**4**) **Sampling complexity:** They require the measurement operator $\mathfrak{M}$ to satisfy the $(4R, \delta)$ t-RIP condition, whereas we only require $\mathfrak{M}$ to satisfy the $(2r + 1, \delta)$ t-RIP condition. As a result, the sampling complexity in (Liu et al., 2024b) grows with the degree of over-parameterization, while our requirement remains mild and independent of $R$.*

**Remark 4.** *(**Comparison with Karnik et al. (2025)**) Another related work is Karnik et al. (2025), which studies tubal tensor recovery under small initialization. Our work differs from theirs in several key aspects. (**1**) **Problem setting:** While they focus on the implicit regularization effect of small initialization, our goal is to provide theoretical guarantees for low-tubal-rank tensor recovery with noise under small initialization. (**2**) **Technical tools:** Their analysis splits the FGD trajectory into only two stages-the spectral stage and the convergence stage, which does not allow a precise characterization of the noise evolution. In contrast, we introduce a four-phase decomposition that provides a much finer description of the trajectory, enabling us to track the effect of noise throughout all stages. Consequently, directly extending their results to the noisy tensor setting does not yield minimax-optimal recovery guarantees. (**3**) **Theoretical results:** Our analysis requires less restrictive bounds on parameters. For example, the upper bound on the initialization scale $\alpha$ in our Theorem 2 is significantly more relaxed than that in [Karnik et al. (2025), Theorem 3.1].*

**Remark 5.** *(**Discussion with tubal-rank estimation methods**) Over the past five years, many low-tubal-rank tensor recovery methods with rank estimation strategies have been proposed (Shi et al., 2021; Zheng et al., 2023; Zhu et al., 2025). (**1**) **Problem setting:** Our goal is to achieve stable recovery even when the specified tubal-rank upper bound exceeds the true tubal-rank, ensuring that the error does not deteriorate as the upper bound increases. In contrast, tubal-rank estimation methods aim to identify or approximate the true tubal-rank. (**2**) **Noise models:** Shi et al. (2021) and Zhu et al. (2025) considered rank estimation in the presence of sparse noise, while Zheng et al. (2023) focuses on fast and robust rank estimation in the noiseless setting. Our results apply to the situation inthe presence of sub-Gaussian noise. (**3**) **Theoretical guarantees:** To the best of our knowledge, the above works do not provide rank-independent error bounds under the t-SVD and tubal-rank setting. Our main contribution is to establish such tubal-rank-independent guarantees and demonstrate near-minimax statistical accuracy.*

## 3.3 PROOF SKETCH

Define the tensor column subspace of $\mathcal{X}$ as $\mathcal{V}_{\mathcal{X}} \in \mathbb{R}^{n \times r \times k}$. Consider the tensor $\mathcal{V}_{\mathcal{X}}^{\top} * \mathcal{U}_t$ and the corresponding t-SVD $\mathcal{V}_{\mathcal{X}}^{\top} * \mathcal{U}_t = \mathcal{V}_t * \mathcal{S}_t * \mathcal{W}_t^{\top}$ with $\mathcal{W}_t \in \mathbb{R}^{R \times r \times k}$. And we denote $\mathcal{W}_{t,\perp} \in \mathbb{R}^{R \times (n-r) \times k}$ as a tensor whose tensor column subspace is orthogonal to the column subspace of $\mathcal{W}_t$. Then we can decompose $\mathcal{U}_t$ into "signal term" and "over-parameterization term":

$$\mathcal{U}_t = \underbrace{\mathcal{U}_t * \mathcal{W}_t * \mathcal{W}_t^{\top}}_{\text{signal term}} + \underbrace{\mathcal{U}_t * \mathcal{W}_{t,\perp} * \mathcal{W}_{t,\perp}^{\top}}_{\text{over-parameterization term}}. \tag{2}$$

Through this decomposition, we can separately analyze the signal term and the over-parameterization term. Specifically, we consider the following three quantities to study the convergence behavior of FGD:

- $\sigma_{\min}(\mathcal{U}_t * \mathcal{W}_t)$: the magnitude of the signal term;
- $\|\mathcal{U}_t * \mathcal{W}_{t,\perp}\|$: the magnitude of the over-parameterization term;
- $\|\mathcal{V}_{\mathcal{X}^{\perp}}^{\top} * \mathcal{V}_{\mathcal{U}_t * \mathcal{W}_t}\|$: the alignment between the column space of the signal and that of the ground truth.

Then we divide the trajectory of FGD into four phases:

**I. Alignment phase:** At this stage, the column space of the signal term $\mathcal{U}_t * \mathcal{W}_t$ gradually aligns with that of the ground truth $\mathcal{X}_{\star}$, as indicated by the decreasing value of $\|\mathcal{V}_{\mathcal{X}^{\perp}}^{\top} * \mathcal{V}_{\mathcal{U}_t * \mathcal{W}_t}\|$. Both $\sigma_{\min}(\mathcal{U}_t * \mathcal{W}_t)$ and $\|\mathcal{U}_t * \mathcal{W}_{t,\perp}\|$ remain small due to the small initialization.

**II. Signal amplification phase:** Here, $\sigma_{\min}(\mathcal{U}_t * \mathcal{W}_t)$ grows exponentially until it reaches at least $\frac{\sigma_{\min}(\mathcal{X})}{\sqrt{10}}$, while $\|\mathcal{U}_t * \mathcal{W}_{t,\perp}\|$ remains nearly at the scale of the initialization.

**III. Local refinement phase:** In this stage, using the decomposition (3), the error is decomposed as

$$\|\mathcal{U}_t * \mathcal{U}_t^{\top} - \mathcal{X}_{\star}\| \le 4\|\mathcal{V}_{\mathcal{X}}^{\top} * (\mathcal{U}_t * \mathcal{U}_t^{\top} - \mathcal{X}_{\star})\| + \|\mathcal{U}_t * \mathcal{W}_{t,\perp}\|^2.$$

The over-parameterization term $\|\mathcal{U}_t * \mathcal{W}_{t,\perp}\|^2$ remains small, while the in-subspace error $\|\mathcal{V}_{\mathcal{X}}^{\top} * (\mathcal{U}_t * \mathcal{U}_t^{\top} - \mathcal{X}_{\star})\|$ decreases rapidly, leading to the lowest recovery error.

**IV. Overfitting phase:** Eventually, the over-parameterization term $\|\mathcal{U}_t * \mathcal{W}_{t,\perp}\|$ starts to grow, which causes the overall error $\|\mathcal{U}_t * \mathcal{U}_t^{\top} - \mathcal{X}_{\star}\|_F$ to increase and approach the error of spectral initialization.

**The power of small initialization** Through the above four-phase analysis, we can see that small initialization plays a crucial role. Specifically, small initialization ensures that the signal term rapidly increases while keeping the over-parameterization term at a small magnitude, thereby mitigating the negative effects brought by over-parameterization. In particular, during Phase III, the over-parameterization term $\|\mathcal{U}_t * \mathcal{W}_{t,\perp}\|^2$ remains small, and $\|\mathcal{V}_{\mathcal{X}}^{\top} * (\mathcal{U}_t * \mathcal{U}_t^{\top} - \mathcal{X}_{\star})\|$ converges quickly. Moreover, due to the introduction of $\mathcal{V}_{\mathcal{X}}$, we have

$$\|\mathcal{V}_{\mathcal{X}}^{\top} * (\mathcal{U}_t * \mathcal{U}_t^{\top} - \mathcal{X}_{\star})\|_F \le \sqrt{r}\|\mathcal{V}_{\mathcal{X}}^{\top} * (\mathcal{U}_t * \mathcal{U}_t^{\top} - \mathcal{X}_{\star})\|,$$

which ensures that the final recovery error is independent of the over tubal-rank $R$.

**Remark 6.** *We assume that $\mathcal{X}_{\star}$ is symmetric and can be factorized as $\mathcal{X}_{\star} = \mathcal{X} * \mathcal{X}^{\top}$, which aligns with prior works (Liu et al., 2024b; Karnik et al., 2025). Extending to the general asymmetric case where $\mathcal{X}_{asym} \in \mathbb{R}^{m \times n \times k}$ is factorized as $\mathcal{L} * \mathcal{R}^{\top}$ requires several modifications. We provide a brief discussion here, with more details deferred to the Appendix I. First, a symmetrization step is needed to construct a symmetric tensor $\mathcal{X}_{sym} \in \mathbb{R}^{(m+n) \times (m+n) \times k}$ and its corresponding symmetric model. Second, the trajectories of the two factor tensors are coupled, making it necessary to analyze additional imbalance terms, an issue that does not arise in the symmetric setting.*

**Remark 7.** *(Comparison with (Ding et al., 2025)) Our framework reduces to the matrix setting when $n_3 = 1$: the t-product becomes matrix multiplication, tubal-rank becomes matrix rank, and $\mathcal{X} = \mathcal{U} * \mathcal{U}^{\top}$ reduces to $X = UU^{\top}$. In this special case, Theorem 2 recovers the same qualitative phenomenon reported for matrix FGD: small initialization and early stopping yield error bounds that do not deteriorate with the over-specified rank, as shown in literature (Ding et al., 2025). However, extending the matrix setting to the tensor setting is nontrivial, one must address several challenges unique to tensors, as discussed in Remark 8.*

**Remark 8.** *(Tensor specific challenges) First, in the matrix case, the range and the kernel are complementary subspaces. This property no longer holds for third-order tubal tensors. If the true tensor contains non-invertible tubes in its t-SVD, equivalently, if some frequency slices vanish in the Fourier domain, then the range and kernel share common generators. As a result, the classical decomposition of gradient updates into a "signal term" and a "over-parameterization term" fails on these non-invertible tubes. This necessitates introducing a more refined notion of tensor condition number to track the identifiable and unidentifiable components separately. Second, for the power method, each frequency slice of a tubal tensor behaves like an independent matrix power iteration, a known fact in the (gle, 2013). However, in gradient descent for tensor recovery, the measurement operator and its adjoint couple information across all frequency slices. Consequently, the update of any single slice depends on all other slices, making it impossible to analyze the slices independently, as in the power method. Finally, in the matrix setting, Candes & Plan (2011) has already established the minimax error for noisy matrix sensing. To the best of our knowledge, however, no such minimax error analysis exists for the tensor setting.*

### 3.4 EARLY STOPPING VIA VALIDATION

Although Theorem 2 provides the sharpest known error bound, it is clear that the choice of $\hat{t}$ depends on prior knowledge of $\mathcal{X}_\star$, which is often unavailable in practice. As shown in Figure 1, setting $\hat{t}$ too small or too large can lead to increased error. A practical solution is to use validation to determine when to stop the algorithm, a common technique in machine learning (Prechelt, 1998; Stone, 2018; Ding et al., 2025). Specifically, we randomly split the observed data $\{\mathcal{A}_i, y_i\}_{i=1}^m$ into a training set $(\boldsymbol{y}_{\text{train}}, \mathfrak{M}_{\text{train}})$ of size $m_{\text{train}}$ and a validation set $(\boldsymbol{y}_{\text{val}}, \mathfrak{M}_{\text{val}})$ of size $m_{\text{val}}$. We then perform gradient descent using the training set. After each iteration, we compute the validation loss $e_t = \frac{1}{4}||\boldsymbol{y}_{\text{val}} - \mathfrak{M}_{\text{val}}(\mathcal{U}_t * \mathcal{U}_t^\top)||^2$. The final estimate is selected as $\check{t} = \arg\min_t e_t$, and we output $\mathcal{U}_{\check{t}} * \mathcal{U}_{\check{t}}^\top$ as the recovered tensor. The full procedure is described in Algorithm 2, Appendix D.

We then provide a theoretical guarantee showing that, when $\check{t} = \arg\min_{1 \le \check{t} \le T} e_t$, the recovery error $||\mathcal{U}_{\check{t}} * \mathcal{U}_{\check{t}}^\top - \mathcal{X}_\star||_F$ achieves the bound stated in Theorem 2.

**Theorem 4.** *Assume the same conditions as in Theorem 2, except that $(\boldsymbol{y}, \mathfrak{M})$ is replaced by $(y_{train}, \mathfrak{M}_{train})$. In addition, suppose that $m_{val} \ge C_1 \frac{m_{train}^2 \log T}{(rnk\kappa^4)^2}$, and $T$ be the max $\hat{t}$ in Theorem 2. Assume that each entry of the noise vector $\boldsymbol{s}$ is independently sampled from the Gaussian distribution $\mathcal{N}(0, \sigma^2)$. Define $\check{t} = \arg\min_{1 \le t \le T} e_t$. Then, with probability at least $1 - 2T \exp\left(-\frac{C_2 (nkr\kappa^4)^2 m_{val}}{m_{train}^2}\right)$, $||\mathcal{U}_{\check{t}} * \mathcal{U}_{\check{t}}^\top - \mathcal{X}_\star||_F^2 \le C \frac{nkr\sigma^2\kappa^4}{m_{train}}$.*

**Remark 9.** *In Theorem 2, we require $m_{train} \gtrsim nkr^2\kappa^8$. Substituting this into the condition $m_{val} \ge C_1 \frac{m_{train}^2 \log T}{(rnk\kappa^4)^2}$, we obtain $m_{val} \gtrsim r^2\kappa^8 \log T$. This is relatively small compared to $m_{train}$, making it practically feasible. Experiments also show that a relatively small $m_{val}$ suffices to achieve an error close to that under the exact tubal-rank.*

## 4 EXPERIMENTS

We present a series of experiments demonstrating that, under over-rank settings, using small initialization combined with validation achieves recovery error comparable to that under exact parameterization. Compared to FGD with large random or spectral initialization (Liu et al., 2024b), our method achieves the lowest recovery error, highlighting the unique effectiveness of small initialization. Additional simulation studies and real-data experiments are presented in Appendix J.

**Experiments settings** We first generate a ground-truth tensor $\mathcal{X}_\star \in \mathbb{R}^{n \times n \times k}$ of tubal-rank $r$ by setting $\mathcal{X}_\star = \mathcal{X} * \mathcal{X}^\top$, where $\mathcal{X} \in \mathbb{R}^{n \times r \times k}$ has entries independently drawn from a Gaussian distribution $\mathcal{N}(0, 1)$. Next, we normalize the tensor by setting $\mathcal{X}_\star \leftarrow \mathcal{X}_\star / ||\mathcal{X}_\star||_F$. We sample the measurement operator $\mathfrak{M}$ by selecting each entry independently from a Gaussian distribution $\mathcal{N}(0, 1)$. The noise vector $\boldsymbol{s}$ has entries independently drawn from $\mathcal{N}(0, \sigma^2)$. Finally, the observations are obtained via the measurement model $\boldsymbol{y} = \mathfrak{M}(\mathcal{X}_\star) + \boldsymbol{s}$. In all experiments, we set $m = 2C_m nrk$, and $n = 30$, $k = 3$, $r = 3$, $m_{\text{val}} = 0.05m$. For FGD with small initialization, we set the initialization scale to $\alpha = 10^{-10}$. For FGD with spectral initialization, we follow the same initialization procedure as in the original paper. For FGD with large initialization, we set the initialization scale to $\alpha = 10$, with its step size $\eta = 0.001$ to prevent divergence. We use FGD with

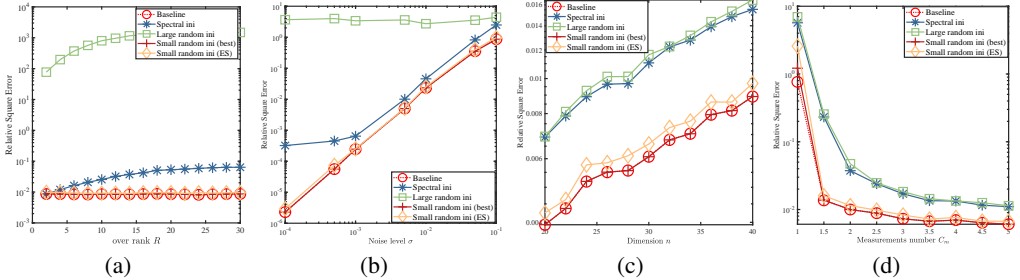

Figure 2: Performance comparison under varying $r$, $\sigma$, $n$, and $m$. Subfigure (a) illustrates the recovery error of all methods under different over-rank values $R$, with parameters set as $m = 10nrk$, $n = 30$, $\sigma = 10^{-3}$, $\eta = 0.1$, and $T = 5000$. Subfigure (b) illustrates the error under varying noise levels $\sigma$, with $m = 10nrk$, $n = 30$, $R = 3r$, $\eta = 0.1$, and $T = 5000$. Subfigure (c) illustrates the error as the problem dimension $n$ changes, where $m = 10nrk$, $R = 3r$, $\eta = 0.1$, $T = 20000$, and $\sigma = 10^{-3}$. Subfigure (d) illustrates the performance under different numbers of measurements $C_m$, with $m = 2C_m nrk$, $n = 30$, $R = 3r$, $\eta = 0.01$, $T = 20000$, and $\sigma = 10^{-3}$.

the exact rank as a baseline method, where "Small random ini (best)" denotes the minimal error obtained by FGD with small random initialization and "Small random ini (ES)" denotes the error obtained by FGD with small random initialization using validation and early stopping. We use the relative square error (RSE) $\frac{||\mathcal{U}_t * \mathcal{U}_t^\top - \mathcal{X}_\star||_F^2}{||\mathcal{X}_\star||_F^2}$ to evaluate the performance of different methods and all experiments are repeated 20 times.

**Comparison of different initialization methods** From Figure 2, we make these observations:

1. In all four settings, using small initialization yields the same minimum error as the baseline method, which demonstrates its effectiveness. Moreover, by combining small initialization with validation-based early stopping, we can achieve errors very close to the baseline without requiring any prior knowledge of the target tensor. This supports the conclusions of Theorems.

2. For spectral initialization and large random initialization, the recovery error increases as the over-estimated rank grows, and remains higher than that of small initialization. The error from large random initialization is particularly high due to its slow convergence. However, in the experiment shown in Figure 2 (c) and (d), where the number of iterations is large enough, its error matches that of spectral initialization.

3. As shown in Figure 2 (d), small initialization also significantly reduces sample complexity. Even when $m = 3nrk$, it still achieves low error, clearly outperforming the other initialization methods.

**Verify the validation and early stopping approach** We verify the effectiveness of the validation and early stopping strategies. As shown in Figure 3 (a), the relative recovery error is minimized when the validation loss reaches its lowest point, demonstrating the reliability of using validation loss as a stopping criterion. Figure 3 (b) shows that when too many samples are used for validation, the recovery error increases compared to the minimum achievable error due to insufficient training data. Conversely, when too few samples (less than 5%) are used for validation, the validation-based method may become unreliable, resulting in increased recovery error. Therefore, allocating 5%–10% of the total samples for validation is a reasonable choice.

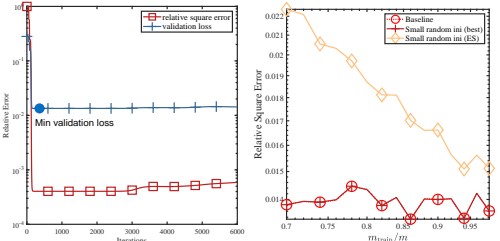

Figure 3: Validation of the algorithm with $m = 10nrk$, $R = 3r$, $n = 30$, $\sigma = 10^{-3}$, $\eta = 0.1$. (a) Validation loss vs. RSE, with the blue dot marking the minimum. (b) Error of the validation-based method compared with the minimum errors of baseline and small-initialization under varying $m_{\text{train}}$.

**Real data experiments on tensor completion**

We conduct real-data experiments on the low-tubal-rank tensor completion problem. We consider the problem of low-tubal-rank tensor completion under the Bernoulli observation model. Let the target tensor be $\mathcal{X}_\star \in \mathbb{R}^{n_1 \times n_2 \times n_3}$ with unknown tubal-rank $r$, where each entry is independently observed with probability $p$. Denote the set of observed indices by $\mathbf{\Omega} \subseteq [n_1] \times [n_2] \times [n_3]$, and

Table 2: Comparison of different methods in terms of average Peak Signal-to-Noise Ratio (PSNR) and average Relative Error (RE) under various sampling rates and noise levels. "FGD-ES" denotes FGD with early stopping, while "FGD-best" refers to the minimum error achieved by FGD over all iterations. We write GTNN-HOP$_{0.3}$ as GTNN for short.

| Methods | $p = 0.2$ | | | | $p = 0.3$ | | | |
|---|---|---|---|---|---|---|---|---|
| | $\sigma = 0.07$ | | $\sigma = 0.1$ | | $\sigma = 0.07$ | | $\sigma = 0.1$ | |
| | PSNR $\uparrow$ | RE $\downarrow$ | PSNR $\uparrow$ | RE $\downarrow$ | PSNR $\uparrow$ | RE $\downarrow$ | PSNR $\uparrow$ | RE $\downarrow$ |
| TCTF | 16.5892 | 0.3175 | 16.5484 | 0.3191 | 20.6744 | 0.2008 | 20.6335 | 0.2024 |
| TNN | 21.2692 | 0.1851 | 19.7672 | 0.2188 | 22.0592 | 0.1681 | 20.1682 | 0.2082 |
| TC-RE | 20.9288 | 0.1921 | 19.5480 | 0.2242 | 21.5387 | 0.1782 | 19.8376 | 0.2161 |
| UTF | 16.3227 | 0.3243 | 14.8770 | 0.3802 | 19.2245 | 0.2355 | 17.8283 | 0.2734 |
| GTNN | 22.1092 | 0.1675 | 20.3132 | 0.2051 | 23.1542 | 0.1481 | 21.1111 | 0.1867 |
| FGD-ES | 22.5912 | 0.1616 | 21.7977 | 0.1765 | 23.6579 | 0.1426 | 22.7157 | 0.1585 |
| FGD-best | **22.7438** | **0.1587** | **21.9268** | **0.1739** | **23.8422** | **0.1395** | **22.8550** | **0.1559** |

define the observation operator as $\mathfrak{P}_\Omega(\mathcal{A}) = \mathbf{\Omega} \odot \mathcal{A}$, where $\odot$ denotes the Hadamard product. The goal is to accurately recover the low-tubal-rank tensor $\mathcal{X}_\star$ from the partial and noisy observations $\mathfrak{P}_\Omega(\mathcal{X}_\star + \mathcal{S}_n)$, where $\mathcal{S}_n$ is assumed to be Gaussian noise with entries i.i.d sampled from Gaussian distribution $\mathcal{N}(0, \sigma^2)$ in this paper. Under the t-product framework, we adopt the Burer–Monteiro factorization $\mathcal{L} * \mathcal{R}^\top$, where $\mathcal{L} \in \mathbb{R}^{n_1 \times R \times n_3}, \mathcal{R} \in \mathbb{R}^{n_2 \times R \times n_3}$. The recovery is formulated by minimizing the following factorized loss function: $f(\mathcal{L}, \mathcal{R}) = \frac{1}{2p}||\mathfrak{P}_\Omega(\mathcal{L} * \mathcal{R}^\top - \mathcal{X}_\star - \mathcal{S}_n)||_F^2$, which can be optimized using gradient descent over $(\mathcal{L}, \mathcal{R})$.

Then we perform color image completion experiments on the Berkeley Segmentation Dataset (Martin et al., 2001). We randomly select 50 color images of size $481 \times 321 \times 3$. We compare three categories of methods: a convex approach: tubal tensor nuclear norm Minimization (TNN) (Lu et al., 2018), non-convex methods: UTF (Du et al., 2021) and GTNN-HOP (Wang et al., 2024), and rank estimation-based methods: TCTF (Zhou et al., 2017) and TC-RE (Shi et al., 2021). We use PSNR and RE as evaluation metrics, and for more detailed experiments settings, please refer to Appendix J.2. The results, shown in Table 2, demonstrate that FGD with small initialization significantly outperforms all other methods, while FGD with early stopping performs slightly worse but remains acceptable. Therefore, even though the tensor completion problem does not require the t-RIP assumption, FGD with small initialization still achieves the lowest reconstruction error. In addition, we evaluate the sensitivity of the algorithm to different tubal ranks. As shown in Figure 4, choosing different values of $R$ has only a minor effect on the recovery performance. Therefore, when the true rank is unknown, selecting a slightly larger rank for recovery is a practical and effective strategy. Moreover, experiments on video completion are presented in Appendix J.2.

## 5 CONCLUSION

We propose a novel procedure, that is, factorized gradient descent with small initialization, to solve the noisy low-tubal-rank tensor recovery problem. We prove that, even when the tubal-rank is overestimated, the recovery error still depends only on the exact tubal-rank $r$, and is independent of the overestimated tubal-rank $R$. This significantly improves upon the error bound in (Liu et al., 2024b), and to the best of our knowledge, is the first error bound for noisy low-tubal-rank tensor recovery that does not depend on the overestimated tubal-rank and is nearly minimax optimal. Moreover, we demonstrate that this error bound can be achieved though a validation and early stopping procedure , without requiring any prior knowledge of the underlying tensor. Numerical experiments are further conducted to support our theoretical findings.

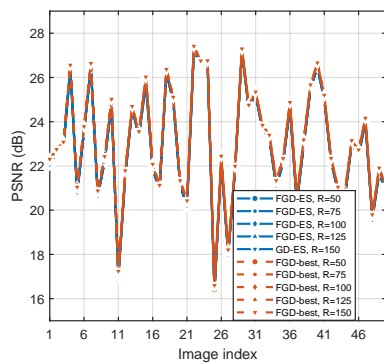

Figure 4: Validation of the sensitivity of FGD to different tubal-ranks.

ACKNOWLEDGMENTS

This work was supported in part by the National Natural Science Foundation of China under Grant T2596040, T2596045 and U23A20343, CAS Project for Young Scientists in Basic Research, Grant YSBR-041, Liaoning Provincial "Selecting the Best Candidates by Opening Competition Mechanism" Science and Technology Program under Grant 2023JH1/10400045, Fundamental Research Project of SIA under Grant 2024JC3K01, Natural Science Foundation of Liaoning Province under Grant 2025-BS-0193, Joint Innovation Fund of DICP & SIA under Grant UN202401.

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

CONTENTS

## A  ORGANIZATION OF APPENDIX

The Appendix is organized as follows:

- Section B provides the statement on the use of large language models.
- Section C presents the reproducibility statement.
- Section D introduces additional preliminaries supporting the main theoretical results.
- Section E gives the detailed proof of Theorem 2 and Corollary 1.
- Section F gives the detailed proof of Theorem 3.
- Section G gives the detailed proof of Theorem 4.
- Section H presents several technical lemmas together with their proofs.
- Section I discusses the extension to asymmetric case.
- Section J reports additional simulation results under various noise distributions, along with real-data experiments.

## B  USE OF LARGE LANGUAGE MODELS

We used GPT-5 exclusively for language polishing and grammatical refinement of this manuscript. The model was not involved in conceiving research ideas, developing algorithms, conducting experiments, or analyzing results. The authors take full responsibility for the technical content, theoretical contributions, and experimental findings presented in this work.

## C  REPRODUCIBILITY STATEMENT

All theoretical results in this paper are fully supported by detailed proofs provided in the appendix. In addition, the code used for the experiments is included in the supplementary material to ensure that all results reported in the paper can be reproduced.

## D  ADDITIONAL PRELIMINARIES

For two positive scalars $x, y$, $x \lesssim y$ (or $x \gtrsim y$) denotes that there exists a universal constant $z > 0$ such that $x \leq zy$ (or $x \geq zy$), and $x \asymp y$ denotes that there exit two universal constants $z_1, z_2 > 0$ such that $z_1 x \leq y \leq z_2 x$.

**Definition 2** (Symmetry and positive semi-definite tensor). *A three order tensor $\mathcal{A} \in \mathbb{R}^{n \times n \times k}$ is called symmetry and positive semi-definite if it satisfies the following condition:*

$$\mathcal{A}^\top = \mathcal{A}, and \, \overline{\mathcal{A}}^{(i)} \, is \, positive \, semi\text{-}definite.$$

**Definition 3** (Block diagonal matrix). *For any tensor $\mathcal{Y} \in \mathbb{R}^{m \times n \times k}$, we denote $\bar{Y} \in \mathbb{C}^{mk \times nk}$ as a block diagonal matrix with it's $i$-th block on the diagonal as the $i$-th frontal slice $\bar{Y}^{(i)}$ of $\bar{\mathcal{Y}}$, i.e.,*

$$\bar{Y} = \mathtt{bdiag}(\bar{\mathcal{Y}}) = \begin{bmatrix} \bar{Y}^{(1)} & & & \\ & \bar{Y}^{(2)} & & \\ & & \ddots & \\ & & & \bar{Y}^{(n_3)} \end{bmatrix}.$$

**Definition 4** (Block circulant matrix (Kilmer & Martin, 2011)). *For a three-order tensor $\mathcal{A} \in \mathbb{R}^{n_1 \times n_2 \times n_3}$, we denote $\mathtt{bcirc}(\mathcal{A}) \in \mathbb{R}^{n_1 n_3 \times n_2 n_3}$ as its block circulant matrix, i.e.,*

$$\mathtt{bcirc}(\mathcal{A}) = \begin{bmatrix} A^{(1)} & A^{(n_3)} & \cdots & A^{(2)} \\ A^{(2)} & A^{(1)} & \cdots & A^{(3)} \\ \vdots & \vdots & \ddots & \vdots \\ A^{(n_3)} & A^{(n_3-1)} & \cdots & A^{(1)} \end{bmatrix}.$$

**Definition 5** (The fold and unfold operations (Kilmer & Martin, 2011)). *For a three-order tensor* $\boldsymbol{\mathcal{A}} \in \mathbb{R}^{n_1 \times n_2 \times n_3}$, *we have*

$$\text{unfold}(\boldsymbol{\mathcal{A}}) = [A^{(1)}; A^{(2)}; \cdots ; A^{(n_3)}]$$
$$\text{fold}(\text{unfold}(\boldsymbol{\mathcal{A}})) = \boldsymbol{\mathcal{A}}.$$

**Definition 6** (T-product(Kilmer & Martin, 2011)). *For* $\boldsymbol{\mathcal{A}} \in \mathbb{R}^{n_1 \times n_2 \times n_3}$, $\boldsymbol{\mathcal{B}} \in \mathbb{R}^{n_2 \times q \times n_3}$, *the t-product of* $\boldsymbol{\mathcal{A}}$ *and* $\boldsymbol{\mathcal{B}}$ *is* $\boldsymbol{\mathcal{C}} \in \mathbb{R}^{n_1 \times q \times n_3}$, *i.e.,*

$$\boldsymbol{\mathcal{C}} = \boldsymbol{\mathcal{A}} * \boldsymbol{\mathcal{B}} = \text{fold}(\text{bcirc}(\boldsymbol{\mathcal{A}}) \cdot \text{unfold}(\boldsymbol{\mathcal{B}})).$$

*The t-product can also be computed by Algorithm 1.*

**Definition 7** (Identity tensor(Kilmer & Martin, 2011)). *The identity tensor, represented by* $\boldsymbol{\mathcal{I}} \in \mathbb{R}^{n \times n \times n_3}$, *is defined such that its first frontal slice corresponds to the* $n \times n$ *identity matrix, while all subsequent frontal slices are comprised entirely of zeros. This can be expressed mathematically as:*

$$\boldsymbol{\mathcal{I}}^{(1)} = \boldsymbol{I}_{n \times n}, \quad \boldsymbol{\mathcal{I}}^{(i)} = 0, i = 2, 3, \ldots, n_3.$$

**Definition 8** (Orthogonal tensor (Kilmer & Martin, 2011)). *A tensor* $\boldsymbol{\mathcal{Q}} \in \mathbb{R}^{n \times n \times n_3}$ *is considered orthogonal if it satisfies the following condition:*

$$\boldsymbol{\mathcal{Q}}^\top * \boldsymbol{\mathcal{Q}} = \boldsymbol{\mathcal{Q}} * \boldsymbol{\mathcal{Q}}^\top = \boldsymbol{\mathcal{I}}.$$

**Definition 9** (F-diagonal tensor (Kilmer & Martin, 2011)). *A tensor is called f-diagonal if each of its frontal slices is a diagonal matrix.*

**Theorem 5** (t-SVD (Kilmer & Martin, 2011; Lu et al., 2018)). *Let* $\boldsymbol{\mathcal{A}} \in \mathbb{R}^{n_1 \times n_2 \times n_3}$, *then it can be factored as*

$$\boldsymbol{\mathcal{A}} = \boldsymbol{\mathcal{U}} * \boldsymbol{\mathcal{S}} * \boldsymbol{\mathcal{V}}^\top,$$

*where* $\boldsymbol{\mathcal{U}} \in \mathbb{R}^{n_1 \times n_1 \times n_3}$, $\boldsymbol{\mathcal{V}} \in \mathbb{R}^{n_2 \times n_2 \times n_3}$ *are orthogonal tensors, and* $\boldsymbol{\mathcal{S}} \in \mathbb{R}^{n_1 \times n_2 \times n_3}$ *is a f-diagonal tensor.*

**Definition 10** (Tubal-rank (Kilmer & Martin, 2011)). *For* $\boldsymbol{\mathcal{A}} \in \mathbb{R}^{n_1 \times n_2 \times n_3}$, *its tubal-rank as* $\text{rank}_t(\boldsymbol{\mathcal{A}})$ *is defined as the nonzero diagonal tubes of* $\boldsymbol{\mathcal{S}}$, *where* $\boldsymbol{\mathcal{S}}$ *is the f-diagonal tensor from the t-SVD of* $\boldsymbol{\mathcal{A}}$. *That is*

$$\text{rank}_t(\boldsymbol{\mathcal{A}}) := \#\{i : S(i, i, :) \neq 0\}.$$

---

**Algorithm 1** Tensor-Tensor Product

**Input:** $\boldsymbol{\mathcal{Y}} \in \mathbb{R}^{n_1 \times n_2 \times n_3}$, $\boldsymbol{\mathcal{Z}} \in \mathbb{R}^{n_2 \times n_4 \times n_3}$.
**Output:** $\boldsymbol{\mathcal{X}} = \boldsymbol{\mathcal{Y}} * \boldsymbol{\mathcal{Z}} \in \mathbb{R}^{n_1 \times n_4 \times n_3}$.
 1: Compute $\bar{\boldsymbol{\mathcal{Y}}} = \text{fft}(\boldsymbol{\mathcal{Y}}, [], 3)$ and $\bar{\boldsymbol{\mathcal{Z}}} = \text{fft}(\boldsymbol{\mathcal{Z}}, [], 3)$
 2: Compute each frontal slice of $\bar{\boldsymbol{\mathcal{C}}}$ by

$$\bar{\mathbf{X}}^{(i)} = \begin{cases} \bar{\mathbf{Y}}^{(i)} \bar{\mathbf{Z}}^{(i)}, & i = 1, \ldots, \left\lceil \dfrac{n_3 + 1}{2} \right\rceil, \\ \text{conj}(\bar{\mathbf{X}}^{(n_3 - i + 2)}), & i = \left\lceil \dfrac{n_3 + 1}{2} \right\rceil + 1, \ldots, n_3. \end{cases}$$

 3: Compute $\boldsymbol{\mathcal{X}} = \text{ifft}((\bar{\boldsymbol{\mathcal{X}}}), [], 3)$.

---

The t-SVD of a tensor $\boldsymbol{\mathcal{Y}} \in \mathbb{R}^{n \times r \times k}$ as $\boldsymbol{\mathcal{Y}} = \boldsymbol{\mathcal{V}}_{\boldsymbol{\mathcal{Y}}} * \boldsymbol{\mathcal{S}}_{\boldsymbol{\mathcal{Y}}} * \boldsymbol{\mathcal{W}}_{\boldsymbol{\mathcal{Y}}}^\top$. In addition, we define $\boldsymbol{\mathcal{V}}_{\boldsymbol{\mathcal{Y}}}$ as the tensor-column subspace of $\boldsymbol{\mathcal{Y}}$, and $\boldsymbol{\mathcal{V}}_{\boldsymbol{\mathcal{Y}}^\perp}$ as its orthogonal complement, i.e., $\boldsymbol{\mathcal{V}}_{\boldsymbol{\mathcal{Y}}}^\top * \boldsymbol{\mathcal{V}}_{\boldsymbol{\mathcal{Y}}^\perp} = 0$.

Based on the t-RIP condition, we introduce the following two definitions to facilitate our analysis.

**Definition 11.** *(S2S-t-RIP) A linear map* $\mathfrak{M} : \mathbb{R}^{n \times n \times k} \to \mathbb{R}^m$ *is said to satisfy the spectral-to-spectral* $(r, \delta)$ *tensor Restricted Isometry Property (t-RIP)* $[(r, \delta)$ *S2S-t-RIP] if for all tensors* $\boldsymbol{\mathcal{Y}} \in \mathbb{R}^{n \times n \times k}$ *with tubal-rank* $\leq r$,

$$\left\| \left( \mathfrak{I} - \frac{\mathfrak{M}^* \mathfrak{M}}{m} \right) (\boldsymbol{\mathcal{Y}}) \right\| \leq \delta \|\boldsymbol{\mathcal{Y}}\|.$$

**Definition 12.** *(S2N-t-RIP) A linear map* $\mathfrak{M} : \mathbb{R}^{n \times n \times k} \to \mathbb{R}^m$ *is said to satisfy the spectral-to-nuclear* $\delta$ *tensor Restricted Isometry Property (t-RIP) [$\delta$-S2N-t-RIP] if for all tensors* $\mathcal{Y} \in \mathbb{R}^{n \times n \times k}$ *with tubal-rank* $\leq r$,

$$\left\| \left( \mathfrak{I} - \frac{\mathfrak{M}^* \mathfrak{M}}{m} \right) (\mathcal{Y}) \right\| \leq \delta \|\mathcal{Y}\|_*.$$

Then, we provide the detailed pseudocode of Algorithm 2 described in Section 3.4.

---

**Algorithm 2** Solving (1) by FGD with early stopping

---

**Input:** Train data $(\boldsymbol{y}_{\text{train}}, \mathfrak{M}_{\text{train}})$, validation data $(\boldsymbol{y}_{\text{val}}, \mathfrak{M}_{\text{val}})$, initialization scale $\alpha$, step size $\eta$, estimated tubal-rank $R$, iteration number T

**Initialization:** Initialize $\mathcal{U}_0$, where each entry of $\mathcal{U}_0$ is i.i.d. from $\mathcal{N}(0, \frac{\alpha^2}{R})$.

1: **for** $t = 0$ to $T - 1$ **do**
2: $\quad \mathcal{U}_{t+1} = \mathcal{U}_t - \frac{\eta}{m} \mathfrak{M}^*_{\text{train}}(\mathfrak{M}_{\text{train}}(\mathcal{U}_t * \mathcal{U}_t^\top) - \boldsymbol{y}_{\text{train}}) * \mathcal{U}_t$
3: $\quad$ Validation loss: $e_t = \frac{1}{2m} \|\boldsymbol{y}_{\text{val}} - \mathfrak{M}_{\text{val}}(\mathcal{U}_t * \mathcal{U}_t^\top)\|^2$
4: **end for**
5: **Output:** $\mathcal{U}_{\check{t}}$ where $\check{t} = \arg\min_{1 \leq t \leq T} e_t$.

---

# E  PROOF OF THEOREM 2

In this section, we absorb the additional $\frac{1}{\sqrt{m}}$ factor into $\mathfrak{M}$ for the convenience of presentation, i.e., $\mathcal{A}_i \leftarrow \mathcal{A}_i / \sqrt{m}$. Thus, we have $(1 - \delta)\|\mathcal{Y}\|_F^2 \leq \|\mathfrak{M}(\mathcal{Y})\|^2 \leq (1 + \delta)\|\mathcal{Y}\|_F^2$.

## E.1  ANALYSIS THE FOUR PHASES

Define the tensor column subspace of $\mathcal{X}$ as $\mathcal{V}_{\mathcal{X}} \in \mathbb{R}^{n \times r \times k}$. Consider the tensor $\mathcal{V}_{\mathcal{X}}^\top * \mathcal{U}_t$ and the corresponding t-SVD $\mathcal{V}_{\mathcal{X}}^\top * \mathcal{U}_t = \mathcal{V}_t * \mathcal{S}_t * \mathcal{W}_t^\top$ with $\mathcal{W}_t \in \mathbb{R}^{r \times R \times k}$. And we denote $\mathcal{W}_{t,\perp}$ as a tensor whose tensor column subspace is orthogonal to the column subspace of $\mathcal{W}_t$. Then we can decompose $\mathcal{U}_t$ into "signal term" and "over-parameterization term":

$$\mathcal{U}_t = \underbrace{\mathcal{U}_t * \mathcal{W}_t * \mathcal{W}_t^\top}_{\text{signal term}} + \underbrace{\mathcal{U}_t * \mathcal{W}_{t,\perp} * \mathcal{W}_{t,\perp}^\top}_{\text{over-parameterization term}}. \tag{3}$$

Through this decomposition, we can separately analyze the signal term and the over-parameterization term. Specifically, we consider the following three quantities to study the convergence behavior of FGD:

- $\sigma_{\min}(\mathcal{U}_t * \mathcal{W}_t)$: the magnitude of the signal term;
- $\|\mathcal{U}_t * \mathcal{W}_{t,\perp}\|$: the magnitude of the over-parameterization term;
- $\|\mathcal{V}_{\mathcal{X}^\perp}^\top * \mathcal{V}_{\mathcal{U}_t * \mathcal{W}_t}\|$: the alignment between the column space of the signal and that of the ground truth.

Using these three indicators and the recovery error $\|\mathcal{U}_t * \mathcal{U}_t^\top - \mathcal{X}_\star\|_F$, we identify four phases in the FGD trajectory and analyze them one by one.

### E.1.1  PHASE I: ALIGNMENT PHASE

In the first phase, Lemma 1 states that if the initialization scale is sufficiently small, and under appropriate t-RIP conditions, step size constraints, and an upper bound on the noise spectral norm, the signal term is nearly aligned with the column space of the ground truth tensor $\mathcal{X}_\star$. At this stage, both the magnitude of the signal term and that of the over-parameterization term remain small, but the former is significantly larger than the latter.

**Lemma 1.** *Fix a sufficiently small constant $c > 0$. Let $\mathcal{U} \in \mathbb{R}^{n \times R \times k}$ be a random tubal tensor with i.i.d. $\mathcal{N}(0, \frac{\alpha^2}{R})$ entries, and let $\epsilon \in (0,1)$. Assume that $\mathfrak{M} : \mathbb{R}^{n \times n \times k} \to \mathbb{R}^m$ satisfies the $\delta_1$-S2R-t-RIP for some constant $\delta_1 > 0$. Also, assume that*

$$\mathcal{M} := \mathfrak{M}^* \mathfrak{M}(\mathcal{X} * \mathcal{X}^\top) + \mathcal{E} = \mathcal{X} * \mathcal{X}^\top + \mathcal{E}_{\mathcal{X}}$$

*with $\|\mathcal{E}_{\mathcal{X}}^{(j)}\| \leq \delta \lambda_r(\overline{\mathcal{X}}^{(j)}(\overline{\mathcal{X}}^{(j)})^{\mathrm{H}})$ for each $1 \leq j \leq k$, where $\delta \leq c_1 \kappa^{-2}$ and $\|\mathcal{E}\| \leq c_1 \kappa^{-2} \sigma_{\min}^2(\mathcal{X})$. Let $\mathcal{U}_0 = \mathcal{U}$ where*

$$\alpha^2 \lesssim \begin{cases} \dfrac{\epsilon(R \wedge n)\|\mathcal{X}\|^2}{k^2 n^{3/2} \kappa^2} \left( \dfrac{2\kappa^2 k n^{3/2}}{c_3(R \wedge n)^{3/2}\epsilon} \right)^{-15\kappa^2} & \text{if } R \geq 3r \\[4ex] \dfrac{\epsilon\|\mathcal{X}\|^2}{k^2 n^{3/2} \kappa^2} \left( \dfrac{2\kappa^2 k n^{3/2}}{c_3 r^{1/2}\epsilon} \right)^{-15\kappa^2} & \text{if } R < 3r \end{cases}$$

*Assume the step size satisfies $\eta \leq c_2 \kappa^{-2} \|\mathcal{X}\|^{-2}$. Then, with probability at least $1 - p$ where*

$$p = \begin{cases} k(\tilde{C}\epsilon)^{R-2r+1} + ke^{-\tilde{c}R} & \text{if } R \geq 2r \\ k\epsilon^2 + ke^{-\tilde{c}R} & \text{if } R < 2r \end{cases}$$

*the following statement holds. After*

$$t_* \lesssim \begin{cases} \dfrac{1}{\eta \min_{1 \leq j \leq k} \sigma_r(\bar{X}^{(j)})^2} \ln\left( \dfrac{2\kappa^2\sqrt{n}}{c_3\epsilon\sqrt{(R \wedge n)}} \right) & \text{if } R \geq 3r \\[4ex] \dfrac{1}{\eta \min_{1 \leq j \leq k} \sigma_r(\bar{X}^{(j)})^2} \ln\left( \dfrac{2\kappa^2\sqrt{rn}}{c_3\epsilon} \right) & \text{if } R < 3r \end{cases}$$

*iterations, it holds that*

$$\|\mathcal{U}_{t_*}\| \leq 3\|\mathcal{X}\|$$

$$\|\mathcal{V}_{\mathcal{X}^\perp}^\top * \mathcal{U}_{t_*} * \mathcal{W}_*\| \leq \epsilon\kappa^{-2}$$

*and for each $1 \leq j \leq k$, we have*

$$\sigma_r\left( \overline{\mathcal{U}_{t_*} * \mathcal{W}_{t_*}}^{(j)} \right) \geq \frac{1}{4}\alpha\beta$$

$$\sigma_1\left( \overline{\mathcal{U}_{t_*} * \mathcal{W}_{t_*,\perp}}^{(j)} \right) \leq \frac{\kappa^{-2}}{8}\alpha\beta$$

*where*

$$\beta \lesssim \begin{cases} \epsilon\sqrt{k} \left( \dfrac{2\kappa^2\sqrt{n}}{c_3\epsilon\sqrt{R \wedge n}} \right)^{10\kappa^2} & \text{if } R \geq 3r \\[4ex] \dfrac{\epsilon\sqrt{k}}{r} \left( \dfrac{2\kappa^2\sqrt{rn}}{c_3\epsilon} \right)^{10\kappa^2} & \text{if } R < 3r \end{cases}$$

*and*

$$\beta \gtrsim \begin{cases} \epsilon\sqrt{k} & \text{if } R \geq 3r \\[2ex] \dfrac{\epsilon\sqrt{k}}{r} & \text{if } R < 3r. \end{cases}$$

*Here, $c_1, c_2, c_3 > 0$ are absolute constants only depending on the choice of $c$. Moreover, $\tilde{C}, \tilde{c}$ are absolute numerical constants.*

### E.1.2 PHASE II: SIGNAL AMPLIFICATION PHASE

In the second phase, building upon the results from the first phase, the tensor-column subspace of the signal term remains well-aligned with that of the ground truth $\mathcal{X}_\star$, i.e., $\|\mathcal{V}_{\mathcal{X}^\perp}^\top * \mathcal{V}_{\mathcal{U}_t * \mathcal{w}_t}\|$ remains small. Meanwhile, the magnitude of the signal term, measured by $\sigma_{\min}(\mathcal{V}_{\mathcal{X}}^\top * \mathcal{U}_t)$, grows exponentially. In contrast, the over-parameterization term $\|\mathcal{U}_t * \mathcal{W}_{t,\perp}\|$ stays small due to the small initialization.

**Lemma 2.** *Suppose that the step size satisfies $\eta \leq c_1 \kappa^{-2} \|\boldsymbol{\mathcal{X}}\|^{-2}$ for some small $c_1 > 0$, $\|\boldsymbol{\mathcal{E}}\| \leq c_1 \kappa^{-2} \sigma_{\min}^2(\boldsymbol{\mathcal{X}})$, and $\mathfrak{M} : \mathbb{R}^{n \times n \times k} \to \mathbb{R}^m$ satisfies $(2r+1, \delta)$ t-RIP for some constant $0 < \delta \leq \frac{c_1}{\kappa^4 \sqrt{kr}}$. Set $\gamma \in (0, \frac{1}{2})$, and choose a number of iterations $t_*$ such that $\sigma_{\min}(\boldsymbol{\mathcal{U}}_{t_*} * \boldsymbol{\mathcal{W}}_{t_*}) \geq \gamma$. Also, assume that $\|\boldsymbol{\mathcal{U}}_{t_*} * \boldsymbol{\mathcal{W}}_{t_*,\perp}\| \leq 2\gamma$, $\|\boldsymbol{\mathcal{U}}_{t_*}\| \leq 3\|\boldsymbol{\mathcal{X}}\|$, $\gamma \leq \frac{c_2 \sigma_{\min}(\boldsymbol{\mathcal{X}})}{k^{8/7} \kappa^2 (R \wedge n)}$, and $\|\boldsymbol{\mathcal{V}}_{\boldsymbol{\mathcal{X}}^\perp}^\top * \boldsymbol{\mathcal{V}}_{\boldsymbol{\mathcal{U}}_{t_*} * \boldsymbol{\mathcal{W}}_{t_*}}\| \leq c_2 \kappa^{-2}$ for some small $c_2 > 0$. Set*

$$t_1 = \min \left\{ t \geq t_* : \sigma_{\min}(\boldsymbol{\mathcal{V}}_{\boldsymbol{\mathcal{X}}}^\top * \boldsymbol{\mathcal{U}}_t) \geq \frac{1}{\sqrt{10}} \sigma_{\min}(\overline{\boldsymbol{\mathcal{X}}}) \right\}, \tag{4}$$

*and , Then the following hold for all $t \in [t_*, t_1]$:*

$$\sigma_{\min}(\boldsymbol{\mathcal{V}}_{\boldsymbol{\mathcal{X}}}^\top * \boldsymbol{\mathcal{U}}_t) \geq \frac{1}{2} \gamma \left( 1 + \frac{1}{8} \eta \sigma_{\min}(\boldsymbol{\mathcal{X}})^2 \right)^{t-t_*} \tag{5}$$

$$\|\boldsymbol{\mathcal{U}}_t * \boldsymbol{\mathcal{W}}_{t,\perp}\| \leq 2\gamma \left( 1 + 80 \eta c_2 \sigma_{\min}(\boldsymbol{\mathcal{X}})^2 \right)^{t-t_*} \tag{6}$$

$$\|\boldsymbol{\mathcal{U}}_t\| \leq 3\|\boldsymbol{\mathcal{X}}\| \quad \text{and} \quad \|\boldsymbol{\mathcal{V}}_{\boldsymbol{\mathcal{X}}^\perp}^\top * \boldsymbol{\mathcal{V}}_{\boldsymbol{\mathcal{U}}_t * \boldsymbol{\mathcal{W}}_t}\| \leq c_2 \kappa^{-2}, \tag{7}$$

*where $t_1 - t_* \lesssim \frac{1}{\eta \sigma_{\min}^2} \ln(\frac{\sigma_{\min}}{\gamma})$.*

### E.1.3 PHASE III: LOCAL REFINEMENT PHASE

Once the magnitude of the signal term $\sigma_{\min}(\boldsymbol{\mathcal{V}}_{\boldsymbol{\mathcal{X}}}^\top * \boldsymbol{\mathcal{U}}_t)$ exceeds $\frac{\sigma_{\min}(\boldsymbol{\mathcal{X}})}{\sqrt{10}}$, the algorithm enters the third phase. In this phase, the recovery error can be decomposed as

$$\|\boldsymbol{\mathcal{U}}_t * \boldsymbol{\mathcal{U}}_t^\top - \boldsymbol{\mathcal{X}}_\star\| \leq 4\|\boldsymbol{\mathcal{V}}_{\boldsymbol{\mathcal{X}}}^\top * (\boldsymbol{\mathcal{U}}_t * \boldsymbol{\mathcal{U}}_t^\top - \boldsymbol{\mathcal{X}}_\star)\| + \|\boldsymbol{\mathcal{U}}_t * \boldsymbol{\mathcal{W}}_{t,\perp}\|^2.$$

Due to the small initialization, the over-parameterization term $\|\boldsymbol{\mathcal{U}}_t * \boldsymbol{\mathcal{W}}_{t,\perp}\|^2$ grows slowly, while the in-subspace error $\|\boldsymbol{\mathcal{V}}_{\boldsymbol{\mathcal{X}}}^\top * (\boldsymbol{\mathcal{U}}_t * \boldsymbol{\mathcal{U}}_t^\top - \boldsymbol{\mathcal{X}}_\star)\|$ decreases rapidly. Moreover, since $\boldsymbol{\mathcal{V}}_{\boldsymbol{\mathcal{X}}} \in \mathbb{R}^{n \times r \times k}$, we have

$$\|\boldsymbol{\mathcal{V}}_{\boldsymbol{\mathcal{X}}}^\top * (\boldsymbol{\mathcal{U}}_t * \boldsymbol{\mathcal{U}}_t^\top - \boldsymbol{\mathcal{X}}_\star)\|_F \leq \sqrt{r}\|\boldsymbol{\mathcal{V}}_{\boldsymbol{\mathcal{X}}}^\top * (\boldsymbol{\mathcal{U}}_t * \boldsymbol{\mathcal{U}}_t^\top - \boldsymbol{\mathcal{X}}_\star)\|,$$

which explains why the final recovery error depends only on the true tubal-rank $r$, despite the over-parameterization.

**Lemma 3.** *Suppose that the assumptions in Lemma 2 hold. If $R > r$, then for*

$$\hat{t} \asymp \frac{1}{\eta \sigma_{\min}(\boldsymbol{\mathcal{X}})^2} \ln \left( \frac{\kappa \|\boldsymbol{\mathcal{X}}\|}{((R \wedge n) - r)\gamma k} \right) + t_1$$

*iterations it holds that*

$$\|\boldsymbol{\mathcal{U}}_{\hat{t}} * \boldsymbol{\mathcal{U}}_{\hat{t}}^\top - \boldsymbol{\mathcal{X}} * \boldsymbol{\mathcal{X}}^\top\|_F \lesssim \sqrt{r} \kappa^{-3/16}((R \wedge n) - r)^{3/4} k^{3/4} \gamma^{21/16} \|\boldsymbol{\mathcal{X}}\|^{11/16} + \sqrt{r} \kappa^2 \|\boldsymbol{\mathcal{E}}\| \cdot ; \tag{8}$$

*if $R = r$, then for any $t \geq t_1$,*

$$\|\boldsymbol{\mathcal{U}}_t * \boldsymbol{\mathcal{U}}_t^\top - \boldsymbol{\mathcal{X}} * \boldsymbol{\mathcal{X}}^\top\|_F \lesssim \sqrt{r}(1 - \frac{\eta}{400} \sigma_{\min}^2(\boldsymbol{\mathcal{X}}))^{t-t_1} + \sqrt{r} \kappa^2 \|\boldsymbol{\mathcal{E}}\|. \tag{9}$$

### E.1.4 PHASE IV: OVERFITTING PHASE

The fourth stage is a natural continuation of the third. Consider the decomposition from Phase III:

$$\|\boldsymbol{\mathcal{U}}_t * \boldsymbol{\mathcal{U}}_t^\top - \boldsymbol{\mathcal{X}}_\star\| \leq 4\|\boldsymbol{\mathcal{V}}_{\boldsymbol{\mathcal{X}}}^\top * (\boldsymbol{\mathcal{U}}_t * \boldsymbol{\mathcal{U}}_t^\top - \boldsymbol{\mathcal{X}}_\star)\| + \|\boldsymbol{\mathcal{U}}_t * \boldsymbol{\mathcal{W}}_{t,\perp}\|^2.$$

In the fourth stage, the over-parameterization term $\|\boldsymbol{\mathcal{U}}_t * \boldsymbol{\mathcal{W}}_{t,\perp}\|^2$ starts to grow, eventually dominating the recovery error until it matches that of spectral initialization.

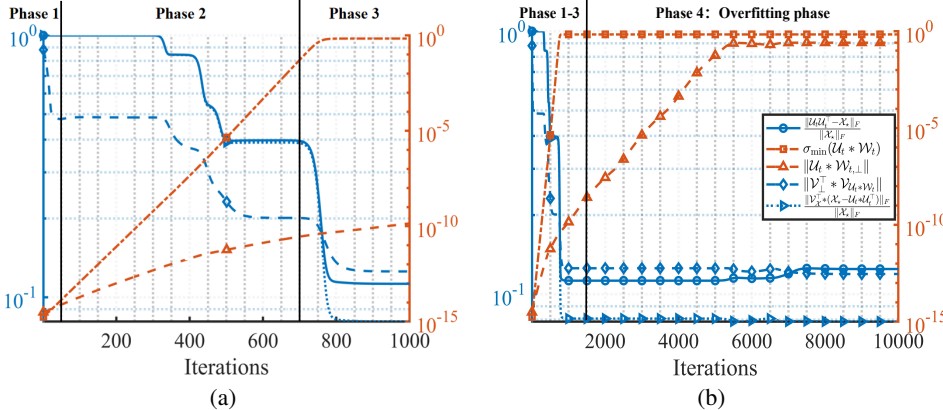

Figure 5: Validation of the four-phase convergence analysis in Section 3.3. The left panel shows the first 1,000 iterations; the right panel shows the full 10,000 iterations. The orange curve corresponds to the orange axis on the right, and the blue curve corresponds to the blue axis on the left. Parameter settings: $n = 10$, $k = 3$, $r = 2$, $R = 10$, $m = 5knR$, $\eta = 0.1$, noise standard deviation $\sigma = 0.01$, and initialization scale $\alpha = 10^{-7}$.

## E.2 VALIDATE FOUR PHASE IN SECTION 3.3

We conducted experiments to validate the four-phase convergence described in Section 3.3. As shown in Figure 5, we observed that:

- In Phase 1, the column space of the signal term $\mathcal{U}_t * \mathcal{W}_t$ gradually aligns with that of the ground truth $\mathcal{X}_\star$, as indicated by the decreasing value of $\|\mathcal{V}_{\mathcal{X}^\perp}^\top * \mathcal{V}_{\mathcal{U}_t*\mathcal{W}_t}\|$. Both $\sigma_{\min}(\mathcal{U}_t * \mathcal{W}_t)$ and $\|\mathcal{U}_t * \mathcal{W}_{t,\perp}\|$ remain small due to the small initialization.

- In Phase 2, $\sigma_{\min}(\mathcal{U}_t * \mathcal{W}_t)$ grows exponentially until it reaches at least $\frac{\sigma_{\min}(\mathcal{X})}{\sqrt{10}}$, while $\|\mathcal{U}_t * \mathcal{W}_{t,\perp}\|$ remains nearly at the scale of the initialization.

- In Phase 3, the over-parameterization term $\|\mathcal{U}_t * \mathcal{W}_{t,\perp}\|^2$ remains small, while the in-subspace error $\|\mathcal{V}_{\mathcal{X}}^\top * (\mathcal{U}_t * \mathcal{U}_t^\top - \mathcal{X}_\star)\|$ decreases rapidly, leading to the lowest recovery error.

- In Phase 4, the in-subspace error $\|\mathcal{V}_{\mathcal{X}}^\top * (\mathcal{U}_t * \mathcal{U}_t^\top - \mathcal{X}_\star)\|$ continues to decrease, but only very slightly, while the over-parameterization term $\|\mathcal{U}_t * \mathcal{W}_{t,\perp}\|$ grows rapidly and dominates the total recovery error, causing the overall error $\|\mathcal{U}_t * \mathcal{U}_t^\top - \mathcal{X}_\star\|_F$ to increase.

## E.3 PROOF OF THEOREM 2

Since the linear map $\mathfrak{M}$ satisfies $(2r + 1, \delta)$ t-RIP, then by Lemma 14, $\mathfrak{M}$ satisfies $(2r, \sqrt{2rk}\delta)$ S2S-t-RIP. Therefore,

$$
\begin{aligned}
\|\mathcal{E}_{\mathcal{X}}\| &= \|(\mathfrak{I} - \mathfrak{M}^*\mathfrak{M})(\mathcal{X} * \mathcal{X}^\top) + \mathfrak{M}^*(s)\| \\
&\leq \sqrt{2rk}\delta\|\mathcal{X} * \mathcal{X}^\top\| + \|\mathfrak{M}^*(s)\| \\
&\leq \sqrt{2rk} \cdot c\kappa^{-4}(rk)^{-1/2} \cdot \|\mathcal{X}\|^2 + \|\mathfrak{M}^*(s)\| \\
&= \sqrt{2}c\kappa^{-2}\sigma_{\min}(\mathcal{X})^2 + \|\mathfrak{M}^*(s)\| \\
&\overset{(a)}{\leq} \sqrt{2}c\kappa^{-2}\sigma_{\min}(\mathcal{X})^2 + c_1\kappa^{-2}\sigma_{\min}(\mathcal{X})^2 \\
&\lesssim c\kappa^{-2}\sigma_{\min}(\mathcal{X})^2
\end{aligned}
\tag{10}
$$

where $(a)$ use the assumption $\|\mathcal{E}\| \leq c_1\kappa^{-2}\sigma_{\min}^2(\mathcal{X})$.

Then Lemma 1 holds with probability at least $1 - ke^{-\tilde{c}R} - \max\{k(\tilde{C}\epsilon)^{R-r+1}, k\epsilon^2\}$. We then divide the proof of Theorem 2 into three cases: $R = r$, $r < R < 3r$, and $R \geq 3r$.

### E.3.1 CASE 1 : $R = r$

In this case, by the results of Lemma 1, the following statement holds: choose

$$\alpha^2 \lesssim \frac{||\boldsymbol{\mathcal{X}}||^2}{k^2 n^{3/2} \kappa^2} \left( \frac{2\kappa^2 k n^{3/2}}{\tilde{c}_3 r^{1/2}} \right)^{-15\kappa^2} \text{ and } \tilde{c}_3 = c_3 \epsilon,$$

then after

$$t_* \lesssim \frac{1}{\eta \sigma_{\min}(\boldsymbol{\mathcal{X}})^2} \ln \left( \frac{2\kappa^2 \sqrt{rn}}{\tilde{c}_3} \right)$$

iterations, it holds that

$$||\boldsymbol{\mathcal{U}}_{t_*}|| \leq 3||\boldsymbol{\mathcal{X}}|| \text{ and } ||\boldsymbol{\mathcal{V}}_{\boldsymbol{\mathcal{X}}^\perp} * \boldsymbol{\mathcal{V}}_{\boldsymbol{\mathcal{U}}_{t_*} * \boldsymbol{\mathcal{W}}_{t_*}}|| \leq c\kappa^{-2} \tag{11}$$

for each $1 \leq j \leq k$, we have

$$\sigma_r \left( \overline{\boldsymbol{\mathcal{U}}_{t_*} * \boldsymbol{\mathcal{W}}_{t_*}}^{(j)} \right) \geq \frac{1}{4} \alpha\beta$$

$$\sigma_1 \left( \overline{\boldsymbol{\mathcal{U}}_{t_*} * \boldsymbol{\mathcal{W}}_{t_*,\perp}}^{(j)} \right) \leq \frac{\kappa^{-2}}{8} \alpha\beta,$$

where

$$\frac{\sqrt{k}}{r} \lesssim \beta \lesssim \frac{\sqrt{k}}{r} \left( \frac{2\kappa^2 \sqrt{nr}}{\tilde{c}_3} \right)^{10\kappa^2}$$

and $\tilde{c}_3 = \epsilon c_3 = e^{-\tilde{c}/2} c_3$. By taking

$$\alpha \lesssim \frac{\sqrt{r} \sigma_{\min}(\boldsymbol{\mathcal{X}})}{k^{23/14}(R \wedge n)\kappa^2} \left( \frac{2\kappa^2 \sqrt{rn}}{\tilde{c}_3} \right)^{-10\kappa^2},$$

we have $\gamma = \frac{1}{4}\alpha\beta \lesssim \frac{c_2 \sigma_{\min}(\boldsymbol{\mathcal{X}})}{k^{8/7}\kappa^2(R \wedge n)}$. Also, we have

$$||\boldsymbol{\mathcal{U}}_{t_*} * \boldsymbol{\mathcal{W}}_{t_*,\perp}|| \leq \frac{\kappa^{-2}}{8} \alpha\beta \leq \frac{\gamma}{2\kappa^2} \leq 2\gamma.$$

Therefore, the assumptions of Lemmas 2 and 3 hold, then we can use the results of Lemma 3 to obtain:

$$||\boldsymbol{\mathcal{U}}_t * \boldsymbol{\mathcal{U}}_t^\top - \boldsymbol{\mathcal{X}} * \boldsymbol{\mathcal{X}}^\top||_F \lesssim \sqrt{r}(1 - \frac{\eta}{400}\sigma_{\min}^2(\boldsymbol{\mathcal{X}}))^{t-t_1} + \sqrt{r}\kappa^2||\boldsymbol{\mathcal{E}}||,$$

for all $t \geq t_1$, where

$$t_1 \lesssim t_* + (t_1 - t_*) \lesssim \frac{1}{\eta \sigma_{\min}(\boldsymbol{\mathcal{X}})^2} \ln \left( \frac{2\kappa^2 \sqrt{rn}}{\tilde{c}_3} \right) + \frac{1}{\eta \sigma_{\min}^2(\boldsymbol{\mathcal{X}})} \ln \left( \frac{\sigma_{\min}(\boldsymbol{\mathcal{X}})}{\gamma} \right) \tag{12}$$

$$\overset{(a)}{\lesssim} \frac{1}{\eta \sigma_{\min}(\boldsymbol{\mathcal{X}})^2} \ln \left( \frac{\kappa^2 \sqrt{rn} \sigma_{\min}(\boldsymbol{\mathcal{X}})}{\tilde{c}_3 \alpha\beta} \right) \tag{13}$$

$$\overset{(b)}{\lesssim} \frac{1}{\eta \sigma_{\min}(\boldsymbol{\mathcal{X}})^2} \ln \left( \frac{\kappa^2 r^{3/2} \sqrt{n} \sigma_{\min}(\boldsymbol{\mathcal{X}})}{\sqrt{k}\alpha} \right), \tag{14}$$

where $(a)$ uses the fact that $\gamma = \frac{\alpha\beta}{4}$; (b) uses the fact that $\beta \gtrsim \frac{\sqrt{k}}{r}$.

Then define $\mu := \frac{\eta}{400}\sigma_{\min}^2(\boldsymbol{\mathcal{X}}_\star) \in (0,1)$. Using the fact that $(1-\mu)^s \leq e^{-\mu s}$, we have

$$||\boldsymbol{\mathcal{U}}_t * \boldsymbol{\mathcal{U}}_t^\top - \boldsymbol{\mathcal{X}} * \boldsymbol{\mathcal{X}}^\top||_F \lesssim \sqrt{r}e^{-\mu(t-t_1)} + \sqrt{r}\kappa^2||\boldsymbol{\mathcal{E}}||.$$

### E.3.2 CASE 2 : $r < R < 3r$

The analysis for this case is almost the same way as that of the previous case, except that it relies on a different result from Lemma 3, namely that when $R > r$, we have

$$||\boldsymbol{\mathcal{U}}_{\hat{t}} * \boldsymbol{\mathcal{U}}_{\hat{t}}^\top - \boldsymbol{\mathcal{X}} * \boldsymbol{\mathcal{X}}^\top||_F \lesssim \kappa^{-3/16} r^{1/2}((R \wedge n) - r)^{3/4} k^{3/4} \gamma^{21/16} ||\boldsymbol{\mathcal{X}}||^{11/16} + \sqrt{r}\kappa^2||\boldsymbol{\mathcal{E}}||,$$

where

$$\hat{t} \asymp t_1 + \frac{1}{\eta \sigma_{\min}(\boldsymbol{\mathcal{X}})^2} \ln \left( \frac{\kappa||\boldsymbol{\mathcal{X}}||}{((R \wedge n) - r)\gamma k} \right).$$

Taking the bound in Case 1 for $t_1$, we have

$$\hat{t} \asymp (t_1 - t_*) + t_* + \frac{1}{\eta\sigma_{\min}(\boldsymbol{\mathcal{X}})^2} \ln\left(\frac{\kappa||\boldsymbol{\mathcal{X}}||}{((R \wedge n) - r)\gamma k}\right) \tag{15}$$

$$\asymp \frac{1}{\eta\sigma_{\min}^2(\boldsymbol{\mathcal{X}})} \ln\left(\frac{n^{1/2}r^{5/2}\kappa^2||\boldsymbol{\mathcal{X}}||^2}{k^2[(R \wedge n) - r]\alpha^2}\right). \tag{16}$$

To obtain the result $||\boldsymbol{\mathcal{U}}_{\hat{t}} * \boldsymbol{\mathcal{U}}_{\hat{t}}^\top - \boldsymbol{\mathcal{X}} * \boldsymbol{\mathcal{X}}^\top|| \lesssim \kappa^2||\boldsymbol{\mathcal{E}}||$, we need to ensure $\kappa^{-3/16}r((R \wedge n) - r)^{3/4}k^{3/4}\gamma^{21/16}||\boldsymbol{\mathcal{X}}||^{11/16} \leq \kappa^2||\boldsymbol{\mathcal{E}}||$, which leads to

$$\alpha \lesssim \kappa^{35/21}[(R \wedge n) - r]^{-4/7}rk^{-15/14}||\boldsymbol{\mathcal{X}}||^{-11/21}||\boldsymbol{\mathcal{E}}||^{16/21}\left(\frac{2\kappa^2\sqrt{rn}}{\tilde{c}_3}\right)^{-10\kappa^2}. \tag{17}$$

Using the facts that $\gamma = \frac{\alpha\beta}{4}$ and $\beta \lesssim \frac{\sqrt{k}}{r}\left(\frac{2\kappa^2\sqrt{rn}}{\tilde{c}_3}\right)^{10\kappa^2}$, in order to satisfy the assumption $\gamma \lesssim \frac{c_2\sigma_{\min}(\boldsymbol{\mathcal{X}})}{k^{8/7}\kappa^2(R \wedge n)}$, we also need

$$\alpha \lesssim \frac{c_2 r\sigma_{\min}(\boldsymbol{\mathcal{X}})}{\kappa^2(R \wedge n)k^{23/14}}\left(\frac{2\kappa^2\sqrt{rn}}{\tilde{c}_3}\right)^{-10\kappa^2}. \tag{18}$$

Combining the bounds (17) and (18), we obtain the bounds for $\alpha$ :

$$\alpha \lesssim \min\left\{\frac{r\sigma_{\min}(\boldsymbol{\mathcal{X}})}{k^{23/14}(R \wedge n)\kappa^2}, \frac{r\kappa^{35/21}||\boldsymbol{\mathcal{E}}||^{16/21}}{k^{15/14}[(R \wedge n) - r]^{4/7}||\boldsymbol{\mathcal{X}}||^{11/21}}\right\}\left(\frac{2\kappa^2\sqrt{rn}}{\tilde{c}_3}\right)^{-10\kappa^2} \tag{19}$$

### E.3.3 CASE 3: $R \geq 3r$

In this case, we also use the result from Lemma 3. However, according to Lemma 1, the bounds for $t_*$ and $\beta$ are different.

Specifically, we have

$$\begin{aligned} t_* &\lesssim \frac{1}{\eta\sigma_{\min}(\boldsymbol{\mathcal{X}})^2} \ln\left(\frac{2\kappa^2\sqrt{n}}{c_3\epsilon\sqrt{(R \wedge n)}}\right) \\ \epsilon\sqrt{k} &\lesssim \beta \lesssim \epsilon\sqrt{k}\left(\frac{2\kappa^2\sqrt{n}}{c_3\epsilon(R \wedge n)}\right)^{10\kappa^2}, \end{aligned} \tag{20}$$

which implies

$$\begin{aligned} \hat{t} &\asymp t_* + t_1 - t_* + \hat{t} - t_1 \\ &\asymp \frac{1}{\eta\sigma_{\min}(\boldsymbol{\mathcal{X}})^2} \ln\left(\frac{2\kappa^2\sqrt{n}}{c_3\epsilon\sqrt{(R \wedge n)}}\right) + \frac{1}{\eta\sigma_{\min}(\boldsymbol{\mathcal{X}})^2} \ln\left(\frac{\sigma_{\min}(\boldsymbol{\mathcal{X}})}{\gamma}\right) \\ &\quad + \frac{1}{\eta\sigma_{\min}(\boldsymbol{\mathcal{X}})^2} \ln\left(\frac{\kappa||\boldsymbol{\mathcal{X}}||}{((R \wedge n) - r)\gamma k}\right) \\ &\asymp \frac{1}{\eta\sigma_{\min}(\boldsymbol{\mathcal{X}})^2} \ln\left(\frac{\sqrt{n}\kappa^2||\boldsymbol{\mathcal{X}}||^2}{k^2((R \wedge n) - r)(R \wedge r)\alpha^2}\right). \end{aligned} \tag{21}$$

Using the relation $\gamma = \frac{1}{4}\alpha\beta \lesssim \frac{c_2\sigma_{\min}(\boldsymbol{\mathcal{X}})}{\kappa^2(R \wedge n)}$, we obtain

$$\alpha \lesssim \frac{\sigma_{\min}(\boldsymbol{\mathcal{X}})}{k^{23/14}(R \wedge n)\kappa^2}\left(\frac{2\kappa^2\sqrt{n}}{\tilde{c}_3\sqrt{(R \wedge n)}}\right)^{-10\kappa^2} \tag{22}$$

Moreover, according to the result of Lemma 3, in order to obtain $||\boldsymbol{\mathcal{U}}_{\hat{t}} * \boldsymbol{\mathcal{U}}_{\hat{t}}^\top - \boldsymbol{\mathcal{X}} * \boldsymbol{\mathcal{X}}^\top|| \lesssim \kappa^2||\boldsymbol{\mathcal{E}}||$, we need to bound $\alpha$ as:

$$\alpha \lesssim \kappa^{35/21}[(R \wedge n) - r]^{-4/7}k^{-4/7}\beta^{-1}||\boldsymbol{\mathcal{X}}||^{-11/21}||\boldsymbol{\mathcal{E}}||^{16/21}$$

$$\overset{(a)}{\to} \alpha \lesssim \kappa^{35/21}[(R \wedge n) - r]^{-4/7}k^{-4/7}||\boldsymbol{\mathcal{X}}||^{-11/21}\frac{1}{\epsilon\sqrt{k}}\left(\frac{2\kappa^2\sqrt{n}}{\tilde{c}_3\sqrt{(R \wedge n)}}\right)^{-10\kappa^2}, \tag{23}$$

where $(a)$ uses the upper bound for $\beta$. Combining these two bounds (22) (23), we obtain the bound for $\alpha$ :

$$\alpha \lesssim \min\left\{\frac{\sigma_{\min}(\boldsymbol{\mathcal{X}})}{k^{23/14}(R\wedge n)\kappa^2}, \frac{\kappa^{35/21}||\boldsymbol{\mathcal{E}}||^{16/21}}{k^{15/14}||\boldsymbol{\mathcal{X}}||^{11/21}[(R\wedge n)-r]^{4/7}}\right\}\left(\frac{2\kappa^2\sqrt{n}}{\tilde{c}_3\sqrt{(R\wedge n)}}\right)^{-10\kappa^2}. \quad (24)$$

Therefore, we complete the proof of Theorem 2.

### E.4 PROOF OF COROLLARY 1

The proof of Corollary 1 follows directly from Theorem 2 combined with the spectral norm bound of $||\boldsymbol{\mathcal{E}}||$. Note that

$$||\mathfrak{M}^*(s)|| \overset{(a)}{\lesssim} \sqrt{\frac{nk}{m}}\sigma \overset{(b)}{\leq} c\kappa^{-2}\sigma_{\min}^2(\boldsymbol{\mathcal{X}}), \quad (25)$$

where $(a)$ use the result in (Liu et al., 2024b); (b) use the assumption that $m \gtrsim nk\kappa^4\sigma^2/\sigma_{\min}^2(\boldsymbol{\mathcal{X}})$. Thus the assumption (3) in Theorem 2 is satisfied. Then we can directly use the results in Theorem 2 to get

$$||\boldsymbol{\mathcal{U}}_{\hat{t}} * \boldsymbol{\mathcal{U}}_{\hat{t}}^\top - \boldsymbol{\mathcal{X}}_\star||_F^2 \lesssim r\kappa^4||\boldsymbol{\mathcal{E}}||^2 \lesssim \frac{nkr\sigma^2\kappa^4}{m}. \quad (26)$$

### E.5 PROOF OF LEMMA 1

Lemma 1 is proved based on [(Karnik et al., 2025), Lemma D.8 and Lemma D.9], with the substitution of $\mathcal{M} := \mathfrak{M}^*\mathfrak{M}(\boldsymbol{\mathcal{X}})$ by $\mathfrak{M}^*\mathfrak{M}(\boldsymbol{\mathcal{X}}) + \boldsymbol{\mathcal{E}}$, where $\boldsymbol{\mathcal{E}} = \mathfrak{M}^*(s)$.

**Lemma 4.** *Suppose that the linear map $\mathfrak{M} : \mathbb{R}^{n\times n\times k} \to \mathbb{R}^m$ satisfies $(2,\delta_1)$ t-RIP and define $t^*$ as*

$$t^* = \min\left\{j \in \mathbb{N} : ||\tilde{\boldsymbol{\mathcal{U}}}_{j-1} - \boldsymbol{\mathcal{U}}_{j-1}|| \geq ||\tilde{\boldsymbol{\mathcal{U}}}_{j-1}||\right\}.$$

*Then for all $1 \leq t \leq t^*$, we have*

$$||\boldsymbol{\mathcal{E}}_t^{\boldsymbol{\mathcal{U}}}|| = ||\boldsymbol{\mathcal{U}}_t - \tilde{\boldsymbol{\mathcal{U}}}_t|| \leq 8(1+\delta_1\sqrt{k})\sqrt{(R\wedge n)}\frac{\alpha^3}{||\mathcal{M}||}||\boldsymbol{\mathcal{U}}||^3(1+\eta||\mathcal{M}||)^{3t}.$$

*Proof.* The proof of this lemma builds upon [(Karnik et al., 2025), Lemma D.1]. By incorporating the results from Lemma 14, Lemma 15 and Lemma 16, we can derive the theorem. Compared to [(Karnik et al., 2025), Lemma D.1], this lemma leverages the inequality $||\boldsymbol{\mathcal{U}}_{j-1}||_F \leq \sqrt{(R\wedge n)}||\boldsymbol{\mathcal{U}}_{j-1}||$ to reduce the dependence on the third dimension $k$, leading to a tighter upper bound on $||\boldsymbol{\mathcal{E}}_t^{\boldsymbol{\mathcal{U}}}||$. $\qquad\square$

**Lemma 5.** *Suppose that the linear map $\mathfrak{M} : \mathbb{R}^{n\times n\times k} \to \mathbb{R}^m$ satisfies $(2,\delta_1)$ t-RIP. Consider tensor $\mathcal{M} := \mathfrak{M}^*\mathfrak{M}(\boldsymbol{\mathcal{X}} * \boldsymbol{\mathcal{X}}^\top) + \boldsymbol{\mathcal{E}} \in \mathbb{R}^{n\times n\times k}$ and $\tilde{\boldsymbol{\mathcal{U}}}_t := (\boldsymbol{\mathcal{I}} + \eta\mathcal{M})^t * \boldsymbol{\mathcal{U}}_0$. Let $\overline{\mathcal{M}} \in \mathbb{C}^{nk\times nk}$ be the corresponding block diagonal matrix of the tensor $\mathcal{M}$ with the leading eigenvector $v_1 \in \mathbb{C}^{nk}$, then we have*

$$t^* \geq \left\lceil \frac{\ln\left(\frac{||\mathcal{M}||\cdot||\overline{\boldsymbol{\mathcal{U}}_0}^H v_1||_{l_2}}{8(1+\delta_1\sqrt{k})\sqrt{(R\wedge n)}\alpha^3||\boldsymbol{\mathcal{U}}||^3}\right)}{2\ln(1+\eta||\mathcal{M}||)} \right\rceil.$$

*Proof.* The proof of this lemma can be obtained by incorporating the result of Lemma 4 into the proof of [(Karnik et al., 2025), Lemma D.2]. $\qquad\square$

**Lemma 6.** *Assume that $\mathfrak{M} : \mathbb{R}^{n\times n\times k} \to \mathbb{R}^m$ satisfies the $\delta_1\sqrt{k}$-S2N-t-RIP for some $\delta_1 > 0$. Also, assume that*

$$\mathcal{M} := \mathfrak{M}^*\mathfrak{M}(\boldsymbol{\mathcal{X}} * \boldsymbol{\mathcal{X}}^\top) + \boldsymbol{\mathcal{E}} = \boldsymbol{\mathcal{X}} * \boldsymbol{\mathcal{X}}^\top + \underbrace{\mathfrak{M}^*\mathfrak{M}(\boldsymbol{\mathcal{X}} * \boldsymbol{\mathcal{X}}^\top) + \boldsymbol{\mathcal{E}} - \boldsymbol{\mathcal{X}} * \boldsymbol{\mathcal{X}}^\top}_{\varepsilon_{\boldsymbol{\mathcal{x}}}}$$

with $||\overline{E}_X^{(j)}|| \leq \delta\lambda_r(\overline{X}^{(j)}\overline{X}^{(j)^H})$ for each $1 \leq j \leq k$ and $\delta \leq c_1\kappa^2$. Denote the t-SVD of $\mathfrak{M}$ as $\mathcal{V}_{\mathcal{M}} * \mathcal{S}_{\mathcal{M}} * \mathcal{W}_{\mathcal{M}}^{\top}$, then define $\mathcal{L} := \mathcal{V}_{\mathcal{M}}(:, 1:r, :) \in \mathbb{R}^{n \times r \times k}$, and define the initialization $\mathcal{U}_0 = \alpha\mathcal{U}$ with the scale parameter such that:

$$\alpha^2 \leq \frac{c||\mathcal{X}||^2}{12\sqrt{(R \wedge n)k}\kappa^2||\mathcal{U}||^3} \left( \frac{2\kappa^2||\mathcal{U}||^3}{c_3\sigma_{\min}(\overline{\mathcal{V}_{\mathcal{L}}^{\top} * \mathcal{U}})} \right)^{-48\kappa^2} \min\left\{ \sigma_{\min}(\overline{\mathcal{V}_{\mathcal{L}}^{\top} * \mathcal{U}}), ||\overline{\mathcal{U}_0}^H v_1||_{l_2} \right\},$$

where $v_1 \in \mathbb{C}^{nk}$ is the leading eigenvector of matrix $\overline{\mathcal{M}} \in \mathbb{R}^{nk \times nk}$.

Assume that the learning rate $\eta$ satisfies $\eta \leq c_3\kappa^{-2}||\mathcal{X}||^{-2}$, then after $t_*$ iterations with

$$t_* \asymp \frac{1}{\eta \max_{1 \leq j \leq k} \sigma_r(\overline{X}^{(j)})^2} \ln\left( \frac{2\kappa^2||\mathcal{U}||}{c_3\sigma_{\min}(\overline{\mathcal{V}_{\mathcal{L}}^{\top} * \mathcal{U}})} \right)$$

the following statements hold:

$$||\mathcal{U}_{t_*}|| \leq 3||\mathcal{X}||, \quad ||\mathcal{V}_{\mathcal{X}^{\perp}} * \mathcal{V}_{\mathcal{U}_{t_*} * \mathcal{W}_{t_*}}|| \leq c\kappa^{-2}$$

and for each $1 \leq j \leq k$, we have

$$\sigma_r\left( \overline{\mathcal{U}_{t_*} * \mathcal{W}_{t_*}}^{(j)} \right) \geq \frac{1}{4}\alpha\beta$$

$$\sigma_r\left( \overline{\mathcal{U}_{t_*} * \mathcal{W}_{t_*,\perp}}^{(j)} \right) \leq \frac{\kappa^{-2}}{8}\alpha\beta \tag{27}$$

where $\beta$ satisfies $\sigma_{\min}(\overline{\mathcal{V}_{\mathcal{L}}^{\top} * \mathcal{U}}) \leq \beta \leq \sigma_{\min}(\overline{\mathcal{V}_{\mathcal{L}}^{\top} * \mathcal{U}}) \left( \frac{2\kappa^2||\mathcal{U}||^3}{c_3\sigma_{\min}(\overline{\mathcal{V}_{\mathcal{L}}^{\top} * \mathcal{U}})} \right)^{10\kappa^2}$.

*Proof.* The proof of this lemma relies on the result of [(Karnik et al., 2025), Lemma D.7]. The first condition in [(Karnik et al., 2025), Lemma D.7] is:

$$\gamma := \frac{\alpha \max_{1 \leq j \leq k} \sigma_{r+1}(\overline{Z}_t^{(j)})||\mathcal{U}|| + ||\mathcal{E}_t^{\mathcal{U}}||}{\min_{1 \leq j \leq k} \sigma_r(\overline{Z}_t^{(j)})} \cdot \frac{1}{\sigma_{\min}\left( \overline{\mathcal{V}_{\mathcal{L}}^{\top} * \mathcal{U}} \right)} \leq c_2\kappa^2.$$

By the definition of $\gamma$, it is sufficient to show that

$$\max_{1 \leq j \leq k} \sigma_{r+1}(\overline{Z}_t^{(j)})||\mathcal{U}|| \leq \frac{c_3}{2\kappa^2} \min_{1 \leq j \leq k} \sigma_r(\overline{Z}_t^{(j)})\sigma_{\min}\left( \overline{\mathcal{V}_{\mathcal{L}}^{\top} * \mathcal{U}} \right) \tag{28}$$

and

$$||\mathcal{E}_t^{\mathcal{U}}|| \leq \frac{c_3}{2\kappa^2}\alpha \min_{1 \leq j \leq k} \sigma_r(\overline{Z}_t^{(j)})\sigma_{\min}\left( \overline{\mathcal{V}_{\mathcal{L}}^{\top} * \mathcal{U}} \right). \tag{29}$$

Since for $\mathcal{Z}_t = (\mathcal{I} + \eta\mathcal{M})^t$ the transformation in the Fourier domain leads to the blocks

$$\overline{Z}_t^{(j)} = (Id + \eta\overline{M}^{(j)})^t,$$

combining the result of inequality (28) leads to

$$\frac{2\kappa^2||\mathcal{U}||}{c_3\sigma_{\min}(\overline{\mathcal{V}_{\mathcal{L}}^{\top} * \mathcal{U}})} \leq \frac{\min_{1 \leq j \leq k} \sigma_r(\overline{Z}_t^{(j)})}{\max_{1 \leq j \leq k} \sigma_{r+1}(\overline{Z}_t^{(j)})} = \left( \frac{1 + \eta \min_{1 \leq j \leq k} \sigma_r(\overline{M}^{(j)})}{1 + \eta \max_{1 \leq j \leq k} \sigma_{r+1}(\overline{M}^{(j)})} \right)^t. \tag{30}$$

Taking the logarithm on both sides of the inequality yields

$$\ln\left( \frac{2\kappa^2||\mathcal{U}||}{c_3\sigma_{\min}(\overline{\mathcal{V}_{\mathcal{L}}^{\top} * \mathcal{U}})} \right) \leq t \ln\left( \frac{1 + \eta \min_{1 \leq j \leq k} \sigma_r(\overline{M}^{(j)})}{1 + \eta \max_{1 \leq j \leq k} \sigma_{r+1}(\overline{M}^{(j)})} \right). \tag{31}$$

Therefore, if we take $t_*$ as

$$t_* := \left\lceil \ln\left(\frac{2\kappa^2||\boldsymbol{\mathcal{U}}||}{c_3\sigma_{\min}(\overline{\boldsymbol{\mathcal{V}}_{\mathcal{L}}^\top * \boldsymbol{\mathcal{U}}})}\right) \middle/ \ln\left(\frac{1 + \eta \min\limits_{1 \leq j \leq k} \sigma_r(\overline{M}^{(j)})}{1 + \eta \max\limits_{1 \leq j \leq k} \sigma_{r+1}(\overline{M}^{(j)})}\right) \right\rceil, \tag{32}$$

then condition (28) will be satisfied in each block in the Fourier domain. For notational simplicity, we define

$$\phi := \ln\left(\frac{2\kappa^2||\boldsymbol{\mathcal{U}}||}{c_3\sigma_{\min}(\overline{\boldsymbol{\mathcal{V}}_{\mathcal{L}}^\top * \boldsymbol{\mathcal{U}}})}\right). \tag{33}$$

Then we use Lemma 4 to show that the second condition, i.e., inequality (29) is satisfied. To use Lemma 4, we need to guarantee that $t_* \leq t^*$. As proved in Lemma 5, we have

$$t^* \geq \left\lceil \frac{\ln\left(\frac{||\boldsymbol{\mathcal{M}}||\cdot||\overline{\boldsymbol{\mathcal{U}}_0}^H v_1||_{l_2}}{8(1+\delta_1\sqrt{k})\sqrt{(R\wedge n)}\alpha^3||\boldsymbol{\mathcal{U}}||^3}\right)}{2\ln(1 + \eta||\boldsymbol{\mathcal{M}}||)} \right\rceil. \tag{34}$$

In order to guarantee $t_* \leq t^*$, we need to prove

$$\frac{\phi}{\ln\left(\frac{1+\eta\min_{1\leq j\leq k}\sigma_r(\overline{M}^{(j)})}{1+\eta\max 1\leq j\leq k\sigma_{r+1}(\overline{M}^{(j)})}\right)} \leq \frac{1}{2} \cdot \frac{\ln\left(\frac{||\boldsymbol{\mathcal{M}}||\cdot||\overline{\boldsymbol{\mathcal{U}}_0}^H v_1||_{l_2}}{8(1+\delta_1\sqrt{k})\sqrt{(R\wedge n)}\alpha^3||\boldsymbol{\mathcal{U}}||^3}\right)}{2\ln(1 + \eta||\boldsymbol{\mathcal{M}}||)}. \tag{35}$$

To prove this inequality, we first bound $\ln(1 + \eta||\boldsymbol{\mathcal{M}}||) / \ln\left(\frac{1+\eta\min_{1\leq j\leq k}\sigma_r(\overline{M}^{(j)})}{1+\eta\max 1\leq j\leq k\sigma_{r+1}(\overline{M}^{(j)})}\right)$. Using the fact $\frac{x}{1+x} \leq \ln(1 + x) \leq x$, we have

$$\frac{\ln(1 + \eta||\boldsymbol{\mathcal{M}}||)}{\ln\left(\frac{1+\eta\min_{1\leq j\leq k}\sigma_r(\overline{M}^{(j)})}{1+\eta\max 1\leq j\leq k\sigma_{r+1}(\overline{M}^{(j)})}\right)} \leq \frac{||\boldsymbol{\mathcal{M}}||(1 + \eta\min_{1\leq j\leq k}\sigma_r(\overline{M}^{(j)}))}{\min_{1\leq j\leq k}\sigma_r(\overline{M}^{(j)}) - \max_{1\leq j\leq k}\sigma_{r+1}(\overline{M}^{(j)})}. \tag{36}$$

Using the assumptions $\delta \leq \frac{1}{3}$ and $\eta \leq c_3\kappa^{-2}||\boldsymbol{\mathcal{X}}||^{-2}$ and the result of [(Karnik et al., 2025), Lemma D.6], we have

$$\frac{||\boldsymbol{\mathcal{M}}||(1 + \eta\min_{1\leq j\leq k}\sigma_r(\overline{M}^{(j)}))}{\min_{1\leq j\leq k}\sigma_r(\overline{M}^{(j)}) - \max_{1\leq j\leq k}\sigma_{r+1}(\overline{M}^{(j)})} \leq \frac{(1+\delta)||\boldsymbol{\mathcal{T}}||}{(1-\delta)\lambda_r(\overline{T}^{(j)})}\left(1 + c_3(1+\delta)\left(\frac{\lambda_1(\overline{X}^{(j)})}{\kappa||\boldsymbol{\mathcal{X}}||}\right)^2\right)$$

$$\leq \kappa^2\frac{1+\delta}{1-2\delta}(1 + c_3(1+\delta)\frac{1}{\kappa^2}) \overset{(a)}{\leq} 5\kappa^2, \tag{37}$$

where (a) uses the fact that $\delta \leq 1/3$ and $c_3$ is sufficiently small. Therefore, we have

$$\frac{\ln(1 + \eta||\boldsymbol{\mathcal{M}}||)}{\ln\left(\frac{1+\eta\min_{1\leq j\leq k}\sigma_r(\overline{M}^{(j)})}{1+\eta\max 1\leq j\leq k\sigma_{r+1}(\overline{M}^{(j)})}\right)} \leq 5\kappa^2. \tag{38}$$

With this upper bound, we recall inequality (35)

$$20\kappa^2 \cdot \ln\left(\frac{2\kappa^2||\boldsymbol{\mathcal{U}}||}{c_3\sigma_{\min}(\overline{\boldsymbol{\mathcal{V}}_{\mathcal{L}}^\top * \boldsymbol{\mathcal{U}}})}\right) \leq \ln\left(\frac{||\boldsymbol{\mathcal{M}}|| \cdot ||\overline{\boldsymbol{\mathcal{U}}_0}^H v_1||_{l_2}}{8(1 + \delta_1\sqrt{k})\sqrt{(R \wedge n)}\alpha^3||\boldsymbol{\mathcal{U}}||^3}\right), \tag{39}$$

which is equal to

$$\left(\frac{2\kappa^2||\boldsymbol{\mathcal{U}}||}{c_3\sigma_{\min}(\overline{\boldsymbol{\mathcal{V}}_{\mathcal{L}}^\top * \boldsymbol{\mathcal{U}}})}\right)^{20\kappa^2} \leq \frac{||\boldsymbol{\mathcal{M}}|| \cdot ||\overline{\boldsymbol{\mathcal{U}}_0}^H v_1||_{l_2}}{8(1 + \delta_1\sqrt{k})\sqrt{(R \wedge n)}\alpha^3||\boldsymbol{\mathcal{U}}||^3} \overset{(a)}{=} \frac{||\boldsymbol{\mathcal{M}}|| \cdot ||\overline{\boldsymbol{\mathcal{U}}}^H v_1||_{l_2}}{8(1 + \delta_1\sqrt{k})\sqrt{(R \wedge n)}\alpha^2||\boldsymbol{\mathcal{U}}||^3}, \tag{40}$$

where (a) uses the fact that $||\overline{\mathcal{U}}_0^H v_1||_{l_2}/\alpha = ||\overline{\mathcal{U}}^H v_1||_{l_2}$. To prove inequality (40), we choose $\alpha$ as

$$\alpha^2 \leq \left(\frac{2\kappa^2||\mathcal{U}||}{c_3\sigma_{\min}(\overline{\mathcal{V}_{\mathcal{L}}^\top * \mathcal{U}})}\right)^{-20\kappa^2} \cdot \frac{||\mathcal{M}|| \cdot ||\overline{\mathcal{U}}^H v_1||_{l_2}}{8(1+\delta_1\sqrt{k})\sqrt{(R \wedge n)}||\mathcal{U}||^3} \tag{41}$$

With the fact that $\delta \leq \frac{1}{3}$ and $||\mathcal{M}|| \geq \frac{2}{3}||\mathcal{X}||^2$, we set $\alpha$ smaller as

$$\alpha^2 \leq \left(\frac{2\kappa^2||\mathcal{U}||}{c_3\sigma_{\min}(\overline{\mathcal{V}_{\mathcal{L}}^\top * \mathcal{U}})}\right)^{-20\kappa^2} \cdot \frac{||\mathcal{X}||^2 \cdot ||\overline{\mathcal{U}}^H v_1||_{l_2}}{16\sqrt{(R \wedge n)k}||\mathcal{U}||^3}. \tag{42}$$

Thus $t_* \leq t^*$ is satisfied, then the conditions in [(Karnik et al., 2025), Lemma D.7] hold. Therefore, using the results of [(Karnik et al., 2025), Lemma D.7], we have

$$||\mathcal{E}_{t_*}^{\mathcal{U}}|| \leq 8(1+\delta_1\sqrt{k})\sqrt{(R \wedge n)}\frac{\alpha^3}{||\mathcal{M}||}||\mathcal{U}||^3(1+\eta||\mathcal{M}||)^{3t_*}$$
$$\overset{(a)}{\leq} 12\sqrt{(R \wedge n)k}\frac{\alpha^3}{||\mathcal{M}||}||\mathcal{U}||^3(1+\eta||\mathcal{M}||)^{3t_*}, \tag{43}$$

where (a) uses the fact that $\delta \leq \frac{1}{3}$ and $||\mathcal{M}|| \geq \frac{2}{3}||\mathcal{X}||^2$ from [(Karnik et al., 2025), Lemma D.6]. Thus, using that $\overline{Z}_t^{(j)} = (Id + \eta\overline{M}^{(j)})^t$, inequality (29) holds if

$$12\sqrt{(R \wedge n)}\frac{\alpha^3}{||\mathcal{M}||}||\mathcal{U}||^3(1+\eta||\mathcal{M}||)^{3t_*} \leq \frac{c_3}{2\kappa^2}\min_{1 \leq j \leq k}\sigma_r\left((Id + \eta\overline{M}^{(j)})^{t_*}\right)\sigma_{\min}\left(\overline{\mathcal{V}_{\mathcal{L}}^\top * \mathcal{U}}\right), \tag{44}$$

which is equal to

$$\alpha^2 \leq c_3\frac{\sigma_{\min}\left(\overline{\mathcal{V}_{\mathcal{L}}^\top * \mathcal{U}}\right)||\mathcal{X}||^2}{12\sqrt{(R \wedge n)k}\kappa^2||\mathcal{U}||^3} \cdot \frac{(Id + \eta\sigma_r(\overline{M}^{(j)}))^{t_*}}{(1+\eta||\mathcal{M}||)^{3t_*}}. \tag{45}$$

Note that

$$\frac{Id + \eta\sigma_r(\overline{M}^{(j)})}{(1+\eta||\mathcal{M}||)^{3t_*}} = \exp\left(t_*\ln\left(\frac{(Id + \eta\sigma_r(\overline{M}^{(j)}))}{(1+\eta||\mathcal{M}||)^3}\right)\right) \geq \exp\left(-3t_*\ln(1+\eta||\mathcal{M}||^3)\right). \tag{46}$$

Using the definition of $t_*$, i.e., $t_* = \left\lceil \phi \middle/ \ln\left(\frac{1+\eta\min_{1 \leq j \leq k}\sigma_r(\overline{M}^{(j)})}{1+\eta\max_{1 \leq j \leq k}\sigma_{r+1}(\overline{M}^{(j)})}\right)\right\rceil$ and inequality (38), we have

$$\exp\left(-3t_*\ln(1+\eta||\mathcal{M}||^3)\right) \geq \exp\left(-15\phi\kappa^2\right) = \left(\frac{2\kappa^2||\mathcal{U}||}{c_3\sigma_{\min}(\overline{\mathcal{V}_{\mathcal{L}}^\top * \mathcal{U}})}\right)^{-15\kappa^2}. \tag{47}$$

Combining inequalities (45) and (47), we choose

$$\alpha^2 \leq c_3\frac{\sigma_{\min}\left(\overline{\mathcal{V}_{\mathcal{L}}^\top * \mathcal{U}}\right)||\mathcal{X}||^2}{12\sqrt{(R \wedge n)k}\kappa^2||\mathcal{U}||^3} \cdot \left(\frac{2\kappa^2||\mathcal{U}||}{c_3\sigma_{\min}(\overline{\mathcal{V}_{\mathcal{L}}^\top * \mathcal{U}})}\right)^{-15\kappa^2}. \tag{48}$$

With this $\alpha$, inequality (29) holds, and the condition of [(Karnik et al., 2025), Lemma D.7] is satisfied, leading to

$$||\mathcal{V}_{\mathcal{V}^\perp}^\top * \mathcal{V}_{\mathcal{U}_{t_*} * \mathcal{W}_t}|| \overset{(a)}{\leq} 14(\delta + \gamma) \leq c\kappa^{-2}, \tag{49}$$

where (a) uses the assumptions that $\delta \leq c_1\kappa^{-2}$ and $\eta \leq c_3\kappa^{-2}||\mathcal{X}||^{-2}$ and then sets the constants $c_1$ and $c_3$ small enough. Moreover, for each $1 \leq j \leq k$, using the results from [(Karnik et al., 2025), Lemma D.7], we have

$$\sigma_{\min}(\overline{\mathcal{U}_t * \mathcal{W}_t}^{(j)}) \geq \frac{1}{4}\alpha\beta$$
$$\sigma_1(\overline{\mathcal{U}_t * \mathcal{W}_{t,\perp}}^{(j)}) \geq \frac{\kappa^{-2}}{8}\alpha\beta, \tag{50}$$

where $\beta := \min_{1 \le j \le k} \sigma_r(\overline{Z}_t^{(j)}) \sigma_{\min}(\overline{\mathcal{V}_{\mathcal{L}}^{\perp} * \mathcal{U}})$.

Then we prove the bounds for $\beta$ and $||\mathcal{U}_{t_*}||$.

Consider $\beta := \min_{1 \le j \le k} \sigma_r(\overline{Z}_t^{(j)}) \sigma_{\min}(\overline{\mathcal{V}_{\mathcal{L}}^{\perp} * \mathcal{U}})$. By the definition of $\overline{Z}_t^{(j)}$ and inequality (38), we have

$$
\begin{aligned}
(1 + \eta \sigma_r(\overline{M}^{(j)}))^{t_*} &\le \exp\left(t_* \ln(1 + \eta \sigma_r(\overline{M}^{(j)}))\right) \le \exp\left(t_* \ln(1 + \eta ||\mathcal{M}||)\right) \\
&\le \exp\left(2\phi \max_{1 \le j \le k} \frac{\ln(1 + \eta ||\mathcal{M}||)}{\frac{1 + \eta \sigma_r(\overline{M}^{(j)})}{1 + \eta \sigma_{r+1}(\overline{M}^{(j)})}}\right) \le \exp\left(10\phi\kappa^2\right) \\
&= \left(\frac{2\kappa^2 ||\mathcal{U}||}{c_3 \sigma_{\min}(\overline{\mathcal{V}_{\mathcal{L}}^{\top} * \mathcal{U}})}\right)^{10\kappa^2}
\end{aligned}
\tag{51}
$$

holds for all $1 \le j \le k$.

Then we have

$$
\beta \le \sigma_{\min}(\overline{\mathcal{V}_{\mathcal{L}}^{\top} * \mathcal{U}}) \left(\frac{2\kappa^2 ||\mathcal{U}||}{c_3 \sigma_{\min}(\overline{\mathcal{V}_{\mathcal{L}}^{\top} * \mathcal{U}})}\right)^{10\kappa^2}.
\tag{52}
$$

Finally, we prove that $||\mathcal{U}_{t_*}|| \le 3||\mathcal{U}||$. By the definition of $\mathcal{U}_{t_*} = \mathcal{Z}_{t_*} * \mathcal{U}_0 + \mathcal{E}_{t_*}^{\mathcal{U}}$, we have

$$
||\mathcal{U}_{t_*}|| = \alpha ||\mathcal{Z}_{t_*}|| \cdot ||\mathcal{U}|| + ||\mathcal{E}_{t_*}^{\mathcal{U}}||.
\tag{53}
$$

By inequality (29), we have

$$
||\mathcal{E}_t^{\mathcal{U}}|| \le \frac{c_3}{2\kappa^2} \alpha ||\mathcal{Z}_t|| \sigma_{\min}\left(\overline{\mathcal{V}_{\mathcal{L}}^{\top} * \mathcal{U}}\right) \le \frac{c_3}{2\kappa^2} \alpha ||\mathcal{Z}_t|| \sigma_{\min}(\overline{\mathcal{V}_{\mathcal{L}}^H}) \sigma_{\max}(\overline{\mathcal{U}}) \le \alpha ||\mathcal{Z}_t|| ||\mathcal{U}||,
\tag{54}
$$

which leads to

$$
\begin{aligned}
||\mathcal{U}_{t_*}|| &\le 2\alpha ||\mathcal{Z}_t|| ||\mathcal{U}|| \le 2\alpha(1 + \eta ||\mathcal{M}||)^{t_*} ||\mathcal{U}|| \\
&= 2\alpha \ln(t_*(1 + \eta ||\mathcal{M}||)) ||\mathcal{U}|| \overset{(a)}{\le} 2\alpha ||\mathcal{U}|| \left(\frac{2\kappa^2 ||\mathcal{U}||}{c_3 \sigma_{\min}(\overline{\mathcal{V}_{\mathcal{L}}^{\top} * \mathcal{U}})}\right)^{10\kappa^2} \\
&\overset{(b)}{\le} 2||\mathcal{U}|| c_3 \sqrt{\frac{\sigma_{\min}\left(\overline{\mathcal{V}_{\mathcal{L}}^{\top} * \mathcal{U}}\right) ||\mathcal{X}||^2}{12\sqrt{(R \wedge n)k}\kappa^2 ||\mathcal{U}||^3}} \cdot \left(\frac{2\kappa^2 ||\mathcal{U}||}{c_3 \sigma_{\min}(\overline{\mathcal{V}_{\mathcal{L}}^{\top} * \mathcal{U}})}\right)^{-15\kappa^2/2} \\
&= 2c_3 ||\mathcal{X}|| \sqrt{\frac{\sigma_{\min}\left(\overline{\mathcal{V}_{\mathcal{L}}^{\top} * \mathcal{U}}\right)}{12\sqrt{(R \wedge n)k}\kappa^2 ||\mathcal{U}||}} \cdot \left(\frac{2\kappa^2 ||\mathcal{U}||}{c_3 \sigma_{\min}(\overline{\mathcal{V}_{\mathcal{L}}^{\top} * \mathcal{U}})}\right)^{-15\kappa^2/2} \le 3||\mathcal{U}||,
\end{aligned}
\tag{55}
$$

where (a) uses inequality (51); (b) uses the inequality (48). Lemma 1 can be obtained as a direct consequence of Lemma 6 and the proof strategy used in [(Karnik et al., 2025), Lemma D.9]. $\square$

### E.6  PROOF OF LEMMA 2

Note that for $t = t_*$, these four inequalities trivially hold using the assumptions. Before prove the $t + 1$ case, we bound $\left\| (\mathfrak{M}^*\mathfrak{M} - \mathfrak{I}) \left( \mathcal{X} * \mathcal{X}^\top - \mathcal{U}_t * \mathcal{U}_t^\top \right) + \mathcal{E} \right\|$ as:

$$
\begin{aligned}
&|| (\mathfrak{M}^*\mathfrak{M} - \mathfrak{I}) \left( \mathcal{X} * \mathcal{X}^\top - \mathcal{U}_t * \mathcal{U}_t^\top \right) + \mathcal{E}|| \\
&\leq || (\mathfrak{M}^*\mathfrak{M} - \mathfrak{I}) \left( \mathcal{X} * \mathcal{X}^\top - \mathcal{U}_t * \mathcal{W}_t * \mathcal{W}_t^\top * \mathcal{U}_t^\top \right) || \\
&\quad + || (\mathfrak{M}^*\mathfrak{M} - \mathfrak{I}) (\mathcal{U}_t * \mathcal{W}_{t,\perp} * \mathcal{W}_{t,\perp}^\top * \mathcal{U}_t^\top)|| + ||\mathcal{E}|| \\
(a) \;\; &\leq \delta\sqrt{kr}||\mathcal{X} * \mathcal{X}^\top - \mathcal{U}_t * \mathcal{W}_t * \mathcal{W}_t^\top * \mathcal{U}_t^\top|| + \delta\sqrt{k}||\mathcal{U}_t * \mathcal{W}_{t,\perp} * \mathcal{W}_{t,\perp}^\top * \mathcal{U}_t^\top||_* + ||\mathcal{E}|| \\
&\leq \delta\sqrt{kr} \left( ||\mathcal{X} * \mathcal{X}^\top|| + ||\mathcal{U}_t * \mathcal{W}_t * \mathcal{W}_t^\top * \mathcal{U}_t^\top|| \right) + \delta\sqrt{k}||\mathcal{U}_t * \mathcal{W}_{t,\perp} * \mathcal{W}_{t,\perp}^\top * \mathcal{U}_t^\top||_* + ||\mathcal{E}|| \\
&= \delta\sqrt{kr} \left( ||\mathcal{X}||^2 + ||\mathcal{U}_t * \mathcal{W}_t||^2 \right) + \delta\sqrt{k}||\mathcal{U}_t * \mathcal{W}_{t,\perp} * \mathcal{W}_{t,\perp}^\top * \mathcal{U}_t^\top||_* + ||\mathcal{E}|| \\
&\leq \delta\sqrt{kr} \left( ||\mathcal{X}||^2 + ||\mathcal{U}_t||^2 \right) + \delta\sqrt{k}||\mathcal{U}_t * \mathcal{W}_{t,\perp} * \mathcal{W}_{t,\perp}^\top * \mathcal{U}_t^\top||_* + ||\mathcal{E}|| \\
(b) \;\; &\leq \delta\sqrt{kr} \left( ||\mathcal{X}||^2 + 9||\mathcal{X}||^2 \right) + \delta\sqrt{k^3}((R \wedge n) - r)||\mathcal{U}_t * \mathcal{W}_{t,\perp} * \mathcal{W}_{t,\perp}^\top * \mathcal{U}_t^\top|| + ||\mathcal{E}|| \\
&\leq 10\delta\sqrt{kr}\kappa^2\sigma_{\min}^2(\mathcal{X}) + \delta\sqrt{k^3}((R \wedge n) - r)||\mathcal{U}_t * \mathcal{W}_{t,\perp}||^2 + ||\mathcal{E}|| \\
(c) \;\; &\leq 10c_1\kappa^{-2}\sigma_{\min}^2(\mathcal{X}) + 4\delta\sqrt{k^3}((R \wedge n) - r)\gamma^2(1 + 80\eta c_2\sqrt{k}\sigma_{\min}^2(\mathcal{X}))^{2(t-t_*)} + c_1\kappa^{-2}\sigma_{\min}^2(\mathcal{X}) \\
(d) \;\; &\leq 10c_1\kappa^{-2}\sigma_{\min}^2(\mathcal{X}) + 8\delta\sqrt{k^3}((R \wedge n) - r)\gamma^{7/4}\sigma_{\min}(\mathcal{X})^{1/4} + c_1\kappa^{-2}\sigma_{\min}^2(\mathcal{X}) \\
(e) \;\; &\leq 40c_1\kappa^{-2}\sigma_{\min}^2(\mathcal{X}),
\end{aligned}
\tag{56}
$$

where (a) uses the assumptions that $\mathfrak{M}$ satisfies $(r, \delta\sqrt{kr})$ S2S-t-RIP and $\sqrt{k}\delta$-S2N-t-RIP; (b) follows from the assumption $||\mathcal{U}_t|| \leq 3||\mathcal{X}||$ and $||\mathcal{U}_t * \mathcal{W}_{t,\perp} * \mathcal{W}_{t,\perp}^\top * \mathcal{U}_t^\top||_* \leq k((R \wedge n) - r)||\mathcal{U}_t * \mathcal{W}_{t,\perp} * \mathcal{W}_{t,\perp}^\top * \mathcal{U}_t^\top||$; (c) uses the assumptions $\delta \leq \frac{c_1}{\kappa^4\sqrt{kr}}$ and the induction hypothesis; (d) uses the definition of $t_1$ and $t_*$; (e) uses the assumption $\gamma \leq \frac{c_2\sigma_{\min}(\mathcal{X})}{k^{8/7}\kappa^2(R \wedge n)}$ and chooses a sufficiently small $c_2$. With this inequality, one can replace $|| (\mathcal{A}^*\mathcal{A} - \mathcal{I}) \left( \mathcal{X} * \mathcal{X}^\top - \mathcal{U}_t * \mathcal{U}_t^\top \right) ||$ in [(Karnik et al., 2025), Lemma E.1-Lemma E.7] with $|| (\mathfrak{M}^*\mathfrak{M} - \mathfrak{I}) \left( \mathcal{X} * \mathcal{X}^\top - \mathcal{U}_t * \mathcal{U}_t^\top \right) + \mathcal{E}||$ since they have the same upper bound.

By choosing a sufficiently small $c_2$, together with other assumptions in Lemma 2, we have the assumptions in [(Karnik et al., 2025), Lemma E.6] satisfied, then we can directly use the result in [(Karnik et al., 2025), Lemma E.6] to prove $||\mathcal{U}_{t+1}|| \leq 3||\mathcal{X}||$.

Also, the assumptions in [(Karnik et al., 2025), Lemma E.1] are satisfied, then we use the result of [(Karnik et al., 2025), Lemma E.1] to prove the induction hypothesis (5):

$$
\begin{aligned}
\sigma_{\min}(\mathcal{V}_{\mathcal{X}}^\top * \mathcal{U}_{t+1}) &\geq \sigma_{\min}(\mathcal{V}_{\mathcal{X}}^\top * \mathcal{U}_{t+1} * \mathcal{W}_{t+1}) \\
&\geq \sigma_{\min}(\mathcal{V}_{\mathcal{X}}^\top * \mathcal{U}_{t+1}) \left( 1 + \frac{1}{4}\eta\sigma_{\min}(\mathcal{X})^2 - \eta\sigma_{\min}(\mathcal{V}_{\mathcal{X}}^\top * \mathcal{U}_t)^2 \right) \\
&\geq \sigma_{\min}(\mathcal{V}_{\mathcal{X}}^\top * \mathcal{U}_{t+1}) \left( 1 + \frac{1}{4}\eta\sigma_{\min}(\mathcal{X})^2 - 0.1\eta\sigma_{\min}(\mathcal{X})^2 \right) \\
&\geq \sigma_{\min}(\mathcal{V}_{\mathcal{X}}^\top * \mathcal{U}_{t+1}) \left( 1 + \frac{1}{8}\eta\sigma_{\min}(\mathcal{X})^2 \right) \\
&\geq \left( 1 + \frac{1}{8}\eta\sigma_{\min}(\mathcal{X})^2 \right) \cdot \frac{1}{2}\gamma \left( 1 + \frac{1}{8}\eta\sigma_{\min}(\mathcal{X})^2 \right)^{t-t_*} \\
&= \frac{1}{2}\gamma \left( 1 + \frac{1}{8}\eta\sigma_{\min}(\mathcal{X})^2 \right)^{(t+1)-t_*}.
\end{aligned}
\tag{57}
$$

This inequality implies that all singular values of $\mathcal{V}_{\mathcal{X}}^\top * \mathcal{U}_{t+1}$ are positive, and then together with the assumptions of Lemma 2 and equation (56), the assumptions of [(Karnik et al., 2025), Lemma E.3] are satisfied. Then we can use the result of [(Karnik et al., 2025), Lemma E.3] to prove the induction hypothesis (6):

$$
\begin{aligned}
||&\overline{\mathcal{U}_{t+1} * \mathcal{W}_{t+1,\perp}}^{(j)}|| \\
&\leq \left(1 - \frac{\eta}{2}||\overline{\mathcal{U}_t * \mathcal{W}_{t,\perp}}^{(j)}||^2 + 9\eta||\overline{\mathcal{V}_{\mathcal{X}^\perp}^\top * \mathcal{V}_{\mathcal{U}_t * \mathcal{W}_t}}^{(j)}|| \cdot ||\mathcal{X}||^2\right)||\overline{\mathcal{U}_t * \mathcal{W}_{t,\perp}}^{(j)}|| \\
&\quad + 2\eta||(\mathfrak{M}^*\mathfrak{M} - \mathfrak{I})(\mathcal{X} * \mathcal{X}^\top - \mathcal{U}_t * \mathcal{U}_t^\top) + \mathcal{E}|| \cdot ||\overline{\mathcal{U}_t * \mathcal{W}_{t,\perp}}^{(j)}|| \\
&\leq \left(1 - \frac{\eta}{2} \cdot 4\gamma^2(1 + 80\eta c_2\sigma_{\min}^2(\mathcal{X})) + 9\eta c_2\kappa^{-2}||\mathcal{X}||^2\right)||\overline{\mathcal{U}_t * \mathcal{W}_{t,\perp}}^{(j)}|| \\
&\quad + 2\eta \cdot 40c_1\kappa^{-2}\sigma_{\min}(\mathcal{X})^2||\overline{\mathcal{U}_t * \mathcal{W}_{t,\perp}}^{(j)}|| \\
&\leq \left(1 - 2\eta\gamma^2(1 + 80\eta c_2\sigma_{\min}^2(\mathcal{X})) + 9\eta c_2\sigma_{\min}(\mathcal{X})^2\right)||\overline{\mathcal{U}_t * \mathcal{W}_{t,\perp}}^{(j)}|| \\
&\quad + 80\eta \cdot c_1\kappa^{-2}\sigma_{\min}(\mathcal{X})^2||\overline{\mathcal{U}_t * \mathcal{W}_{t,\perp}}^{(j)}|| \\
&\leq (1 + 80c_2\eta\sigma_{\min}(\mathcal{X})^2)||\overline{\mathcal{U}_t * \mathcal{W}_{t,\perp}}^{(j)}|| \\
&\leq 2\gamma(1 + 80c_2\eta\sigma_{\min}(\mathcal{X})^2)^{t+1-t_*}.
\end{aligned}
\tag{58}
$$

Note that for any block diagonal matrix $A = \begin{bmatrix} A_1 & & & \\ & A_2 & & \\ & & \ddots & \\ & & & A_n \end{bmatrix}$, we have $||A|| \leq \max_i ||A_i||$.

Then we have $||\overline{\mathcal{U}_{t+1} * \mathcal{W}_{t+1,\perp}}|| \leq \max_j ||\overline{\mathcal{U}_{t+1} * \mathcal{W}_{t+1,\perp}}^{(j)}||$ since $\overline{\mathcal{U}_{t+1} * \mathcal{W}_{t+1,\perp}}$ is a block diagonal matrix. Therefore we complete the proof of induction hypothesis (6).

Then we proceed to prove $||\mathcal{V}_{\mathcal{X}}^\top * \mathcal{V}_{\mathcal{U}_{t+1} * \mathcal{W}_{t+1}}|| \leq c_2\kappa^{-2}$ via [(Karnik et al., 2025), Lemma E.5]. Note that the assumptions in [(Karnik et al., 2025), Lemma E.5] are satisfied using the assumptions of Lemma 2 and the induction hypothesis (5)-(7).

$$
\begin{aligned}
||&\mathcal{V}_{\mathcal{X}}^\top * \mathcal{V}_{\mathcal{U}_{t+1} * \mathcal{W}_{t+1}}|| \\
&\leq (1 - \frac{\eta}{4}\sigma_{\min}(\mathcal{X})^2)||\mathcal{V}_{\mathcal{X}}^\top * \mathcal{V}_{\mathcal{U}_t * \mathcal{W}_t}|| + 150\eta||(\mathfrak{M}^*\mathfrak{M} - \mathfrak{I})\left(\mathcal{X} * \mathcal{X}^\top - \mathcal{U}_t * \mathcal{U}_t^\top\right) + \mathcal{E}|| \\
&\quad + 500\eta^2||\mathcal{X} * \mathcal{X}^\top - \mathcal{U}_t * \mathcal{U}_t^\top||^2 \\
&\leq (1 - \frac{\eta}{4}\sigma_{\min}(\mathcal{X})^2)c_2\kappa^{-2} + 150\eta \cdot 40c_1\kappa^{-2}\sigma_{\min}^2(\mathcal{X}) + 500\eta^2(||\mathcal{X}||^2 + ||\mathcal{U}_t||^2)^2 \\
&\leq (1 - \frac{\eta}{4}\sigma_{\min}(\mathcal{X})^2)c_2\kappa^{-2} + 6000c_1\eta\kappa^{-2}\sigma_{\min}^2(\mathcal{X}) + 500\eta^2(||\mathcal{X}||^2 + 9||\mathcal{X}||^2)^2 \\
&= (1 - \frac{\eta}{4}\sigma_{\min}(\mathcal{X})^2)c_2\kappa^{-2} + 6000c_1\eta\kappa^{-2}\sigma_{\min}^2(\mathcal{X}) + 50000\eta^2||\mathcal{X}||^4 \\
&\leq (1 - \frac{\eta}{4}\sigma_{\min}(\mathcal{X})^2)c_2\kappa^{-2} + 6000c_1\eta\kappa^{-2}\sigma_{\min}^2(\mathcal{X}) + 50000\eta \cdot c_1\kappa^{-4}||\mathcal{X}||^{-2} \cdot ||\mathcal{X}||^4 \\
&\leq (1 - \frac{\eta}{4}\sigma_{\min}(\mathcal{X})^2)c_2\kappa^{-2} + 6000c_1\eta\kappa^{-2}\sigma_{\min}^2(\mathcal{X}) + 50000\eta \cdot c_1\kappa^{-4}||\mathcal{X}||^{-2} \cdot ||\mathcal{X}||^4 \\
&\leq (1 - \frac{\eta}{4}\sigma_{\min}(\mathcal{X})^2)c_2\kappa^{-2} + 6000c_1\eta\kappa^{-2}\sigma_{\min}^2(\mathcal{X}) + 50000\eta c_1\kappa^{-2}\sigma_{\min}(\mathcal{X})^2 \\
&\leq (1 - \frac{\eta}{4}\sigma_{\min}(\mathcal{X})^2)c_2\kappa^{-2} + 56000\eta c_1\kappa^{-2}\sigma_{\min}(\mathcal{X})^2.
\end{aligned}
\tag{59}
$$

By taking a sufficiently small $c_1$ (i.e., $c_1 \lesssim \frac{4}{56000}$), we have $||\mathcal{V}_{\mathcal{X}}^\top * \mathcal{V}_{\mathcal{U}_{t+1} * \mathcal{W}_{t+1}}|| \leq c_2\kappa^{-2}$. Therefore, we complete the induction proof.

### E.7 PROOF OF LEMMA 3

Using the definition of $t_1$ (equation (4)) and

$$\sigma_{\min}(\mathcal{V}_{\mathcal{X}}^\top * \mathcal{U}_{t_1}) \geq \frac{1}{2}\gamma\left(1 + \frac{1}{8}\eta\sigma_{\min}(\mathcal{X})^2\right)^{t_1-t_*}$$

from Lemma 2, we have

$$\frac{1}{\sqrt{10}}\sigma_{\min}(\mathcal{X}) \geq \sigma_{\min}(\mathcal{V}_{\mathcal{X}}^\top * \mathcal{U}_{t_1}) \geq \frac{1}{2}\gamma\left(1 + \frac{1}{8}\eta\sigma_{\min}(\mathcal{X})^2\right)^{t_1-t_*},$$

which leads to

$$t_1 - t_* \leq \frac{\log\left(\frac{2}{\gamma\sqrt{10}}\sigma_{\min}(\mathcal{X})\right)}{\log\left(1 + \frac{1}{8}\eta\sigma_{\min}(\mathcal{X})^2\right)} \overset{(a)}{\leq} \frac{16}{\eta\sigma_{\min}(\mathcal{X})^2}\log\left(\frac{2}{\gamma\sqrt{10}}\sigma_{\min}(\mathcal{X})\right), \tag{60}$$

where in (a), we use the fact that $\frac{1}{\log(1+x)} \leq \frac{2}{x}$ for $0 < x < 1$.

Therefore, we bound $||\mathcal{U}_{t_1} * \mathcal{W}_{t_1,\perp}||$ as

$$\begin{aligned}
||\mathcal{U}_{t_1} * \mathcal{W}_{t_1,\perp}|| &\leq 2\gamma\left(1 + 80\eta c_2\sigma_{\min}(\mathcal{X})^2\right)^{t_1-t_*} \\
&\overset{(a)}{\leq} 2\gamma\left(\frac{2}{\sqrt{10}} \cdot \frac{\sigma_{\min}(\mathcal{X})}{\gamma}\right)^{1280c_2} \\
&\overset{(b)}{\leq} 2\gamma\left(\frac{2}{\sqrt{10}} \cdot \frac{\sigma_{\min}(\mathcal{X})}{\gamma}\right)^{1/64} \\
&\overset{(c)}{\leq} 3\gamma^{63/64}\sigma_{\min}(\mathcal{X})^{1/64} \leq 3\gamma^{7/8}\sigma_{\min}(\mathcal{X})^{1/8},
\end{aligned} \tag{61}$$

where (a) follows from Equation (60); (b) uses the assumption that $c_2$ is chosen sufficiently small; (c) uses the fact that $\sigma_{\min}(\mathcal{X}) \geq \gamma$.

Then we divide the proof of Lemma 3 into two cases: the exact-rank case and the over-parameterized (over-rank) case.

**Over-rank case:** Set $\hat{t} := t_1 + \left\lceil \frac{300}{\eta\sigma_{\min}^2(\mathcal{X})} \ln\left(\frac{\kappa^{1/4}}{16k((R\wedge n)-r)} \cdot \frac{||\mathcal{X}||^{7/4}}{\gamma^{7/4}}\right)\right\rceil$. We first state our induction hypothesis for $t_1 \leq t \leq \hat{t}$:

$$\sigma_{\min}(\mathcal{U}_t * \mathcal{W}_t) \geq \sigma_{\min}(\mathcal{V}_{\mathcal{X}}^\top * \mathcal{U}_t) \geq \frac{\sigma_{\min}(\mathcal{X})}{\sqrt{10}}, \tag{62}$$

$$||\mathcal{U}_t * \mathcal{W}_{t,\perp}|| \leq (1 + 80\eta c_2\sigma_{\min}^2(\mathcal{X}))^{t-t_1}||\mathcal{U}_{t_1} * \mathcal{W}_{t_1,\perp}||, \tag{63}$$

$$||\mathcal{U}_t|| \leq 3||\mathcal{X}||, \tag{64}$$

$$||\mathcal{V}_{\mathcal{X}^\perp}^\top * \mathcal{V}_{\mathcal{U}_t*\mathcal{W}_t}|| \leq c_2\kappa^{-2}, \tag{65}$$

$$||\mathcal{V}_{\mathcal{X}}^\top * (\mathcal{X} * \mathcal{X}^\top - \mathcal{U}_t * \mathcal{U}_t^\top)|| \leq 10(1 - \frac{\eta}{400}\sigma_{\min}^2(\mathcal{X}))^{t-t_1}||\mathcal{X}||^2 \tag{66}$$

$$+ 18\eta||\mathcal{X}||^2||\mathcal{E}|| \sum_{\tau=t_1+1}^{t} (1 - \frac{\eta}{200}\sigma_{\min}^2(\mathcal{X}))^{\tau-t_1-1}. \tag{67}$$

When $t = t_1$, the inequalities (62), (64), and (65) follow from Lemma 2. As for inequality (63), it holds when $t = t_1$ obviously. When $t = t_1$, we have

$$\begin{aligned}
||\mathcal{V}_{\mathcal{X}}^\top * (\mathcal{X} * \mathcal{X}^\top - \mathcal{U}_{t_1} * \mathcal{U}_{t_1}^\top)|| &= ||\mathcal{V}_{\mathcal{X}}^\top * (\mathcal{X} * \mathcal{X}^\top - \mathcal{U}_{t_1} * \mathcal{W}_{t_1} * \mathcal{W}_{t_1}^\top * \mathcal{U}_{t_1}^\top)|| \\
&\leq ||\mathcal{X} * \mathcal{X}^\top|| + ||\mathcal{U}_{t_1} * \mathcal{W}_{t_1} * \mathcal{W}_{t_1}^\top * \mathcal{U}_{t_1}^\top|| \\
&\leq ||\mathcal{X}||^2 + ||\mathcal{U}_{t_1}||^2||\mathcal{W}_{t_1}||^2 \overset{(a)}{\leq} 10||\mathcal{X}||^2,
\end{aligned}$$

where (a) follows inequality (64). Next, we aim to prove that these inequalities also hold at step $t + 1$. To do so, we need to bound the term $\left\| (\mathfrak{M}^* \mathfrak{M} - \mathfrak{I}) \left( \mathcal{X} * \mathcal{X}^\top - \mathcal{U}_t * \mathcal{U}_t^\top \right) + \mathcal{E} \right\|$ as

$$
\begin{aligned}
&|| (\mathfrak{M}^* \mathfrak{M} - \mathfrak{I}) \left( \mathcal{X} * \mathcal{X}^\top - \mathcal{U}_t * \mathcal{U}_t^\top \right) + \mathcal{E} || \\
(a) \quad &\leq 10\delta\sqrt{rk}\kappa^2\sigma_{\min}^2(\mathcal{X}) + \delta\sqrt{k^3}((R \wedge n) - r)||\mathcal{U}_t * \mathcal{W}_{t,\perp}||^2 + ||\mathcal{E}|| \\
(b) \quad &\leq 10c_1\kappa^{-2}\sigma_{\min}^2(\mathcal{X}) + \delta\sqrt{k^3}((R \wedge n) - r)(1 + 80\eta c_2\sigma_{\min}^2(\mathcal{X}))^{2(t-t_1)}||\mathcal{U}_{t_1} * \mathcal{W}_{t_1,\perp}||^2 \\
&\quad + c_1\kappa^{-2}\sigma_{\min}^2(\mathcal{X}) \\
(c) \quad &\leq 10c_1\kappa^{-2}\sigma_{\min}^2(\mathcal{X}) + 9\delta\sqrt{k^3}((R \wedge n) - r)(1 + 80\eta c_2\sigma_{\min}^2(\mathcal{X}))^{2(\hat{t}-t_1)}\gamma^{7/4}\sigma_{\min}(\mathcal{X})^{1/4} \\
&\quad + c_1\kappa^{-2}\sigma_{\min}^2(\mathcal{X}) \\
(d) \quad &\leq 10c_1\kappa^{-2}\sigma_{\min}^2(\mathcal{X}) + 9\delta\sqrt{k^3}((R \wedge n) - r)\left( \frac{\kappa^{1/4}}{16k((R \wedge n) - r)} \cdot \frac{||\mathcal{X}||^{7/4}}{\gamma^{7/4}} \right)^{\mathcal{O}(c_2)} \\
&\quad + c_1\kappa^{-2}\sigma_{\min}^2(\mathcal{X}) \\
(e) \quad &\leq 40c_1\kappa^{-2}\sigma_{\min}^2(\mathcal{X}),
\end{aligned}
\tag{68}
$$

where (a) uses the result of equation (56); (b) uses the induction hypothesis (63) and the assumption of $||\mathcal{E}||$; (c) uses the results of (61) and induction hypothesis (63); (d) uses the definition of $\hat{t}$; (e) uses the assumption that $c_2$ are sufficiently small.

Therefore, the condition required for bound (62), (64), and (65) in Theorem E.1 (Karnik et al., 2025) is satisfied. We can thus invoke the corresponding result to conclude that inequalities (62), (64), and (65) also hold at iteration $t + 1$.

Note that we have all singular values of $\mathcal{V}_{\mathcal{X}}^\top * \mathcal{U}_{t+1} * \mathcal{W}_t$ are positive using the induction hypothesis (62), then we have the assumptions of [(Karnik et al., 2025), Lemma E.3] are satisfied. Therefore, we use the result of [(Karnik et al., 2025), Lemma E.3] to prove the induction hypothesis (63), which is exactly the way as proving inequality (6). We directly present the result without detailed proof:

$$
||\mathcal{U}_{t+1} * \mathcal{W}_{t+1}|| \leq (1 + 80c_2\eta\sigma_{\min}^2(\mathcal{X}))^{t+1-t_1}||\mathcal{U}_t * \mathcal{W}_t||. \tag{69}
$$

Then we proceed to prove inequality (67). Note that the condition (76) in Lemma 7 is satisfied since inequality (68). Moreover, the other conditions of Lemma 7 are satisfied using the induction hypothesis (62), (64), and (65). Therefore, we have

$$
\begin{aligned}
&||\mathcal{V}_{\mathcal{X}}^\top * (\mathcal{X} * \mathcal{X}^\top - \mathcal{U}_{t+1} * \mathcal{U}_{t+1}^\top)|| \\
&\stackrel{(a)}{\leq} \left( 1 - \frac{\eta}{200}\sigma_{\min}(\mathcal{X})^2 \right) ||\mathcal{V}_{\mathcal{X}}^\top * (\mathcal{X} * \mathcal{X}^\top - \mathcal{U}_t * \mathcal{U}_t^\top)|| \\
&\quad + \frac{\eta}{100}\sigma_{\min}(\mathcal{X})^2||\mathcal{U}_t * \mathcal{W}_{t,\perp} * \mathcal{W}_{t,\perp}^\top * \mathcal{U}_t^\top|| + 18\eta||\mathcal{X}||^2||\mathcal{E}|| \\
&\stackrel{(b)}{\leq} 10 \left( 1 - \frac{\eta}{200}\sigma_{\min}(\mathcal{X})^2 \right) (1 - \frac{\eta}{400}\sigma_{\min}^2(\mathcal{X}))^{t-t_1}||\mathcal{X}||^2 \\
&\quad + \frac{\eta}{100}\sigma_{\min}(\mathcal{X})^2||\mathcal{U}_t * \mathcal{W}_{t,\perp} * \mathcal{W}_{t,\perp}^\top * \mathcal{U}_t^\top|| \\
&\quad + 18\eta||\mathcal{X}||^2||\mathcal{E}|| \sum_{\tau=t_1+1}^{t+1} (1 - \frac{\eta}{200}\sigma_{\min}^2(\mathcal{X}))^{\tau-t_1-1},
\end{aligned}
\tag{70}
$$

where step (a) follows the result of Lemma 7; step (b) uses the induction hypothesis (67). Note that inequality (67) holds for $t + 1$ if

$$
||\mathcal{U}_t * \mathcal{W}_{t,\perp} * \mathcal{W}_{t,\perp}^\top * \mathcal{U}_t||_* \leq \frac{1}{4}(1 - \frac{\eta}{400}\sigma_{\min}^2(\mathcal{X}))^{t-t_1}||\mathcal{X}||^2. \tag{71}
$$

Using the relationship between operator norm and tubal tensor nuclear norm, we have

$$
\begin{aligned}
||\mathcal{U}_t * \mathcal{W}_{t,\perp} * \mathcal{W}_{t,\perp}^\top * \mathcal{U}_t||_* &\leq ((R \wedge n) - r)||\mathcal{U}_t * \mathcal{W}_{t,\perp}||^2 \\
&\overset{(a)}{\leq} k((R \wedge n) - r)(1 + 80\eta c_2 \sigma_{\min}^2(\mathcal{X}))^{2(t-t_1)}||\mathcal{U}_{t_1} * \mathcal{W}_{t_1,\perp}||^2 \\
&\overset{(b)}{\leq} 9k((R \wedge n) - r)(1 + 80\eta c_2 \sigma_{\min}^2(\mathcal{X}))^{2(t-t_1)}\sigma_{\min}(\mathcal{X})^{1/4}\gamma^{7/4}
\end{aligned}
\tag{72}
$$

where (a) uses the induction hypothesis (63); (b) uses inequality (61).

Then we need to bound term $||\mathcal{U}_{t_1} * \mathcal{W}_{t_1,\perp}||$.

Combining Equations (72) and (61), we note that the inequality (71) holds if $c_2$ is sufficiently small and

$$
9k((R \wedge n) - r)\gamma^{7/4}\sigma_{\min}(\mathcal{X})^{1/4} \leq \left(1 - \frac{\eta}{350}\sigma_{\min}(\mathcal{X})^2\right)^{t-t_1}||\mathcal{X}||^2
$$

This inequality holds so long as $t \leq \hat{t} = t_1 + \left\lceil \frac{300}{\eta\sigma_{\min}^2(\mathcal{X})} \ln\left(\frac{\kappa^{1/4}}{9k((R \wedge n)-r)}\frac{||\mathcal{X}||^{7/4}}{\gamma^{7/4}}\right)\right\rceil$ by using the fact that $\ln(1+x) \geq \frac{x}{1-x}$. Therefore, we complete the induction of over-rank case.

Then we proceed to prove the upper bound for $||\mathcal{X} * \mathcal{X}^\top - \mathcal{U}_t * \mathcal{U}_t^\top||_F$:

$$
\begin{aligned}
||\mathcal{X} * \mathcal{X}^\top - \mathcal{U}_t * \mathcal{U}_t^\top||_F &\overset{(a)}{\leq} 4||\mathcal{V}_{\mathcal{X}}^\top * (\mathcal{X} * \mathcal{X}^\top - \mathcal{U}_{\hat{t}} * \mathcal{U}_{\hat{t}}^\top)||_F + ||\mathcal{U}_{\hat{t}} * \mathcal{W}_{\hat{t},\perp} * \mathcal{W}_{\hat{t},\perp}^\top * \mathcal{U}_{\hat{t}}^\top||_* \\
&\overset{(b)}{\lesssim} \sqrt{r}(1 - \frac{\eta}{400}\sigma_{\min}^2(\mathcal{X}))^{\hat{t}-t_1}||\mathcal{X}||^2 \\
&\quad + \sqrt{r}\eta||\mathcal{X}||^2||\mathcal{E}|| \sum_{\tau=t_1+1}^{\hat{t}} (1 - \frac{\eta}{200}\sigma_{\min}^2(\mathcal{X}))^{\tau-t_1-1} \\
&\overset{(c)}{\lesssim} \sqrt{r}\left(\frac{\kappa^{1/4}}{9k((R \wedge n) - r)}\frac{||\mathcal{X}||^{7/4}}{\gamma^{7/4}}\right)^{-3/4}||\mathcal{X}||^2 + \sqrt{r}\kappa^2||\mathcal{E}|| \\
&\lesssim \kappa^{-3/16}r^{1/2}k^{3/4}((R \wedge n) - r)^{3/4}\gamma^{21/16}||\mathcal{X}||^{11/16} + + \sqrt{r}\kappa^2||\mathcal{E}||,
\end{aligned}
\tag{73}
$$

where (a) uses the result of Lemma 8; (b) follows from inequalities (67) and (71); (c) uses the definition of $\hat{t}$.

**Exact rank case:** As $R = r$, we have $\mathcal{U}_t = \mathcal{U}_t * \mathcal{W}_t * \mathcal{W}_t^\top$ and $\mathcal{W}_{t,\perp} = 0$. Using a similar approach as in the over-parameterized case, we can show that the induction hypotheses (62)-(65) hold for all $t \geq t_1$. For induction hypothesis (67), note that

$$
\mathcal{U}_t \mathcal{W}_{t,\perp} \mathcal{W}_{t,\perp}^\top \mathcal{W}_t^\top = 0,
$$

which implies that (67) also holds for all $t \geq t_1$. Therefore, we conclude that:

$$
\begin{aligned}
||\mathcal{U}_t * \mathcal{U}_t^\top - \mathcal{X} * \mathcal{X}^\top||_F &\lesssim \sqrt{r}(1 - \frac{\eta}{400}\sigma_{\min}^2(\mathcal{X}))^{t-t_1} \\
&\quad + \sqrt{r}\eta||\mathcal{X}||^2||\mathcal{E}|| \sum_{\tau=t_1+1}^{t+1} (1 - \frac{\eta}{200}\sigma_{\min}^2(\mathcal{X}))^{\tau-t_1-1} \\
&\lesssim \sqrt{r}(1 - \frac{\eta}{400}\sigma_{\min}^2(\mathcal{X}))^{t-t_1} + \sqrt{r}\kappa^2||\mathcal{E}||.
\end{aligned}
\tag{74}
$$

### E.8 PROOF OF LEMMA 4

**Lemma 7.** *Assume that the following assumptions hold:*

$$
\begin{aligned}
||\mathcal{U}_t|| &\leq 3||\mathcal{X}|| \\
\eta &\leq c\kappa^{-2}||\mathcal{X}||^{-2} \\
\sigma_{\min}(\mathcal{U}_t * \mathcal{W}_t) &\geq \frac{1}{\sqrt{10}}\sigma_{\min}(\mathcal{X}) \\
||\mathcal{V}_{\mathcal{X}^\perp}^\top * \mathcal{V}_{\mathcal{U}*\mathcal{W}_t}|| &\leq c\kappa^{-2}
\end{aligned}
\tag{75}
$$

*and*

$$||(\mathfrak{I} - \mathfrak{M} * \mathfrak{M})(\boldsymbol{\mathcal{X}} * \boldsymbol{\mathcal{X}}^\top - \boldsymbol{\mathcal{U}}_t * \boldsymbol{\mathcal{U}}_t)||$$
$$\leq c\kappa^{-2} \left( ||\boldsymbol{\mathcal{X}} * \boldsymbol{\mathcal{X}} - \boldsymbol{\mathcal{U}}_t * \boldsymbol{\mathcal{W}}_t * \boldsymbol{\mathcal{W}}_t^\top * \boldsymbol{\mathcal{U}}_t^\top|| + ||\boldsymbol{\mathcal{U}}_t * \boldsymbol{\mathcal{W}}_{t,\perp} * \boldsymbol{\mathcal{W}}_{t,\perp}^\top * \boldsymbol{\mathcal{U}}_t||_* \right), \tag{76}$$

*where the constant $c > 0$ is chosen small enough. Then it holds that*

$$||\boldsymbol{\mathcal{V}}_{\boldsymbol{\mathcal{X}}}^\top * (\boldsymbol{\mathcal{X}} * \boldsymbol{\mathcal{X}}^\top - \boldsymbol{\mathcal{U}}_{t+1} * \boldsymbol{\mathcal{U}}_{t+1}^\top)|| \leq \left( 1 - \frac{\eta}{200} \sigma_{\min}(\boldsymbol{\mathcal{X}})^2 \right) ||\boldsymbol{\mathcal{V}}_{\boldsymbol{\mathcal{X}}}^\top * (\boldsymbol{\mathcal{X}} * \boldsymbol{\mathcal{X}}^\top - \boldsymbol{\mathcal{U}}_t * \boldsymbol{\mathcal{U}}_t^\top)||$$
$$+ \frac{\eta}{100} \sigma_{\min}(\boldsymbol{\mathcal{X}})^2 ||\boldsymbol{\mathcal{U}}_t * \boldsymbol{\mathcal{W}}_{t,\perp} * \boldsymbol{\mathcal{W}}_{t,\perp}^\top * \boldsymbol{\mathcal{U}}_t^\top|| + 18\eta ||\boldsymbol{\mathcal{X}}||^2 ||\boldsymbol{\mathcal{E}}||. \tag{77}$$

*Proof of Lemma 7.* In order to establish Lemma 4, we begin by introducing a key auxiliary lemma and providing its proof.

**Lemma 8.** *Under the assumptions of Lemma 4, the following inequalities hold:*

$$\left\| \boldsymbol{\mathcal{V}}_{\boldsymbol{\mathcal{X}}^\perp}^\top * \boldsymbol{\mathcal{U}}_t * \boldsymbol{\mathcal{U}}_t^\top \right\| \leq 3 \left\| \boldsymbol{\mathcal{V}}_{\boldsymbol{\mathcal{X}}^\perp}^\top * (\boldsymbol{\mathcal{X}} * \boldsymbol{\mathcal{X}}^\top - \boldsymbol{\mathcal{U}}_t * \boldsymbol{\mathcal{U}}_t^\top) \right\| + \left\| \boldsymbol{\mathcal{U}}_t * \boldsymbol{\mathcal{W}}_{t,\perp} * \boldsymbol{\mathcal{W}}_{t,\perp}^\top * \boldsymbol{\mathcal{U}}_t^\top \right\|$$
$$\left\| \boldsymbol{\mathcal{X}} * \boldsymbol{\mathcal{X}}^\top - \boldsymbol{\mathcal{U}}_t * \boldsymbol{\mathcal{U}}_t^\top \right\| \leq 4 \left\| \boldsymbol{\mathcal{V}}_{\boldsymbol{\mathcal{X}}^\perp}^\top * (\boldsymbol{\mathcal{X}} * \boldsymbol{\mathcal{X}}^\top - \boldsymbol{\mathcal{U}}_t * \boldsymbol{\mathcal{U}}_t^\top) \right\| + \left\| \boldsymbol{\mathcal{U}}_t * \boldsymbol{\mathcal{W}}_{t,\perp} * \boldsymbol{\mathcal{W}}_{t,\perp}^\top * \boldsymbol{\mathcal{U}}_t^\top \right\|$$
$$\left\| \boldsymbol{\mathcal{X}} * \boldsymbol{\mathcal{X}}^\top - \boldsymbol{\mathcal{U}}_t * \boldsymbol{\mathcal{W}}_t * \boldsymbol{\mathcal{W}}_t^\top * \boldsymbol{\mathcal{U}}_t^\top \right\| \leq 4 \left\| \boldsymbol{\mathcal{V}}_{\boldsymbol{\mathcal{X}}^\perp}^\top * (\boldsymbol{\mathcal{X}} * \boldsymbol{\mathcal{X}}^\top - \boldsymbol{\mathcal{U}}_t * \boldsymbol{\mathcal{U}}_t^\top) \right\|, \tag{78}$$

*where $\|\cdot\|$ denotes tensor norms, such as spectral norm.*

*Proof.* The first two inequalities are derived based on Lemma E.7 in (Karnik et al., 2025). By leveraging the equivalence between matrix norms, we obtain the desired results by replacing the Frobenius norm in Lemma E.7 with the spectral norm.

Next, we present the proof of the third inequality. We decompose $\boldsymbol{\mathcal{X}} * \boldsymbol{\mathcal{X}}^\top - \boldsymbol{\mathcal{U}}_t * \boldsymbol{\mathcal{W}}_t * \boldsymbol{\mathcal{W}}_t^\top * \boldsymbol{\mathcal{U}}_t^\top$ as

$$\underbrace{\boldsymbol{\mathcal{V}}_{\boldsymbol{\mathcal{X}}} * \boldsymbol{\mathcal{V}}_{\boldsymbol{\mathcal{X}}}^\top * (\boldsymbol{\mathcal{X}} * \boldsymbol{\mathcal{X}}^\top - \boldsymbol{\mathcal{U}}_t * \boldsymbol{\mathcal{W}}_t * \boldsymbol{\mathcal{W}}_t^\top * \boldsymbol{\mathcal{U}}_t^\top)}_{\boldsymbol{\mathcal{Z}}_1} + \underbrace{\boldsymbol{\mathcal{V}}_{\boldsymbol{\mathcal{X}}^\perp} * \boldsymbol{\mathcal{V}}_{\boldsymbol{\mathcal{X}}^\perp}^\top * (\boldsymbol{\mathcal{X}} * \boldsymbol{\mathcal{X}}^\top - \boldsymbol{\mathcal{U}}_t * \boldsymbol{\mathcal{W}}_t * \boldsymbol{\mathcal{W}}_t^\top * \boldsymbol{\mathcal{U}}_t^\top)}_{\boldsymbol{\mathcal{Z}}_2}.$$

For $\boldsymbol{\mathcal{Z}}_1$, we have

$$\boldsymbol{\mathcal{Z}}_1 = \boldsymbol{\mathcal{V}}_{\boldsymbol{\mathcal{X}}} * \boldsymbol{\mathcal{V}}_{\boldsymbol{\mathcal{X}}}^\top * \boldsymbol{\mathcal{X}} * \boldsymbol{\mathcal{X}}^\top - \boldsymbol{\mathcal{V}}_{\boldsymbol{\mathcal{X}}} * \boldsymbol{\mathcal{V}}_{\boldsymbol{\mathcal{X}}}^\top * \boldsymbol{\mathcal{U}}_t * \boldsymbol{\mathcal{W}}_t * \boldsymbol{\mathcal{W}}_t^\top * \boldsymbol{\mathcal{U}}_t^\top$$
$$\overset{(1)}{=} \boldsymbol{\mathcal{V}}_{\boldsymbol{\mathcal{X}}} * \boldsymbol{\mathcal{V}}_{\boldsymbol{\mathcal{X}}}^\top * \boldsymbol{\mathcal{X}} * \boldsymbol{\mathcal{X}}^\top - \boldsymbol{\mathcal{V}}_{\boldsymbol{\mathcal{X}}} * \boldsymbol{\mathcal{V}}_{\boldsymbol{\mathcal{X}}}^\top * \boldsymbol{\mathcal{U}}_t * \boldsymbol{\mathcal{U}}_t^\top, \tag{79}$$

where $(1)$ uses the fact that

$$\boldsymbol{\mathcal{V}}_{\boldsymbol{\mathcal{X}}} * \boldsymbol{\mathcal{V}}_{\boldsymbol{\mathcal{X}}}^\top * \boldsymbol{\mathcal{U}}_t * \boldsymbol{\mathcal{U}}_t^\top$$
$$= \boldsymbol{\mathcal{V}}_{\boldsymbol{\mathcal{X}}} * \boldsymbol{\mathcal{V}}_{\boldsymbol{\mathcal{X}}}^\top * [\boldsymbol{\mathcal{U}}_t * \boldsymbol{\mathcal{W}}_t * \boldsymbol{\mathcal{W}}_t^\top + \boldsymbol{\mathcal{W}}_{t,\perp} * \boldsymbol{\mathcal{W}}_{t,\perp}^\top] * [\boldsymbol{\mathcal{U}}_t * \boldsymbol{\mathcal{W}}_t * \boldsymbol{\mathcal{W}}_t^\top + \boldsymbol{\mathcal{U}}_t * \boldsymbol{\mathcal{W}}_{t,\perp} * \boldsymbol{\mathcal{W}}_{t,\perp}^\top]^\top$$
$$= \boldsymbol{\mathcal{V}}_{\boldsymbol{\mathcal{X}}} * \boldsymbol{\mathcal{V}}_{\boldsymbol{\mathcal{X}}}^\top * \boldsymbol{\mathcal{U}}_t * \boldsymbol{\mathcal{W}}_t * \boldsymbol{\mathcal{W}}_t^\top * \boldsymbol{\mathcal{U}}_t^\top + \boldsymbol{\mathcal{V}}_{\boldsymbol{\mathcal{X}}} * \boldsymbol{\mathcal{V}}_{\boldsymbol{\mathcal{X}}}^\top * \boldsymbol{\mathcal{U}}_t * \boldsymbol{\mathcal{W}}_{t,\perp} * \boldsymbol{\mathcal{W}}_{t,\perp}^\top * \boldsymbol{\mathcal{U}}_t^\top$$
$$= \boldsymbol{\mathcal{V}}_{\boldsymbol{\mathcal{X}}} * \boldsymbol{\mathcal{V}}_{\boldsymbol{\mathcal{X}}}^\top * \boldsymbol{\mathcal{U}}_t * \boldsymbol{\mathcal{W}}_t * \boldsymbol{\mathcal{W}}_t^\top * \boldsymbol{\mathcal{U}}_t^\top + \boldsymbol{\mathcal{V}}_{\boldsymbol{\mathcal{X}}} * \boldsymbol{\mathcal{V}}_t * \boldsymbol{\mathcal{S}}_t * \boldsymbol{\mathcal{W}}_t^\top * \boldsymbol{\mathcal{W}}_{t,\perp} * \boldsymbol{\mathcal{W}}_{t,\perp}^\top * \boldsymbol{\mathcal{U}}_t^\top$$
$$= \boldsymbol{\mathcal{V}}_{\boldsymbol{\mathcal{X}}} * \boldsymbol{\mathcal{V}}_{\boldsymbol{\mathcal{X}}}^\top * \boldsymbol{\mathcal{U}}_t * \boldsymbol{\mathcal{W}}_t * \boldsymbol{\mathcal{W}}_t^\top * \boldsymbol{\mathcal{U}}_t^\top. \tag{80}$$

Therefore, we have

$$||\boldsymbol{\mathcal{Z}}_1|| = \left\| \boldsymbol{\mathcal{V}}_{\boldsymbol{\mathcal{X}}} * \boldsymbol{\mathcal{V}}_{\boldsymbol{\mathcal{X}}}^\top * \boldsymbol{\mathcal{X}} * \boldsymbol{\mathcal{X}}^\top - \boldsymbol{\mathcal{V}}_{\boldsymbol{\mathcal{X}}} * \boldsymbol{\mathcal{V}}_{\boldsymbol{\mathcal{X}}}^\top * \boldsymbol{\mathcal{U}}_t * \boldsymbol{\mathcal{U}}_t^\top \right\| \leq \left\| \boldsymbol{\mathcal{V}}_{\boldsymbol{\mathcal{X}}}^\top * (\boldsymbol{\mathcal{X}} * \boldsymbol{\mathcal{X}}^\top - \boldsymbol{\mathcal{U}}_t * \boldsymbol{\mathcal{U}}_t^\top) \right\|.$$

Then we proceed to bound the term $\mathcal{Z}_2$,

$$
\begin{aligned}
\|\mathcal{Z}_2\| &= \left\| \mathcal{V}_{\mathcal{X}^\perp} * \mathcal{V}_{\mathcal{X}^\perp}^\top * (\mathcal{X} * \mathcal{X}^\top - \mathcal{U}_t * \mathcal{W}_t * \mathcal{W}_t^\top * \mathcal{U}_t^\top) * (\mathcal{V}_{\mathcal{X}} * \mathcal{V}_{\mathcal{X}}^\top + \mathcal{V}_{\mathcal{X}^\perp} * \mathcal{V}_{\mathcal{X}^\perp}^\top) \right\| \\
&\leq \left\| \mathcal{V}_{\mathcal{X}^\perp} * \mathcal{V}_{\mathcal{X}^\perp}^\top * (\mathcal{X} * \mathcal{X}^\top - \mathcal{U}_t * \mathcal{W}_t * \mathcal{W}_t^\top * \mathcal{U}_t^\top) * \mathcal{V}_{\mathcal{X}} \right\| \\
&\quad + \left\| \mathcal{V}_{\mathcal{X}^\perp} * \mathcal{V}_{\mathcal{X}^\perp}^\top * (\mathcal{X} * \mathcal{X}^\top - \mathcal{U}_t * \mathcal{W}_t * \mathcal{W}_t^\top * \mathcal{U}_t^\top) * \mathcal{V}_{\mathcal{X}^\perp} * \mathcal{V}_{\mathcal{X}^\perp}^\top \right\| \\
&\overset{(a)}{\leq} \left\| (\mathcal{X} * \mathcal{X}^\top - \mathcal{U}_t * \mathcal{U}_t^\top) * \mathcal{V}_{\mathcal{X}} \right\| + \underbrace{\left\| \mathcal{V}_{\mathcal{X}^\perp}^\top * \mathcal{U}_t * \mathcal{W}_t * \mathcal{W}_t^\top * \mathcal{U}_t^\top * \mathcal{V}_{\mathcal{X}^\perp} \right\|}_{\mathcal{Z}_3},
\end{aligned}
$$

$$(81)$$

where (a) using the facts that $\mathcal{V}_{\mathcal{X}}^\top * \mathcal{U}_t * \mathcal{W}_{t,\perp} = 0$ and $\mathcal{V}_{\mathcal{X}}^\top * \mathcal{U}_t * \mathcal{U}_t^\top = \mathcal{V}_{\mathcal{X}}^\top * \mathcal{U}_t * \mathcal{W}_t * \mathcal{W}_t^\top * \mathcal{U}_t^\top$. For term $\mathcal{Z}_3$, we have

$$
\begin{aligned}
&\left\| \mathcal{V}_{\mathcal{X}^\perp}^\top * \mathcal{U}_t * \mathcal{W}_t * \mathcal{W}_t^\top * \mathcal{U}_t^\top * \mathcal{V}_{\mathcal{X}^\perp} \right\| \\
&= \left\| \mathcal{V}_{\mathcal{X}^\perp}^\top * \mathcal{V}_{\mathcal{U}_t*\mathcal{W}_t} * \mathcal{V}_{\mathcal{U}_t*\mathcal{W}_t}^\top * \mathcal{U}_t * \mathcal{W}_t * \mathcal{W}_t^\top * \mathcal{U}_t^\top * \mathcal{V}_{\mathcal{X}^\perp} \right\| \\
&= \left\| \mathcal{V}_{\mathcal{X}^\perp}^\top * \mathcal{V}_{\mathcal{U}_t*\mathcal{W}_t} * \left( \mathcal{V}_{\mathcal{X}}^\top * \mathcal{V}_{\mathcal{U}_t*\mathcal{W}_t} \right)^{-1} * \mathcal{V}_{\mathcal{X}}^\top * \mathcal{V}_{\mathcal{U}_t*\mathcal{W}_t} * \mathcal{V}_{\mathcal{U}_t*\mathcal{W}_t}^\top * \mathcal{U}_t * \mathcal{W}_t * \mathcal{W}_t^\top * \mathcal{U}_t^\top * \mathcal{V}_{\mathcal{X}^\perp} \right\| \\
&\leq \|\mathcal{V}_{\mathcal{X}^\perp}^\top * \mathcal{V}_{\mathcal{U}_t*\mathcal{W}_t}\| \cdot \|(\mathcal{V}_{\mathcal{X}}^\top * \mathcal{V}_{\mathcal{U}_t*\mathcal{W}_t})^{-1}\| \left\| \mathcal{V}_{\mathcal{X}}^\top * \mathcal{V}_{\mathcal{U}_t*\mathcal{W}_t} * \mathcal{V}_{\mathcal{U}_t*\mathcal{W}_t}^\top * \mathcal{U}_t * \mathcal{W}_t * \mathcal{W}_t^\top * \mathcal{U}_t^\top * \mathcal{V}_{\mathcal{X}^\perp} \right\| \\
&= \frac{\|\mathcal{V}_{\mathcal{X}^\perp}^\top * \mathcal{V}_{\mathcal{U}_t*\mathcal{W}_t}\|}{\sigma_{\min}(\mathcal{V}_{\mathcal{X}}^\top * \mathcal{V}_{\mathcal{U}_t*\mathcal{W}_t})} \left\| \mathcal{V}_{\mathcal{X}}^\top * \mathcal{U}_t * \mathcal{U}_t^\top * \mathcal{V}_{\mathcal{X}^\perp} \right\| \\
&= \frac{\|\mathcal{V}_{\mathcal{X}^\perp}^\top * \mathcal{V}_{\mathcal{U}_t*\mathcal{W}_t}\|}{\sigma_{\min}(\mathcal{V}_{\mathcal{X}}^\top * \mathcal{V}_{\mathcal{U}_t*\mathcal{W}_t})} \left\| \mathcal{V}_{\mathcal{X}}^\top * \left( \mathcal{X} * \mathcal{X}^\top - \mathcal{U}_t * \mathcal{U}_t^\top \right) * \mathcal{V}_{\mathcal{X}^\perp} \right\| \\
&\leq 2 \left\| \mathcal{V}_{\mathcal{X}}^\top * \left( \mathcal{X} * \mathcal{X}^\top - \mathcal{U}_t * \mathcal{U}_t^\top \right) \right\|.
\end{aligned}
$$

$$(82)$$

Therefore, we have the third inequality holds. $\qquad\square$

Based on the results of Lemma 8, we proceed to prove Lemma 7. We decompose $\mathcal{X} * \mathcal{X}^\top - \mathcal{U}_{t+1} * \mathcal{U}_{t+1}^\top$ into five terms by using the update formulation

$$
\mathcal{U}_{t+1} = \mathcal{U}_t + \eta[(\mathfrak{M}^*\mathfrak{M})(\mathcal{X} * \mathcal{X}^\top - \mathcal{U}_t * \mathcal{U}_t^\top) + \mathcal{E}] * \mathcal{U}_t :
$$

$$
\begin{aligned}
\mathcal{X} &* \mathcal{X}^\top - \mathcal{U}_{t+1} * \mathcal{U}_{t+1}^\top \\
&= \underbrace{(\mathcal{I} - \eta\mathcal{U}_t * \mathcal{U}_t^\top) * (\mathcal{X} * \mathcal{X}^\top - \mathcal{U}_t * \mathcal{U}_t^\top) * (\mathcal{I} - \eta\mathcal{U}_t * \mathcal{U}_t^\top)}_{\mathcal{K}_1} \\
&\quad + \eta \underbrace{[(\mathfrak{I} - \mathfrak{M}^*\mathfrak{M})(\mathcal{X} * \mathcal{X}^\top - \mathcal{U}_t * \mathcal{U}_t^\top) + \mathcal{E}] * \mathcal{U}_t * \mathcal{U}_t^\top}_{\mathcal{K}_2} \\
&\quad + \eta \underbrace{\mathcal{U}_t * \mathcal{U}_t * [(\mathfrak{I} - \mathfrak{M}^*\mathfrak{M})(\mathcal{X} * \mathcal{X}^\top - \mathcal{U}_t * \mathcal{U}_t^\top) + \mathcal{E}]}_{\mathcal{K}_3} \\
&\quad - \eta^2 \underbrace{\mathcal{U}_t * \mathcal{U}_t^\top * (\mathcal{X} * \mathcal{X}^\top - \mathcal{U}_t * \mathcal{U}_t^\top) * \mathcal{U}_t * \mathcal{U}_t^\top}_{\mathcal{K}_4} \\
&\quad - \eta^2 \underbrace{[(\mathfrak{M}^*\mathfrak{M})(\mathcal{X} * \mathcal{X}^\top - \mathcal{U}_t * \mathcal{U}_t^\top) + \mathcal{E}] * \mathcal{U}_t * \mathcal{U}_t^\top * [(\mathfrak{M}^*\mathfrak{M})(\mathcal{X} * \mathcal{X}^\top - \mathcal{U}_t * \mathcal{U}_t^\top) + \mathcal{E}]}_{\mathcal{K}_5}.
\end{aligned}
$$

$$(83)$$

We now bound each of these terms separately.

**Bounding $\mathcal{K}_1$:** We note that

$$\mathcal{V}_{\boldsymbol{\mathcal{X}}}^{\top} * (\boldsymbol{\mathcal{I}} - \eta \boldsymbol{\mathcal{U}}_t * \boldsymbol{\mathcal{U}}_t^{\top}) * (\boldsymbol{\mathcal{X}} * \boldsymbol{\mathcal{X}}^{\top} - \boldsymbol{\mathcal{U}}_t * \boldsymbol{\mathcal{U}}_t^{\top}) * (\boldsymbol{\mathcal{I}} - \eta \boldsymbol{\mathcal{U}}_t * \boldsymbol{\mathcal{U}}_t^{\top})$$

$$= \mathcal{V}_{\boldsymbol{\mathcal{X}}}^{\top} * (\boldsymbol{\mathcal{I}} - \eta \boldsymbol{\mathcal{U}}_t * \boldsymbol{\mathcal{U}}_t^{\top}) * \mathcal{V}_{\boldsymbol{\mathcal{X}}} * \mathcal{V}_{\boldsymbol{\mathcal{X}}}^{\top} * (\boldsymbol{\mathcal{X}} * \boldsymbol{\mathcal{X}}^{\top} - \boldsymbol{\mathcal{U}}_t * \boldsymbol{\mathcal{U}}_t^{\top}) * (\boldsymbol{\mathcal{I}} - \eta \boldsymbol{\mathcal{U}}_t * \boldsymbol{\mathcal{U}}_t^{\top})$$

$$\quad + \mathcal{V}_{\boldsymbol{\mathcal{X}}}^{\top} * (\boldsymbol{\mathcal{I}} - \eta \boldsymbol{\mathcal{U}}_t * \boldsymbol{\mathcal{U}}_t^{\top}) * \mathcal{V}_{\boldsymbol{\mathcal{X}}\perp} * \mathcal{V}_{\boldsymbol{\mathcal{X}}\perp}^{\top} * (\boldsymbol{\mathcal{X}} * \boldsymbol{\mathcal{X}}^{\top} - \boldsymbol{\mathcal{U}}_t * \boldsymbol{\mathcal{U}}_t^{\top}) * (\boldsymbol{\mathcal{I}} - \eta \boldsymbol{\mathcal{U}}_t * \boldsymbol{\mathcal{U}}_t^{\top})$$

$$= \mathcal{V}_{\boldsymbol{\mathcal{X}}}^{\top} * (\boldsymbol{\mathcal{I}} - \eta \boldsymbol{\mathcal{U}}_t * \boldsymbol{\mathcal{U}}_t^{\top}) * \mathcal{V}_{\boldsymbol{\mathcal{X}}} * \mathcal{V}_{\boldsymbol{\mathcal{X}}}^{\top} * (\boldsymbol{\mathcal{X}} * \boldsymbol{\mathcal{X}}^{\top} - \boldsymbol{\mathcal{U}}_t * \boldsymbol{\mathcal{U}}_t^{\top}) * (\boldsymbol{\mathcal{I}} - \eta \boldsymbol{\mathcal{U}}_t * \boldsymbol{\mathcal{U}}_t^{\top}) \qquad (84)$$

$$\quad + \eta * \mathcal{V}_{\boldsymbol{\mathcal{X}}}^{\top} * \boldsymbol{\mathcal{U}}_t * \boldsymbol{\mathcal{U}}_t^{\top} * \mathcal{V}_{\boldsymbol{\mathcal{X}}\perp} * \mathcal{V}_{\boldsymbol{\mathcal{X}}\perp}^{\top} * \boldsymbol{\mathcal{U}}_t * \boldsymbol{\mathcal{U}}_t^{\top} * (\boldsymbol{\mathcal{I}} - \boldsymbol{\mathcal{U}}_t * \boldsymbol{\mathcal{U}}_t^{\top})$$

$$= (\boldsymbol{\mathcal{I}} - \eta \mathcal{V}_{\boldsymbol{\mathcal{X}}}^{\top} * \boldsymbol{\mathcal{U}}_t * \boldsymbol{\mathcal{U}}_t^{\top} * \mathcal{V}_{\boldsymbol{\mathcal{X}}}) * \mathcal{V}_{\boldsymbol{\mathcal{X}}}^{\top} * (\boldsymbol{\mathcal{X}} * \boldsymbol{\mathcal{X}}^{\top} - \boldsymbol{\mathcal{U}}_t * \boldsymbol{\mathcal{U}}_t^{\top}) * (\boldsymbol{\mathcal{I}} - \eta \boldsymbol{\mathcal{U}}_t * \boldsymbol{\mathcal{U}}_t^{\top})$$

$$\quad + \eta * \mathcal{V}_{\boldsymbol{\mathcal{X}}}^{\top} * \boldsymbol{\mathcal{U}}_t * \boldsymbol{\mathcal{U}}_t^{\top} * \mathcal{V}_{\boldsymbol{\mathcal{X}}\perp} * \mathcal{V}_{\boldsymbol{\mathcal{X}}\perp}^{\top} * \boldsymbol{\mathcal{U}}_t * \boldsymbol{\mathcal{U}}_t^{\top} * (\boldsymbol{\mathcal{I}} - \boldsymbol{\mathcal{U}}_t * \boldsymbol{\mathcal{U}}_t^{\top}).$$

detail

Therefore, we obtain

$$||\mathcal{V}_{\boldsymbol{\mathcal{X}}}^{\top} * \mathcal{K}_1|| = ||\mathcal{V}_{\boldsymbol{\mathcal{X}}}^{\top} * (\boldsymbol{\mathcal{I}} - \eta \boldsymbol{\mathcal{U}}_t * \boldsymbol{\mathcal{U}}_t^{\top}) * (\boldsymbol{\mathcal{X}} * \boldsymbol{\mathcal{X}}^{\top} - \boldsymbol{\mathcal{U}}_t * \boldsymbol{\mathcal{U}}_t^{\top}) * (\boldsymbol{\mathcal{I}} - \eta \boldsymbol{\mathcal{U}}_t * \boldsymbol{\mathcal{U}}_t^{\top})||$$

$$\leq \left(1 - \frac{\eta}{40} \sigma_{\min}^2(\boldsymbol{\mathcal{X}})\right) ||\mathcal{V}_{\boldsymbol{\mathcal{X}}}^{\top} * (\boldsymbol{\mathcal{X}} * \boldsymbol{\mathcal{X}}^{\top} - \boldsymbol{\mathcal{U}}_t * \boldsymbol{\mathcal{U}}_t^{\top})|| \qquad (85)$$

$$+ \eta \frac{\sigma_{\min}^2(\boldsymbol{\mathcal{X}})}{400} ||\boldsymbol{\mathcal{U}}_t * \mathcal{W}_{t,\perp} * \mathcal{W}_{t,\perp}^{\top} * \boldsymbol{\mathcal{U}}_t||.$$

**Bounding $\mathcal{K}_2$:** Note that

$$||\mathcal{V}_{\boldsymbol{\mathcal{X}}}^{\top} * \mathcal{K}_2|| = ||\mathcal{V}_{\boldsymbol{\mathcal{X}}}^{\top} * [(\mathfrak{I} - \mathfrak{M}^* \mathfrak{M})(\boldsymbol{\mathcal{X}} * \boldsymbol{\mathcal{X}}^{\top} - \boldsymbol{\mathcal{U}}_t * \boldsymbol{\mathcal{U}}_t^{\top}) + \boldsymbol{\mathcal{E}}] * \boldsymbol{\mathcal{U}}_t * \boldsymbol{\mathcal{U}}_t^{\top}||$$

$$\leq \left(||(\mathfrak{I} - \mathfrak{M}^* \mathfrak{M})(\boldsymbol{\mathcal{X}} * \boldsymbol{\mathcal{X}}^{\top} - \boldsymbol{\mathcal{U}}_t * \boldsymbol{\mathcal{U}}_t^{\top})|| + ||\mathcal{V}_{\boldsymbol{\mathcal{X}}}^{\top} * \boldsymbol{\mathcal{E}}||\right) ||\boldsymbol{\mathcal{U}}_t||^2$$

$$\overset{(1)}{\leq} 9 \left(||(\mathfrak{I} - \mathfrak{M}^* \mathfrak{M})(\boldsymbol{\mathcal{X}} * \boldsymbol{\mathcal{X}}^{\top} - \boldsymbol{\mathcal{U}}_t * \boldsymbol{\mathcal{U}}_t^{\top})|| + ||\mathcal{V}_{\boldsymbol{\mathcal{X}}}^{\top} * \boldsymbol{\mathcal{E}}||\right) ||\boldsymbol{\mathcal{X}}||^2$$

$$\overset{(2)}{\leq} 9c\sigma_{\min}^2(\boldsymbol{\mathcal{U}}) \left(||\boldsymbol{\mathcal{X}} * \boldsymbol{\mathcal{X}} - \boldsymbol{\mathcal{U}}_t * \mathcal{W}_t * \mathcal{W}_t^{\top} * \boldsymbol{\mathcal{U}}_t^{\top}|| + ||\boldsymbol{\mathcal{U}}_t * \mathcal{W}_{t,\perp} * \mathcal{W}_{t,\perp}^{\top} * \boldsymbol{\mathcal{U}}_t||_*\right)$$

$$\quad + 9||\mathcal{V}_{\boldsymbol{\mathcal{X}}}^{\top} * \boldsymbol{\mathcal{E}}|| ||\boldsymbol{\mathcal{X}}||^2$$

$$\overset{(3)}{\leq} 9c\sigma_{\min}^2(\boldsymbol{\mathcal{U}}) \left(||\mathcal{V}_{\boldsymbol{\mathcal{X}}}^{\top} * (\boldsymbol{\mathcal{X}} * \boldsymbol{\mathcal{X}} - \boldsymbol{\mathcal{U}}_t * \boldsymbol{\mathcal{U}}_t^{\top})|| + ||\boldsymbol{\mathcal{U}}_t * \mathcal{W}_{t,\perp} * \mathcal{W}_{t,\perp}^{\top} * \boldsymbol{\mathcal{U}}_t||_*\right)$$

$$\quad + 9||\mathcal{V}_{\boldsymbol{\mathcal{X}}}^{\top} * \boldsymbol{\mathcal{E}}|| ||\boldsymbol{\mathcal{X}}||^2$$

$$(86)$$

where (1) use the assumption $||\boldsymbol{\mathcal{U}}_t|| \leq 3||\boldsymbol{\mathcal{X}}||$; (2) use the assumption (76); (3) use the the result of Lemma 8. Taking a small constant $c > 0$, we obtain

$$||\mathcal{V}_{\boldsymbol{\mathcal{X}}}^{\top} * [(\mathfrak{I} - \mathfrak{M}^* \mathfrak{M})(\boldsymbol{\mathcal{X}} * \boldsymbol{\mathcal{X}}^{\top} - \boldsymbol{\mathcal{U}}_t * \boldsymbol{\mathcal{U}}_t^{\top}) + \boldsymbol{\mathcal{E}}] * \boldsymbol{\mathcal{U}}_t * \boldsymbol{\mathcal{U}}_t^{\top}||$$

$$\leq \frac{1}{1000} \sigma_{\min}^2(\boldsymbol{\mathcal{X}}) \left(||\mathcal{V}_{\boldsymbol{\mathcal{X}}}^{\top} * (\boldsymbol{\mathcal{X}} * \boldsymbol{\mathcal{X}} - \boldsymbol{\mathcal{U}}_t * \boldsymbol{\mathcal{U}}_t^{\top})|| + ||\boldsymbol{\mathcal{U}}_t * \mathcal{W}_{t,\perp} * \mathcal{W}_{t,\perp}^{\top} * \boldsymbol{\mathcal{U}}_t||_*\right) \qquad (87)$$

$$\quad + 9||\mathcal{V}_{\boldsymbol{\mathcal{X}}}^{\top} * \boldsymbol{\mathcal{E}}|| ||\boldsymbol{\mathcal{X}}||^2.$$

**Bounding $\mathcal{K}_3$:** Similar to $\mathcal{K}_2$, we have

$$||\mathcal{V}_{\boldsymbol{\mathcal{X}}}^{\top} * \boldsymbol{\mathcal{U}}_t * \boldsymbol{\mathcal{U}}_t * [(\mathfrak{I} - \mathfrak{M}^* \mathfrak{M})(\boldsymbol{\mathcal{X}} * \boldsymbol{\mathcal{X}}^{\top} - \boldsymbol{\mathcal{U}}_t * \boldsymbol{\mathcal{U}}_t^{\top}) + \boldsymbol{\mathcal{E}}]||$$

$$\leq \frac{1}{1000} \sigma_{\min}^2(\boldsymbol{\mathcal{X}}) \left(||\mathcal{V}_{\boldsymbol{\mathcal{X}}}^{\top} * (\boldsymbol{\mathcal{X}} * \boldsymbol{\mathcal{X}} - \boldsymbol{\mathcal{U}}_t * \boldsymbol{\mathcal{U}}_t^{\top})|| + ||\boldsymbol{\mathcal{U}}_t * \mathcal{W}_{t,\perp} * \mathcal{W}_{t,\perp}^{\top} * \boldsymbol{\mathcal{U}}_t||_*\right) \qquad (88)$$

$$\quad + 9||\mathcal{V}_{\boldsymbol{\mathcal{X}}}^{\top} * \boldsymbol{\mathcal{E}}|| ||\boldsymbol{\mathcal{X}}||^2.$$

**Bounding $\mathcal{K}_4$:** Note that

$$
\begin{aligned}
||\mathcal{V}_{\mathcal{X}}^\top &* \mathcal{U}_t * \mathcal{U}_t^\top * (\mathcal{X} * \mathcal{X}^\top - \mathcal{U}_t * \mathcal{U}_t^\top) * \mathcal{U}_t * \mathcal{U}_t^\top|| \\
&\leq ||\mathcal{U}_t||^4 ||\mathcal{X} * \mathcal{X}^\top - \mathcal{U}_t * \mathcal{U}_t^\top|| \\
&\overset{(1)}{\lesssim} ||\mathcal{X}||^4 ||\mathcal{X} * \mathcal{X}^\top - \mathcal{U}_t * \mathcal{U}_t^\top|| \\
&\overset{(2)}{\lesssim} ||\mathcal{X}||^4 \left( ||\mathcal{V}_{\mathcal{X}}^\top * (\mathcal{X} * \mathcal{X}^\top - \mathcal{U}_t * \mathcal{U}_t^\top)|| + ||\mathcal{U}_t * \mathcal{W}_{t,\perp} * \mathcal{W}_{t,\perp}^\top * \mathcal{U}_t^\top|| \right),
\end{aligned}
\tag{89}
$$

where (1) uses the assumption $||\mathcal{U}_t|| \leq 3||\mathcal{X}||$; (2) uses the result of Lemma 8. Then combining the assumption $\eta \leq c\kappa^{-2}||\mathcal{X}||^{-2}$, then we obtain

$$
\begin{aligned}
\eta^2 ||\mathcal{V}_{\mathcal{X}}^\top &* \mathcal{U}_t * \mathcal{U}_t^\top * (\mathcal{X} * \mathcal{X}^\top - \mathcal{U}_t * \mathcal{U}_t^\top) * \mathcal{U}_t * \mathcal{U}_t^\top|| \\
&\leq \frac{\eta}{200} \sigma_{\min}^2(\mathcal{X}) ||\mathcal{V}_{\mathcal{X}}^\top * (\mathcal{X} * \mathcal{X}^\top - \mathcal{U}_t * \mathcal{U}_t^\top)|| + \eta \frac{\sigma_{\min}^2(\mathcal{X})}{1000} ||\mathcal{U}_t * \mathcal{W}_{t,\perp} * \mathcal{W}_{t,\perp}^\top * \mathcal{U}_t^\top||.
\end{aligned}
\tag{90}
$$

**Bounding $\mathcal{K}_5$:** Note that

$$
\begin{aligned}
||(\mathfrak{M}^* \mathfrak{M})&(\mathcal{X} * \mathcal{X}^\top - \mathcal{U}_t * \mathcal{U}_t^\top)|| \\
&\leq ||\mathcal{X} * \mathcal{X}^\top - \mathcal{U}_t * (\mathcal{W}_t * \mathcal{W}_t^\top + \mathcal{W}_{t,\perp} * \mathcal{W}_{t,\perp}^\top) * \mathcal{U}_t^\top|| \\
&\quad + ||(\mathfrak{M}^* \mathfrak{M} - \mathfrak{I})(\mathcal{X} * \mathcal{X}^\top - \mathcal{U}_t * \mathcal{U}_t^\top)|| \\
&\overset{(a)}{\leq} \left( ||\mathcal{X} * \mathcal{X}^\top - \mathcal{U}_t * \mathcal{W}_t * \mathcal{W}_t^\top * \mathcal{U}_t^\top|| + ||\mathcal{U}_t * \mathcal{W}_{t,\perp} * \mathcal{W}_{t,\perp}^\top \mathcal{U}_t^\top|| \right) \\
&\quad + c\kappa^{-2} \left( ||\mathcal{X} * \mathcal{X}^\top - \mathcal{U}_t * \mathcal{W}_t * \mathcal{W}_t^\top * \mathcal{U}_t^\top|| + ||\mathcal{U}_t * \mathcal{W}_{t,\perp} * \mathcal{W}_{t,\perp}^\top \mathcal{U}_t^\top||_* \right) \\
&\leq 2 \left( ||\mathcal{X} * \mathcal{X}^\top - \mathcal{U}_t * \mathcal{W}_t * \mathcal{W}_t^\top * \mathcal{U}_t^\top|| + ||\mathcal{U}_t * \mathcal{W}_{t,\perp} * \mathcal{W}_{t,\perp}^\top \mathcal{U}_t^\top||_* \right) \\
&\leq 2 \left( ||\mathcal{X}||^2 + ||\mathcal{U}_t * \mathcal{W}_t||^2 + ||\mathcal{U}_t * \mathcal{W}_{t,\perp} * \mathcal{W}_{t,\perp}^\top \mathcal{U}_t^\top||_* \right) \\
&\overset{(b)}{\leq} 2 \left( ||\mathcal{X}||^2 + 2||\mathcal{U}_t||^2 \right),
\end{aligned}
\tag{91}
$$

where (a) uses the assumption (76); (b) uses the assumption $||\mathcal{U}_t * \mathcal{W}_{t,\perp} * \mathcal{W}_{t,\perp}^\top \mathcal{U}_t^\top|| \leq ||\mathcal{U}_t||^2$. Then we have

$$
\begin{aligned}
||\mathcal{V}_{\mathcal{X}}^\top * \mathcal{K}_5|| &= ||\mathcal{V}_{\mathcal{X}}^\top * [(\mathfrak{M}^* \mathfrak{M})(\mathcal{X} * \mathcal{X}^\top - \mathcal{U}_t * \mathcal{U}_t^\top) + \mathcal{E}] * \mathcal{U}_t * \mathcal{U}_t^\top * [(\mathfrak{M}^* \mathfrak{M})(\mathcal{X} * \mathcal{X}^\top - \mathcal{U}_t * \mathcal{U}_t^\top) + \mathcal{E}]|| \\
&\leq \left( ||(\mathfrak{M}^* \mathfrak{M})(\mathcal{X} * \mathcal{X}^\top - \mathcal{U}_t * \mathcal{U}_t^\top)|| + ||\mathcal{E}|| \right) \cdot ||\mathcal{U}_t||^2 \cdot \left( ||(\mathfrak{M}^* \mathfrak{M})(\mathcal{X} * \mathcal{X}^\top - \mathcal{U}_t * \mathcal{U}_t^\top)|| + ||\mathcal{E}|| \right) \\
&\overset{(a)}{\leq} 4 \left( ||\mathcal{X} * \mathcal{X}^\top - \mathcal{U}_t * \mathcal{W}_t * \mathcal{W}_t^\top * \mathcal{U}_t^\top|| + ||\mathcal{U}_t * \mathcal{W}_{t,\perp} * \mathcal{W}_{t,\perp}^\top * \mathcal{U}_t^\top||_* + ||\mathcal{E}|| \right) ||\mathcal{U}_t||^2 \left( ||\mathcal{X}||^2 + 2||\mathcal{U}_t||^2 + ||\mathcal{E}|| \right) \\
&\overset{(b)}{\leq} 432 \left( ||\mathcal{X} * \mathcal{X}^\top - \mathcal{U}_t * \mathcal{W}_t * \mathcal{W}_t^\top * \mathcal{U}_t^\top|| + ||\mathcal{U}_t * \mathcal{W}_{t,\perp} * \mathcal{W}_{t,\perp}^\top * \mathcal{U}_t^\top||_* + ||\mathcal{E}|| \right) ||\mathcal{U}_t||^4 \\
&\overset{(c)}{\leq} 1728 \left( ||\mathcal{V}_{\mathcal{X}}^\top * (\mathcal{X} * \mathcal{X}^\top - \mathcal{U}_t * \mathcal{U}_t^\top)|| + ||\mathcal{U}_t * \mathcal{W}_{t,\perp} * \mathcal{W}_{t,\perp}^\top * \mathcal{U}_t^\top||_* + ||\mathcal{E}|| \right) ||\mathcal{U}_t||^4,
\end{aligned}
\tag{92}
$$

where (a) uses the result of Equation (91); (b) uses the assumptions $||\mathcal{U}_t|| \leq 3||\mathcal{X}||$ and $||\mathcal{E}|| \leq ||\mathcal{X}||^2$; (c) uses the result of Lemma 8. Based on these results and the assumption $\eta \leq c\kappa^{-2}||\mathcal{X}||^{-2}$, we have

$$
\begin{aligned}
\eta^2 ||\mathcal{V}_{\mathcal{X}}^\top &* [(\mathfrak{M}^* \mathfrak{M})(\mathcal{X} * \mathcal{X}^\top - \mathcal{U}_t * \mathcal{U}_t^\top) + \mathcal{E}] * \mathcal{U}_t * \mathcal{U}_t^\top * [(\mathfrak{M}^* \mathfrak{M})(\mathcal{X} * \mathcal{X}^\top - \mathcal{U}_t * \mathcal{U}_t^\top) + \mathcal{E}]|| \\
&\leq \frac{\eta}{1000} \sigma_{\min}^2(\mathcal{X}) ||\mathcal{V}_{\mathcal{X}}^\top * (\mathcal{X} * \mathcal{X}^\top - \mathcal{U}_t * \mathcal{U}_t^\top)|| + \frac{\eta}{400} \sigma_{\min}^2(\mathcal{X}) ||\mathcal{U}_t * \mathcal{W}_{t,\perp} * \mathcal{W}_{t,\perp}^\top * \mathcal{U}_t^\top||_* \\
&\quad + \frac{\eta}{400} \sigma_{\min}^2(\mathcal{X}) ||\mathcal{E}||.
\end{aligned}
\tag{93}
$$

Combining the bounds of these five terms, we obtain

$$||\mathcal{V}_{\mathcal{X}}^\top * (\mathcal{X} * \mathcal{X}^\top - \mathcal{U}_{t+1} * \mathcal{U}_{t+1}^\top)|| \leq \left(1 - \frac{\eta}{200}\sigma_{\min}(\mathcal{X})^2\right)||\mathcal{V}_{\mathcal{X}}^\top * (\mathcal{X} * \mathcal{X}^\top - \mathcal{U}_t * \mathcal{U}_t^\top)||$$
$$+ \frac{\eta}{200}\sigma_{\min}(\mathcal{X})^2||\mathcal{U}_t * \mathcal{W}_{t,\perp} * \mathcal{W}_{t,\perp}^\top * \mathcal{U}_t^\top|| + 18\eta||\mathcal{X}||^2||\mathcal{E}||.$$

$$(94)$$

$\square$

## F PROOF OF THEOREM 3

The proof of the minimax error bound of the low-tubal-rank tensor recovery follows from the proof of the matrix case in (Candes & Plan, 2011). We begin with a standard lemma that characterizes the minimax risk for estimating a vector $x \in \mathbb{R}^n$ in the linear model

$$y = Ax + s \tag{95}$$

where $A \in \mathbb{R}^{m \times n}$ and the entries of $s$ are independently and identically distributed according to a Gaussian distribution $\mathcal{N}(0, \sigma^2)$. For such a model, we have the following lemma that provides its minimax error bound.

**Lemma 9.** *Let $\lambda_i(A^\top A)$ be the eigenvalues of the matrix $A^\top A$. Then*

$$\inf_{\hat{x}} \sup_{x \in \mathbb{R}^n} \mathbb{E}||\hat{x} - x||_{l_2}^2 = \sigma^2 \text{ trace}\left((A^\top A)^{-1}\right) = \sum_i \frac{\sigma^2}{\lambda_i(A^\top A)}. \tag{96}$$

*In particular, if one of the eigenvalues vanishes, then the minimax risk is unbounded.*

*Proof.* We separate the argument into three parts: (A) a lower bound via Bayes risk, (B) an upper bound attained by the ordinary least squares estimator in the nonsingular case.

**(A) Lower bound (Bayes argument)**
Fix $\tau > 0$ and consider the Gaussian prior $x \sim \mathcal{N}(0, \tau^2 I_n)$. Under this prior the posterior covariance matrix for $x$ given $y$ is

$$\Sigma_{\text{post}}(\tau) = \left(\frac{1}{\sigma^2}A^\top A + \frac{1}{\tau^2}I_n\right)^{-1}.$$

The Bayes risk (for the posterior-mean estimator) equals the trace of the posterior covariance:

$$R_{\text{Bayes}}(\tau) := \mathbb{E}_{x,y}\left\|x - \mathbb{E}[x \mid y]\right\|_2^2 = \text{tr}\left(\Sigma_{\text{post}}(\tau)\right)$$
$$= \sum_{i=1}^n \frac{1}{\lambda_i/\sigma^2 + 1/\tau^2} = \sigma^2 \sum_{i=1}^n \frac{1}{\lambda_i + \sigma^2/\tau^2}, \tag{97}$$

where we denote $\lambda_i(A^\top A)$ as $\lambda_i$ for convenience.

For any estimator $\hat{x}$ and any prior $\pi$ we have the standard minimax/Bayes inequality

$$\inf_{\hat{x}} \sup_{x} \mathbb{E}||\hat{x} - x||^2 \geq \inf_{\hat{x}} \mathbb{E}_{x \sim \pi}\mathbb{E}\left[||\hat{x} - x||^2\right] = R_{\text{Bayes}}(\tau),$$

because the supremum over $x$ is at least the average under any prior $\pi$. Hence for every $\tau > 0$,

$$\inf_{\hat{x}} \sup_{x} \mathbb{E}||\hat{x} - x||^2 \geq \sigma^2 \sum_{i=1}^n \frac{1}{\lambda_i + \sigma^2/\tau^2}.$$

If some $\lambda_i = 0$ then the right-hand side equals $+\infty$ as $\tau \to \infty$ (indeed the corresponding summand is $\sigma^2/(0 + \sigma^2/\tau^2) = \tau^2 \to \infty$), so the minimax risk is infinite in that case. Otherwise, if all $\lambda_i > 0$,

send $\tau \to \infty$. For each fixed $i$ the function $\tau \mapsto \sigma^2/(\lambda_i + \sigma^2/\tau^2)$ is monotone increasing in $\tau$ and converges to $\sigma^2/\lambda_i$ as $\tau \to \infty$. By monotone convergence (or by continuity of finite sums) we obtain

$$\inf_{\hat{\boldsymbol{x}}} \sup_{\boldsymbol{x}} \mathbb{E}\|\hat{\boldsymbol{x}} - \boldsymbol{x}\|^2 \geq \lim_{\tau \to \infty} R_{\text{Bayes}}(\tau) = \sigma^2 \sum_{i=1}^{n} \frac{1}{\lambda_i}.$$

**(B) Upper bound (least squares achieves the bound).**
Assume $\lambda_i > 0$ for all $i$, i.e. $\text{rank}(A) = n$. Consider the ordinary least squares estimator

$$\hat{\boldsymbol{x}}_{\text{LS}} = (\boldsymbol{A}^\top \boldsymbol{A})^{-1} \boldsymbol{A}^\top \boldsymbol{y}.$$

Substituting $\boldsymbol{y} = \boldsymbol{A}\boldsymbol{x} + \boldsymbol{s}$ gives

$$\hat{\boldsymbol{x}}_{\text{LS}} - \boldsymbol{x} = (\boldsymbol{A}^\top \boldsymbol{A})^{-1} \boldsymbol{A}^\top \boldsymbol{s}.$$

Since $\boldsymbol{s} \sim \mathcal{N}(0, \sigma^2 I_m)$, the error $\hat{\boldsymbol{x}}_{\text{LS}} - \boldsymbol{x}$ is zero-mean Gaussian with covariance

$$\mathbb{E}\left[(\hat{\boldsymbol{x}}_{\text{LS}} - \boldsymbol{x})(\hat{\boldsymbol{x}}_{\text{LS}} - \boldsymbol{x})^\top\right] = (\boldsymbol{A}^\top \boldsymbol{A})^{-1} \boldsymbol{A}^\top (\sigma^2 \boldsymbol{I_m}) \boldsymbol{A} (\boldsymbol{A}^\top \boldsymbol{A})^{-1} = \sigma^2 (\boldsymbol{A}^\top \boldsymbol{A})^{-1}.$$

Therefore the mean-square risk of $\hat{\boldsymbol{x}}_{\text{LS}}$ (for any fixed $\boldsymbol{x}$) equals

$$\mathbb{E}\|\hat{\boldsymbol{x}}_{\text{LS}} - \boldsymbol{x}\|^2 = \text{tr}\left(\sigma^2 (\boldsymbol{A}^\top \boldsymbol{A})^{-1}\right) = \sigma^2 \sum_{i=1}^{n} \frac{1}{\lambda_i}.$$

This shows

$$\inf_{\hat{\boldsymbol{x}}} \sup_{\boldsymbol{x}} \mathbb{E}\|\hat{\boldsymbol{x}} - \boldsymbol{x}\|^2 \leq \sup_{\boldsymbol{x}} \mathbb{E}\|\hat{\boldsymbol{x}}_{\text{LS}} - \boldsymbol{x}\|^2 = \sigma^2 \sum_{i=1}^{n} \frac{1}{\lambda_i}.$$

Combining (A) and (B) yields the asserted identity. $\qquad\square$

Then we proceed to prove the minimax error bound. Define the set of $\text{rank}_t$ r tensors as

$$\mathcal{D}_r = \{\boldsymbol{\mathcal{X}} : \text{rank}_t(\boldsymbol{\mathcal{X}}) = r, \ \boldsymbol{\mathcal{X}} \in \mathbb{R}^{n \times n \times k}\},$$

and the set of tensors of the form $\boldsymbol{\mathcal{X}} = \boldsymbol{\mathcal{Y}} * \boldsymbol{\mathcal{R}}$ as

$$\mathcal{D}_{\boldsymbol{\mathcal{Y}}} = \{\boldsymbol{\mathcal{X}} : \boldsymbol{\mathcal{X}} = \boldsymbol{\mathcal{Y}} * \boldsymbol{\mathcal{R}}, \ \boldsymbol{\mathcal{X}} \in \mathbb{R}^{n \times r \times k}, \ \boldsymbol{\mathcal{Y}} \in \mathbb{R}^{n \times r \times k}, \ \boldsymbol{\mathcal{Y}}^\top * \boldsymbol{\mathcal{Y}} = \boldsymbol{\mathcal{I}}\}.$$

Note that set $\mathcal{D}_r$ is much larger than set $\mathcal{D}_{\boldsymbol{\mathcal{Y}}}$. Therefore,

$$\inf_{\boldsymbol{\mathcal{X}}_{est}} \sup_{\boldsymbol{\mathcal{X}}:\text{rank}_t(\boldsymbol{\mathcal{X}})=r} \mathbb{E}\|\boldsymbol{\mathcal{X}}_{est} - \boldsymbol{\mathcal{X}}\|_F^2 \geq \inf_{\boldsymbol{\mathcal{X}}_{est}} \sup_{\boldsymbol{\mathcal{X}}:\boldsymbol{\mathcal{X}}=\boldsymbol{\mathcal{Y}}*\boldsymbol{\mathcal{R}}} \mathbb{E}\|\boldsymbol{\mathcal{X}}_{est} - \boldsymbol{\mathcal{X}}\|_F^2. \tag{98}$$

For fixed orthogonal tensor $\boldsymbol{\mathcal{Y}}$, define the orthogonal projection tensor $\boldsymbol{\mathcal{P}}_{\boldsymbol{\mathcal{Y}}} = \boldsymbol{\mathcal{Y}} * \boldsymbol{\mathcal{Y}}^\top$, which satisfies $\boldsymbol{\mathcal{P}}_{\boldsymbol{\mathcal{Y}}}^2 = \boldsymbol{\mathcal{P}}_{\boldsymbol{\mathcal{Y}}}, \boldsymbol{\mathcal{P}}_{\boldsymbol{\mathcal{Y}}}^\top = \boldsymbol{\mathcal{P}}_{\boldsymbol{\mathcal{Y}}}$. Then fix the estimator $\boldsymbol{\mathcal{X}}_{est}$, for any $\boldsymbol{\mathcal{X}} \in \mathcal{D}_{\boldsymbol{\mathcal{Y}}}$, we have:

$$\begin{aligned}
\|\boldsymbol{\mathcal{X}}_{est} - \boldsymbol{\mathcal{X}}\|_F^2 &= \|\boldsymbol{\mathcal{P}}_{\boldsymbol{\mathcal{Y}}} * \boldsymbol{\mathcal{X}}_{est} - \boldsymbol{\mathcal{X}} + (\boldsymbol{\mathcal{I}} - \boldsymbol{\mathcal{P}}_{\boldsymbol{\mathcal{Y}}}) * \boldsymbol{\mathcal{X}}_{est}\|_F^2 \\
&= \|\boldsymbol{\mathcal{P}}_{\boldsymbol{\mathcal{Y}}} * \boldsymbol{\mathcal{X}}_{est} - \boldsymbol{\mathcal{X}}\|_F + \|(\boldsymbol{\mathcal{I}} - \boldsymbol{\mathcal{P}}_{\boldsymbol{\mathcal{Y}}}) * \boldsymbol{\mathcal{X}}_{est}\|_F^2 \\
&\quad + 2\langle \boldsymbol{\mathcal{P}}_{\boldsymbol{\mathcal{Y}}} * \boldsymbol{\mathcal{X}}_{est} - \boldsymbol{\mathcal{X}}, (\boldsymbol{\mathcal{I}} - \boldsymbol{\mathcal{P}}_{\boldsymbol{\mathcal{Y}}}) * \boldsymbol{\mathcal{X}}_{est}\rangle \\
&\overset{(a)}{=} \|\boldsymbol{\mathcal{P}}_{\boldsymbol{\mathcal{Y}}} * \boldsymbol{\mathcal{X}}_{est} - \boldsymbol{\mathcal{X}}\|_F + \|(\boldsymbol{\mathcal{I}} - \boldsymbol{\mathcal{P}}_{\boldsymbol{\mathcal{Y}}}) * \boldsymbol{\mathcal{X}}_{est}\|_F^2,
\end{aligned} \tag{99}$$

where $(a)$ use the fact that the tensor column subspaces of $\boldsymbol{\mathcal{P}}_{\boldsymbol{\mathcal{Y}}} * \boldsymbol{\mathcal{X}}_{est} - \boldsymbol{\mathcal{X}}$ and $(\boldsymbol{\mathcal{I}} - \boldsymbol{\mathcal{P}}_{\boldsymbol{\mathcal{Y}}}) * \boldsymbol{\mathcal{X}}_{est}$ are orthogonal, which implies that their inner product vanishes. Therefore, we can directly obtain

$$\begin{aligned}
\|\boldsymbol{\mathcal{X}}_{est} - \boldsymbol{\mathcal{X}}\|_F^2 &\geq \|\boldsymbol{\mathcal{P}}_{\boldsymbol{\mathcal{Y}}} * \boldsymbol{\mathcal{X}}_{est} - \boldsymbol{\mathcal{X}}\|_F^2 = \|\boldsymbol{\mathcal{Y}} * \boldsymbol{\mathcal{Y}}^\top * \boldsymbol{\mathcal{X}}_{est} - \boldsymbol{\mathcal{Y}} * \boldsymbol{\mathcal{R}}\|_F^2 \\
&= \|\boldsymbol{\mathcal{Y}}^\top * \boldsymbol{\mathcal{X}}_{est} - \boldsymbol{\mathcal{R}}\|_F^2.
\end{aligned} \tag{100}$$

Let $\boldsymbol{\mathcal{R}}_{est} = \boldsymbol{\mathcal{Y}}^{\top} * \boldsymbol{\mathcal{X}}_{est}$, then we have

$$\inf_{\boldsymbol{\mathcal{X}}_{est}} \sup_{\boldsymbol{\mathcal{X}}:\boldsymbol{\mathcal{X}}=\boldsymbol{\mathcal{U}}*\boldsymbol{\mathcal{R}}} \mathbb{E}||\boldsymbol{\mathcal{X}}_{est} - \boldsymbol{\mathcal{X}}||_F^2 \geq \inf_{\boldsymbol{\mathcal{R}}_{est}} \sup_{\boldsymbol{\mathcal{R}}} \mathbb{E}||\boldsymbol{\mathcal{R}}_{est} - \boldsymbol{\mathcal{R}}||_F^2. \tag{101}$$

Therefore, the minimax risk is lower bounded by that of estimating $\boldsymbol{\mathcal{R}}$ from the data

$$\boldsymbol{y} = \mathfrak{M}_{\boldsymbol{\mathcal{Y}}}(\mathrm{vec}(\boldsymbol{\mathcal{R}})) + \boldsymbol{s}, \tag{102}$$

where $\mathfrak{M}_{\boldsymbol{\mathcal{Y}}} : \mathbb{R}^{rnk} \to \mathbb{R}^m$ and vec denotes the vectorization operator. Then we can apply the result of Lemma 9 to show that the minimax risk is lower bounded by

$$\inf_{\boldsymbol{\mathcal{X}}_{est}} \sup_{\boldsymbol{\mathcal{X}}:\mathrm{rank}_t(\boldsymbol{\mathcal{X}})=r} \mathbb{E}||\boldsymbol{\mathcal{X}}_{est} - \boldsymbol{\mathcal{X}}||_F^2 \geq \sum_i^{rnk} \frac{\sigma^2}{\lambda_i(\mathfrak{M}_{\boldsymbol{\mathcal{Y}}}^* \mathfrak{M}_{\boldsymbol{\mathcal{Y}}})}. \tag{103}$$

Then we can bound the term (103) by the following Lemma with t-RIP assumption.

**Lemma 10.** *Let $\boldsymbol{\mathcal{Y}}$ be an $n \times r \times k$ orthonormal tensor, suppose that the linear map $\mathfrak{M}(\cdot)$ satisfies the $(r, \delta)$ t-RIP, then all eigenvalues of $\mathfrak{M}_{\boldsymbol{\mathcal{Y}}}^* \mathfrak{M}_{\boldsymbol{\mathcal{Y}}}$ belong to the interval $[m(1-\delta), m(1+\delta)]$.*

*Proof.* By definition, we have

$$\begin{aligned}
\lambda_{\min}(\mathfrak{M}_{\boldsymbol{\mathcal{Y}}}^* \mathfrak{M}_{\boldsymbol{\mathcal{Y}}}) &= \inf_{||\mathrm{vec}(\boldsymbol{\mathcal{R}})||_F=1} \left\langle \mathrm{vec}(\boldsymbol{\mathcal{R}}), \mathfrak{M}_{\boldsymbol{\mathcal{Y}}}^* \mathfrak{M}_{\boldsymbol{\mathcal{Y}}}(\mathrm{vec}(\boldsymbol{\mathcal{R}})) \right\rangle \\
\lambda_{\max}(\mathfrak{M}_{\boldsymbol{\mathcal{Y}}}^* \mathfrak{M}_{\boldsymbol{\mathcal{Y}}}) &= \sup_{||\mathrm{vec}(\boldsymbol{\mathcal{R}})||_F=1} \left\langle \mathrm{vec}(\boldsymbol{\mathcal{R}}), \mathfrak{M}_{\boldsymbol{\mathcal{Y}}}^* \mathfrak{M}_{\boldsymbol{\mathcal{Y}}}(\mathrm{vec}(\boldsymbol{\mathcal{R}})) \right\rangle.
\end{aligned} \tag{104}$$

Note that

$$\left\langle \boldsymbol{\mathcal{R}}, \mathfrak{M}_{\boldsymbol{\mathcal{Y}}}^* \mathfrak{M}_{\boldsymbol{\mathcal{Y}}}(\mathrm{vec}(\boldsymbol{\mathcal{R}})) \right\rangle = ||\mathfrak{M}_{\boldsymbol{\mathcal{Y}}}(\mathrm{vec}(\boldsymbol{\mathcal{R}}))||^2 = ||\mathfrak{M}(\boldsymbol{\mathcal{Y}} * \boldsymbol{\mathcal{R}})||^2,$$

then we can bound $||\mathfrak{M}(\boldsymbol{\mathcal{Y}} * \boldsymbol{\mathcal{R}})||^2$ by the $(r, \delta)$ t-RIP

$$m(1-\delta)||\boldsymbol{\mathcal{Y}} * \boldsymbol{\mathcal{R}}||_F^2 \leq ||\mathfrak{M}(\boldsymbol{\mathcal{Y}} * \boldsymbol{\mathcal{R}})||^2 \leq m(1+\delta)||\boldsymbol{\mathcal{Y}} * \boldsymbol{\mathcal{R}}||_F^2.$$

Since $||\boldsymbol{\mathcal{Y}} * \boldsymbol{\mathcal{R}}||_F = ||\boldsymbol{\mathcal{R}}||_F = 1$, then the eigenvalues of $\mathfrak{M}_{\boldsymbol{\mathcal{Y}}}^* \mathfrak{M}_{\boldsymbol{\mathcal{Y}}}$ is bounded by $[m(1-\delta), m(1+\delta)]$. $\square$

Combining the result of Lemma 10 and Equation (103), we have

$$\inf_{\boldsymbol{\mathcal{X}}_{est}} \sup_{\boldsymbol{\mathcal{X}}:\mathrm{rank}_t(\boldsymbol{\mathcal{X}})=r} \mathbb{E}||\boldsymbol{\mathcal{X}}_{est} - \boldsymbol{\mathcal{X}}||_F^2 \geq \sum_i^{rnk} \frac{\sigma^2}{\lambda_i(\mathfrak{M}_{\boldsymbol{\mathcal{Y}}}^* \mathfrak{M}_{\boldsymbol{\mathcal{Y}}})} \geq \frac{1}{1+\delta} \frac{nrk\sigma^2}{m}, \tag{105}$$

which finishes the proof of the first inequality in Theorem 3.

Then we proceed to prove the second inequality in Theorem 3. We introduce a technical Lemma firstly.

**Lemma 11** (Lemma 3.14 in (Candes & Plan, 2011))**.** *Suppose that $\boldsymbol{x}, \boldsymbol{y}, \boldsymbol{A}, \boldsymbol{s}$ follow the linear model (95), with $\boldsymbol{s} \sim \mathcal{N}(0, \sigma^2 \boldsymbol{I})$, then*

$$\inf_{\hat{\boldsymbol{x}}} \sup_{\boldsymbol{x} \in \mathbb{R}^n} \mathbb{P}\left( ||\hat{\boldsymbol{x}} - \boldsymbol{x}||^2 \geq \frac{1}{2||\boldsymbol{A}||^2} n\sigma^2 \right) \geq 1 - e^{-n/16}. \tag{106}$$

With the result of Lemmas 11, 10 and the linear model

$$\boldsymbol{y} = \mathfrak{M}_{\boldsymbol{\mathcal{Y}}}(\mathrm{vec}(\boldsymbol{\mathcal{R}})) + \boldsymbol{s}, \ \boldsymbol{\mathcal{R}} \in \mathbb{R}^{n \times r \times k}, \ \boldsymbol{y} \in \mathbb{R}^m,$$

we can obtain

$$\sup_{\boldsymbol{\mathcal{X}}_\star:\mathrm{rank}_t(\boldsymbol{\mathcal{X}}_\star) \leq r} \mathbb{P}\left( ||\boldsymbol{\mathcal{X}}_{est} - \boldsymbol{\mathcal{X}}_\star||_F^2 \geq \frac{nrk\sigma^2}{2m(1+\delta)} \right) \geq 1 - e^{-nrk/16}, \tag{107}$$

which completes the proof of the second inequality.

# G   PROOF OF THEOREM 4

**Lemma 12.** *Suppose that each entry in the validation measurements $\boldsymbol{\mathcal{A}}_i$, $i \in \mathcal{I}_{\text{val}}$ is sampled from independent identically sub-Gaussian distribution with zero mean and variance 1, and each $e_i$ is a zero-mean Gaussian distribution with variance $\sigma^2$, where $c_1$, $c_2 \geq 1$ are some absolute constants. And we also assume that tensors $\boldsymbol{\mathcal{D}}_1$, $\boldsymbol{\mathcal{D}}_2, ..., \boldsymbol{\mathcal{D}}_T$ are independent of $\mathfrak{M}_{\text{val}}$ and $\boldsymbol{e}_{\text{val}}$. Then for any $\delta_{\text{val}} > 0$, given $m_{\text{val}} \geq \frac{C_1 \log T}{\delta_{\text{val}}^2}$, with probability at least $1 - 2T \exp{-C_2 m_{\text{val}} \delta^2}$,*

$$\left| ||\mathfrak{M}_{\text{val}}(\boldsymbol{\mathcal{D}}_t) + \boldsymbol{e}||_F^2 - m_{\text{val}}(||\boldsymbol{\mathcal{D}}_t||_F^2 + \sigma^2) \right| \leq \delta_{\text{val}} m_{\text{val}}(||\boldsymbol{\mathcal{D}}_t||_F^2 + \sigma^2), \forall t = 1, ..., T, \quad (108)$$

*where $C_1$, $C_2 \geq 0$ are constants that may depend on $c_1$ and $c_2$.*

*Proof.* The proof of this lemma follows directly from Lemma D.1 in (Ding et al., 2025) , since $\langle \boldsymbol{\mathcal{A}}_i, \boldsymbol{\mathcal{D}} \rangle + e_i$ is a sub-Gaussian random variable with zero mean and variance $||\boldsymbol{\mathcal{D}}||_F^2 + \sigma^2$, regardless of whether $\boldsymbol{\mathcal{A}}_i$ and $\boldsymbol{\mathcal{D}}$ are matrices or tensors. Therefore, the conclusion of Lemma D.1 applies directly to this lemma. □

**Lemma 13.** *Let $\check{t} = \arg\min_{1 \leq t \leq T} ||\mathfrak{M}_{\text{val}}(\boldsymbol{\mathcal{D}}_t) + \boldsymbol{e}||_2$ and $\hat{t} = \arg\min_{1 \leq t \leq T} ||\boldsymbol{\mathcal{D}}_t||_F$, under the assumptions in Lemma 12, we have*

$$||\boldsymbol{\mathcal{D}}_{\check{t}}||_F^2 \leq \frac{1 + \delta_{\text{val}}}{1 - \delta_{\text{val}}} ||\boldsymbol{\mathcal{D}}_{\hat{t}}||_F^2 + \frac{2\delta_{\text{val}}}{1 - \delta_{\text{val}}} \sigma^2. \quad (109)$$

*Proof.* Under the assumptions of Lemma 12, we have

$$(1 - \delta_{\text{val}})(||\boldsymbol{\mathcal{D}}_t||_F^2 + \sigma^2) \leq \frac{1}{m_{\text{val}}} ||\mathfrak{M}_{\text{val}}(\boldsymbol{\mathcal{D}}_t) + \mathbf{e}||_F^2 \leq (1 + \delta_{\text{val}})(||\boldsymbol{\mathcal{D}}_t||_F^2 + \sigma^2), \forall t = 1, ..., T, \quad (110)$$

from the result of Lemma 12. Then we have

$$
\begin{aligned}
||\boldsymbol{\mathcal{D}}_{\hat{t}}||_F^2 + \sigma^2 &\leq \frac{1}{m_{\text{val}}(1 - \delta_{\text{val}})} ||\mathfrak{M}_{\text{val}}(\boldsymbol{\mathcal{D}}_{\hat{t}}) + \mathbf{e}||_F^2 \\
&\leq \frac{1}{m_{\text{val}}(1 - \delta_{\text{val}})} ||\mathfrak{M}_{\text{val}}(\boldsymbol{\mathcal{D}}_{\hat{t}}) + \mathbf{e}||_F^2 \leq \frac{1 + \delta_{\text{val}}}{1 - \delta_{\text{val}}} (||\boldsymbol{\mathcal{D}}_{\hat{t}}||_F^2 + \sigma^2),
\end{aligned}
\quad (111)
$$

which indicates

$$||\boldsymbol{\mathcal{D}}_{\hat{t}}||_F^2 \leq \frac{1 + \delta_{\text{val}}}{1 - \delta_{\text{val}}} ||\boldsymbol{\mathcal{D}}_{\hat{t}}||_F^2 + \frac{2\delta_{\text{val}}}{1 - \delta_{\text{val}}} \sigma^2. \quad (112)$$

Therefore, we complete the proof of Lemma 13. □

With these two Lemmas, together with Theorem 2, we proceed to prove Theorem 4. Replacing the result of Lemma 13 with $\boldsymbol{\mathcal{D}}_{\check{t}} = \boldsymbol{\mathcal{U}}_{\check{t}} * \boldsymbol{\mathcal{U}}_{\check{t}}^\top - \boldsymbol{\mathcal{X}}_\star$ and $\boldsymbol{\mathcal{D}}_{\hat{t}} = \boldsymbol{\mathcal{U}}_{\hat{t}} * \boldsymbol{\mathcal{U}}_{\hat{t}}^\top - \boldsymbol{\mathcal{X}}_\star$, we have

$$||\boldsymbol{\mathcal{U}}_{\check{t}} * \boldsymbol{\mathcal{U}}_{\check{t}}^\top - \boldsymbol{\mathcal{X}}_\star||_F^2 \leq \frac{1 + \delta_{\text{val}}}{1 - \delta_{\text{val}}} ||\boldsymbol{\mathcal{U}}_{\hat{t}} * \boldsymbol{\mathcal{U}}_{\hat{t}}^\top - \boldsymbol{\mathcal{X}}_\star||_F^2 + \frac{2\delta_{\text{val}}}{1 - \delta \text{val}} \sigma^2. \quad (113)$$

To achieve the error $C \frac{nkr\sigma^2\kappa^4}{m}$, we need the bound $\frac{2\delta}{1-\delta}\sigma^2$, which requires $\delta \leq \frac{nkr\kappa^4}{3m_{\text{train}}}$. Taking $\delta = \frac{nkr\kappa^4}{3m_{\text{train}}}$, then we can verify that $\frac{1 + \delta_{\text{val}}}{1 - \delta_{\text{val}}} \leq 2$ :

$$
\begin{aligned}
\frac{1 + \delta_{\text{val}}}{1 - \delta_{\text{val}}} = \frac{1 + \frac{nkr\kappa^4}{3m_{\text{train}}}}{1 - \frac{nkr\kappa^4}{3m_{\text{train}}}} = \frac{3m_{\text{train}} + nkr\kappa^4}{3m_{\text{train}} - nkr\kappa^4} &= 1 + \frac{2nkr\kappa^4}{3m_{\text{train}} - nkr\kappa^4} \\
&\overset{(a)}{\leq} 1 + \frac{2nkr\kappa^4}{nkr\kappa^4(3r\kappa^4 - 1)} \leq 2,
\end{aligned}
\quad (114)
$$

where $(a)$ uses the assumptions that $m \gtrsim nkr^2\kappa^8$. Therefore, combining the results of Theorem 2, we have

$$||\boldsymbol{\mathcal{U}}_{\hat{t}} * \boldsymbol{\mathcal{U}}_{\hat{t}}^\top - \boldsymbol{\mathcal{X}}_\star||_F^2 \leq C \frac{nkr\sigma^2\kappa^4}{m_{\text{train}}}.$$

Moreover, combining $\frac{nkr\kappa^4}{3m_{\text{train}}}$ with the assumption $m_{\text{val}} \geq C_1 \frac{\log T}{\delta_{\text{val}}^2}$, we have $m_{\text{val}} \geq C_1 \frac{m_{\text{train}}^2 \log T}{(rnk\kappa^4)^2}$.

Therefore, the proof of Theorem 4 is completed.

## H   TECHNIQUE LEMMAS

**Lemma 14.** *Suppose the linear map* $\mathfrak{M} : \mathbb{R}^{n \times n \times k} \to \mathbb{R}^m$ *satisfies* $(r+1, \delta_1)$ *t-RIP with* $\delta_1 \in (0, 1)$, *then* $\mathfrak{M}$ *also satisfies* $(r, \sqrt{kr}\delta_1)$ *S2S-t-RIP.*

*Proof.* The proof of this lemma can be adapted from that of [(Karnik et al., 2025), Lemma G.2] by introducing the inequalities $||\boldsymbol{\mathcal{Z}}||_F \leq \sqrt{r}||\boldsymbol{\mathcal{Z}}||$ and $||\boldsymbol{\mathcal{Z}}||_F = \sqrt{1/k}||\overline{\boldsymbol{\mathcal{Z}}}||_F$.  $\square$

**Lemma 15.** *Suppose the linear map* $\mathfrak{M} : \mathbb{R}^{n \times n \times k} \to \mathbb{R}^m$ *satisfies* $(2, \delta_2)$ *t-RIP with* $\delta_2 \in (0, 1)$, *then* $\mathfrak{M}$ *also satisfies* $\sqrt{k}\delta_2$-*S2N-t-RIP.*

*Proof.* The proof of this lemma can be adapted from that of [(Karnik et al., 2025), Lemma G.3] by introducing the inequalities $||\boldsymbol{\mathcal{Z}}||_F \leq \sqrt{r}||\boldsymbol{\mathcal{Z}}||$ and $||\boldsymbol{\mathcal{Z}}||_F = \sqrt{1/k}||\overline{\boldsymbol{\mathcal{Z}}}||_F$.  $\square$

**Lemma 16.** *For a tensor* $\boldsymbol{\mathcal{Y}} \in \mathbb{R}^{n \times n \times k}$ *with tubal-rank* $r$, *then we have*

$$||\boldsymbol{\mathcal{X}}|| \leq \sqrt{n_3}||\boldsymbol{\mathcal{X}}||_F, \ ||\boldsymbol{\mathcal{X}}||_F \leq \sqrt{r}||\boldsymbol{\mathcal{X}}||, \ \ ||\boldsymbol{\mathcal{X}}||_* \leq r||\boldsymbol{\mathcal{X}}||. \tag{115}$$

## I   EXTENSION TO THE GENERAL TENSOR

In this section, we provide a brief analysis for the extension to the asymmetric case by formulating the asymmetric model into a symmetric model. We first present the asymmetric tensor sensing model:

$$\mathbf{y} = \mathfrak{M}_a(\boldsymbol{\mathcal{X}}_\star) + \mathbf{s}, \tag{116}$$

where $\boldsymbol{\mathcal{X}}_\star \in \mathbb{R}^{n_1 \times n_2 \times k}$, $\mathfrak{M}_a(\boldsymbol{\mathcal{X}}) = [\langle \boldsymbol{\mathcal{B}}_1, \boldsymbol{\mathcal{X}}_\star \rangle, \langle \boldsymbol{\mathcal{B}}_2, \boldsymbol{\mathcal{X}}_\star \rangle, ..., \langle \boldsymbol{\mathcal{B}}_i, \boldsymbol{\mathcal{X}}_\star \rangle]$. Under this asymmetric model, we take an asymmetric factorization $\boldsymbol{\mathcal{X}} = \boldsymbol{\mathcal{L}} * \boldsymbol{\mathcal{R}}^\top$, $\boldsymbol{\mathcal{L}} \in \mathbb{R}^{n_1 \times r \times k}$, $\boldsymbol{\mathcal{R}} \in \mathbb{R}^{n_2 \times r \times k}$. Then we define the symmetric measurement tensors $\boldsymbol{\mathcal{C}}_i \in \mathbb{R}^{(n_1+n_2) \times (n_1+n_2) \times k}$ by:

$$\boldsymbol{\mathcal{C}}_i := \frac{1}{\sqrt{2}} \begin{pmatrix} 0 & \boldsymbol{\mathcal{B}}_i \\ \boldsymbol{\mathcal{B}}_i^\top & 0 \end{pmatrix} \tag{117}$$

and the corresponding linear map $\mathfrak{C} : \mathbb{R}^{(n_1+n_2) \times (n_1+n_2) \times k} \to \mathbb{R}^m$ via

$$(\mathfrak{C}(\boldsymbol{\mathcal{X}}))_i = \langle \boldsymbol{\mathcal{C}}_i, \boldsymbol{\mathcal{X}} \rangle.$$

Define

$$\mathrm{sym}(\boldsymbol{\mathcal{X}}) := \begin{bmatrix} 0 & \boldsymbol{\mathcal{X}} \\ \boldsymbol{\mathcal{X}}^\top & 0 \end{bmatrix}$$

and

$$\boldsymbol{\mathcal{Z}}_t := \frac{1}{\sqrt{2}} \begin{bmatrix} \boldsymbol{\mathcal{L}}_t \\ \boldsymbol{\mathcal{R}}_t \end{bmatrix} \text{ and } \tilde{\boldsymbol{\mathcal{Z}}}_t := \frac{1}{\sqrt{2}} \begin{bmatrix} \boldsymbol{\mathcal{L}}_t \\ -\boldsymbol{\mathcal{R}}_t \end{bmatrix}.$$

With these definitions, we then transfer the asymmetric sensing model into a symmetric model:

$$\frac{1}{\sqrt{2}}\mathfrak{C}(\mathrm{sym}(\boldsymbol{\mathcal{X}})) = \mathfrak{M}_a(\boldsymbol{\mathcal{X}}) \text{ and } \frac{1}{\sqrt{2}}\mathfrak{C}(\boldsymbol{\mathcal{Z}}_t * \boldsymbol{\mathcal{Z}}_t^\top - \tilde{\boldsymbol{\mathcal{Z}}}_t * \tilde{\boldsymbol{\mathcal{Z}}}_t^\top) = \mathfrak{M}_a(\boldsymbol{\mathcal{L}}_t * \boldsymbol{\mathcal{R}}_t^\top).$$

With this model, we have the following objective function:

$$h(\boldsymbol{\mathcal{L}}_t, \boldsymbol{\mathcal{R}}_t) = \frac{1}{2}||\mathfrak{M}(\boldsymbol{\mathcal{X}}_\star - \boldsymbol{\mathcal{L}}_t * \boldsymbol{\mathcal{R}}_t^\top) + s||^2 \tag{118}$$

Then we define the corresponding symmetric loss function:

$$h_{\mathrm{sym}}(\boldsymbol{\mathcal{Z}}_t, \tilde{\boldsymbol{\mathcal{Z}}}_t) = \frac{1}{4}||\mathfrak{C}(\mathrm{sym}(\boldsymbol{\mathcal{X}}_\star) - \boldsymbol{\mathcal{Z}}_t * \boldsymbol{\mathcal{Z}}_t^\top + \tilde{\boldsymbol{\mathcal{Z}}}_t * \tilde{\boldsymbol{\mathcal{Z}}}_t^\top) + \sqrt{2}s||^2.$$

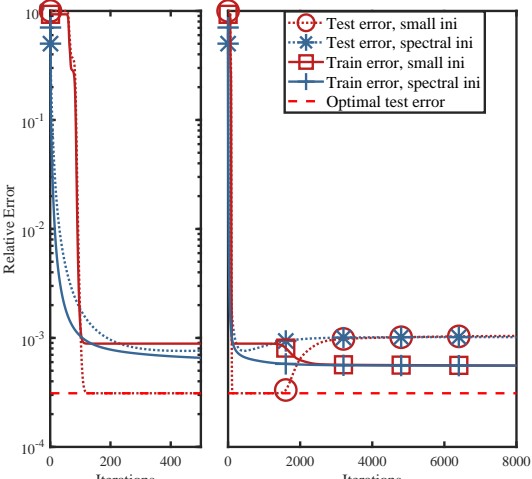

Figure 6: Comparison of training and testing errors for Problem (118) using FGD with spectral vs. small initialization. The ground-truth tensor has tubal-rank $r = 2$, overestimated rank $R = 4$, size $n_1 = n_2 = 20$, $k = 3$, $m = 5kr(2n_1 - r)$ measurements, and noise $\sigma = 10^{-3}$. Spectral initialization follows Liu et al. (2024b), while small initialization uses a near-zero starting point. Training error is $\frac{1}{2}||\boldsymbol{y} - \mathfrak{M}(\mathcal{L} * \mathcal{R}^\top)||^2$, and testing error is $||\mathcal{L} * \mathcal{R}^\top - \mathcal{X}_\star||_F^2/||\mathcal{X}_\star||_F^2$. "Baseline" denotes recovery under exact rank $R = r$. Insets show early (first 500 iterations) vs. full error curves.

The gradient update of $h_{\text{sym}}(\boldsymbol{\mathcal{Z}}_t, \tilde{\boldsymbol{\mathcal{Z}}}_t)$ is

$$
\begin{aligned}
\boldsymbol{\mathcal{Z}}_{t+1} &= \boldsymbol{\mathcal{Z}}_t + \eta[(\mathfrak{C}^*\mathfrak{C})(\text{sym}(\mathcal{X}_\star) - \boldsymbol{\mathcal{Z}}_t * \boldsymbol{\mathcal{Z}}_t^\top + \tilde{\boldsymbol{\mathcal{Z}}}_t * \tilde{\boldsymbol{\mathcal{Z}}}_t) + \mathfrak{C}^*(\sqrt{2}\boldsymbol{s})] * \boldsymbol{\mathcal{Z}}_t \\
\tilde{\boldsymbol{\mathcal{Z}}}_{t+1} &= \tilde{\boldsymbol{\mathcal{Z}}}_t - \eta[(\mathfrak{C}^*\mathfrak{C})(\text{sym}(\mathcal{X}_\star) - \boldsymbol{\mathcal{Z}}_t * \boldsymbol{\mathcal{Z}}_t^\top + \tilde{\boldsymbol{\mathcal{Z}}}_t * \tilde{\boldsymbol{\mathcal{Z}}}_t) + \mathfrak{C}^*(\sqrt{2}\boldsymbol{s})] * \tilde{\boldsymbol{\mathcal{Z}}}_t.
\end{aligned}
\tag{119}
$$

This formulation allows us to leverage some proof techniques from the symmetric case. However, handling the imbalance introduced by the two factor tensors poses a significant challenge, and we are actively investigating this issue. Encouragingly, our experimental result in Figure 6 shows that the phenomenon described in this paper also persists in the asymmetric setting.

## J  ADDITIONAL EXPERIMENTS

### J.1  SIMULATIONS ON DIFFERENT NOISE DISTRIBUTION

We conduct simulation experiments to verify that our theoretical results remain valid under various noise distributions, not limited to Gaussian noise. The experimental setup is identical to that in Section 6, except that we replace the Gaussian noise with two types of sub-exponential noise: Laplace noise and exponential noise. We briefly introduce the two noise models considered in our experiments:

- Laplace noise: The noise vector follows a Laplace distribution,

$$
\boldsymbol{s} \sim \text{Laplace}(\mu, b), \quad f(s_i) = \frac{1}{2b} \exp\left(-\frac{|s_i - \mu|}{b}\right),
$$

which is a symmetric sub-exponential distribution with mean $\mu$ and variance $2b^2$.

- Exponential noise: The noise vector follows an exponential distribution,

$$
\boldsymbol{s} \sim \text{Exp}(\lambda), \quad f(s_i) = \lambda \exp(-\lambda s_i), \quad x\boldsymbol{s}_i \geq 0,
$$

which is an asymmetric sub-exponential distribution with mean $1/\lambda$ and variance $1/\lambda^2$.

The results, shown in Figures 7 and 8, demonstrate that under both noise types, FGD with small initialization achieves the same recovery error as in the exact tubal-rank case, even in the overparameterized regime. This confirms that the guarantee provided by Theorem 2 extends beyond

Gaussian distributions. Moreover, FGD with validation and early stopping yields errors that are very close to those in the exact tubal-rank setting, further validating the effectiveness of this approach and suggesting that the result in Theorem 3 can also be extended to sub-exponential noise.

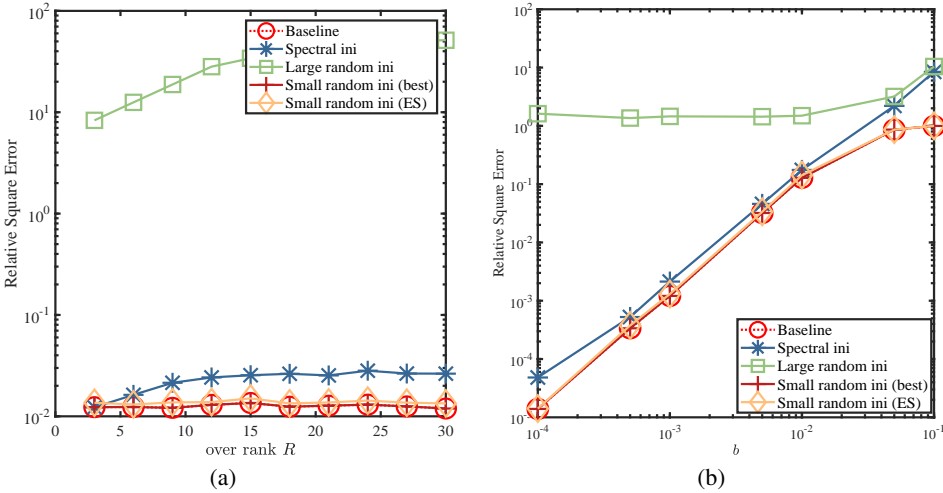

(a)             (b)

Figure 7: Performance comparison under varying $R, b$ with Laplace noise with $\mu = 0$. Subfigure (a) illustrates the recovery error of all methods under different over-rank values $R$, with parameters set as $m = 5kr(2n - r)$, $n = 30$, $b = 10^{-3}$, $\eta = 0.1$, and $T = 5000$. Subfigure (b) illustrates the error under varying noise levels $b$, with $m = 5kr(2n - r)$, $n = 30$, $R = 3r$, $\eta = 0.1$, and $T = 5000$.

## J.2    REAL-DATA EXPERIMENTS

In this section, we provide additional experimental details and results. We first present the algorithm used for the tensor completion task, as shown in Algorithm 3. We then give the definitions of the evaluation metrics, PSNR and relative error:

$$\text{PSNR} = 10 \log_{10} \left( \frac{\|\boldsymbol{\mathcal{X}}_\star\|_\infty^2}{\frac{1}{n_1 n_2 n_3}\|\hat{\boldsymbol{\mathcal{X}}} - \boldsymbol{\mathcal{X}}_\star\|_F^2} \right), \text{RE} = \frac{\|\hat{\boldsymbol{\mathcal{X}}} - \boldsymbol{\mathcal{X}}_\star\|_F}{\|\boldsymbol{\mathcal{X}}_\star\|\|_F},$$

where $\boldsymbol{\mathcal{X}}_\star$ is the ground truth and $\hat{\boldsymbol{\mathcal{X}}}$ is the estimated tensor. Next, we briefly introduce the baseline methods used for comparison:

- TNN (Lu et al., 2018): a classical convex method based on tubal tensor nuclear norm minimization proposed by (), widely used in tensor completion.

- TCTF (Zhou et al., 2017): a tensor factorization–based method with tubal-rank estimation, designed to reduce computational cost.

- UTF (Du et al., 2021): another tensor factorization method that replaces the tubal tensor nuclear norm constraint with Frobenius-norm constraints on two factor tensors.

- TC-RE (Shi et al., 2021): a rank-estimation–based method that first estimates the tubal rank and then performs tensor completion using truncated t-SVD.

- GTNN-HOP$_p$ (Wang et al., 2024): a method that replaces the traditional TNN soft thresholding with a hybrid ordinary-$l_p$ penalty for improved performance.

We conduct experiments on both color image completion and video completion tasks, and compare our method with the above approaches.

### J.2.1    COLOR IMAGE COMPLETION EXPERIMENTS

We perform color image completion experiments on the Berkeley Segmentation Dataset (Martin et al., 2001). We randomly select 50 color images of size $481 \times 321 \times 3$ and set the sampling

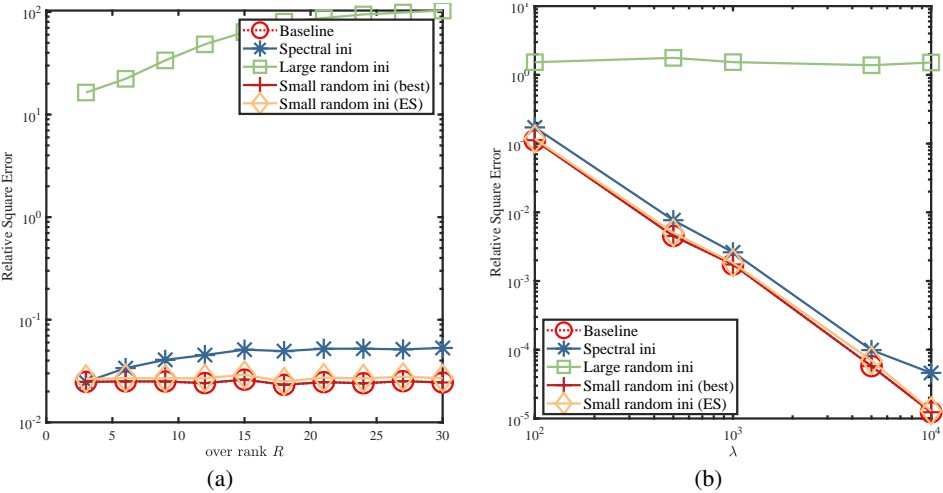

Figure 8: Performance comparison under varying $R$, $\lambda$ with exponential noise. Subfigure (a) illustrates the recovery error of all methods under different over-rank values $R$, with parameters set as $m = 5kr(2n - r)$, $n = 30$, $b = 10^{-3}$, $\eta = 0.1$, $\lambda = 1000$ and $T = 5000$. Subfigure (b) illustrates the error under varying noise levels $\lambda$, with $m = 5kr(2n - r)$, $n = 30$, $R = 3r$, $\eta = 0.1$, and $T = 5000$.

---

**Algorithm 3** Solving tensor completion by FGD with early stopping

---

**Input:** Train data $\mathfrak{P}_{\Omega}^{\text{train}}(\boldsymbol{\mathcal{X}}_\star + \boldsymbol{\mathcal{S}}_n)$, validation data $\mathfrak{P}_{\Omega}^{\text{val}}(\boldsymbol{\mathcal{X}}_\star + \boldsymbol{\mathcal{S}}_n)$, initialization scale $\alpha$, step size $\eta$, estimated tubal-rank $R$, iteration number T

**Initialization:** Initialize $\boldsymbol{\mathcal{L}}_0, \boldsymbol{\mathcal{R}}_0$, where each entry of $\boldsymbol{\mathcal{L}}_0, \boldsymbol{\mathcal{R}}_0$ are i.i.d. from $\mathcal{N}(0, \frac{\alpha^2}{R})$.

1: **for** $t = 0$ to $T - 1$ **do**

2:     $\boldsymbol{\mathcal{L}}_{t+1} = \boldsymbol{\mathcal{L}}_t - \frac{\eta}{p}\mathfrak{P}_{\Omega}^{\text{train}}(\boldsymbol{\mathcal{L}}_t * \boldsymbol{\mathcal{R}}_t^\top - \boldsymbol{\mathcal{X}}_\star - \boldsymbol{\mathcal{S}}_n) * \boldsymbol{\mathcal{R}}_t$

3:     $\boldsymbol{\mathcal{R}}_{t+1} = \boldsymbol{\mathcal{R}}_t - \frac{\eta}{p}\mathfrak{P}_{\Omega}^{\text{train}}(\boldsymbol{\mathcal{L}}_t * \boldsymbol{\mathcal{R}}_t^\top - \boldsymbol{\mathcal{X}}_\star - \boldsymbol{\mathcal{S}}_n)^\top * \boldsymbol{\mathcal{L}}_t$

4:     Validation loss: $e_t = \frac{1}{2p}\left\|\mathfrak{P}_{\Omega}^{\text{val}}(\boldsymbol{\mathcal{L}}_t * \boldsymbol{\mathcal{R}}_t^\top - \boldsymbol{\mathcal{X}}_\star - \boldsymbol{\mathcal{S}}_n)\right\|_F^2$

5: **end for**

6: **Output:** $\boldsymbol{\mathcal{L}}_{\check{t}} * \boldsymbol{\mathcal{R}}_{\check{t}}^\top$ where $\check{t} = \arg\min_{1 \leq t \leq T} e_t$.

---

Table 3: Comparison of different methods in terms of average Peak Signal-to-Noise Ratio (PSNR) and average Relative Error (RE) under various sampling rates and noise levels. A higher PSNR and smaller RE indicates better reconstruction quality."FGD-ES" denotes FGD with early stopping, while "FGD-best" refers to the minimum error achieved by FGD over all iterations.

| Methods | $p = 0.3$ | | | | $p = 0.4$ | | | |
| | $\sigma = 0.03$ | | $\sigma = 0.05$ | | $\sigma = 0.03$ | | $\sigma = 0.05$ | |
| | PSNR ↑ | RSE ↓ | PSNR ↑ | RSE ↓ | PSNR ↑ | RSE ↓ | PSNR ↑ | RSE ↓ |
|---|---|---|---|---|---|---|---|---|
| UTF | 7.8242 | 0.276 | 7.3535 | 0.2884 | 6.8286 | 0.3112 | 5.5376 | 0.3559 |
| TNN | 20.211 | 0.0659 | 17.2288 | 0.0934 | 20.4965 | 0.0639 | 17.1281 | 0.0947 |
| TC-RE | 19.7102 | 0.0698 | 16.9971 | 0.096 | 19.7435 | 0.0698 | 16.4039 | 0.1029 |
| GTNN-HOP$_{0.3}$ | 19.6553 | 0.0706 | 16.1069 | 0.107 | 20.0091 | 0.068 | 16.268 | 0.1051 |
| GTNN-HOP$_{0.6}$ | 20.2583 | 0.0659 | 16.8056 | 0.0987 | 20.5764 | 0.0636 | 16.8933 | 0.0978 |
| FGD-ES | 22.083 | 0.0529 | 21.02 | 0.0597 | 22.2876 | 0.0517 | 21.5831 | 0.0559 |
| FGD-best | **22.1411** | **0.0525** | **21.1517** | **0.0588** | **22.3001** | **0.0516** | **21.7213** | **0.0551** |

rate as $p$ and add Gaussian noise $\mathcal{N}(0, \sigma^2)$. For TNN, UTF, TCTF, and GTNN-HOP, we adopt the initialization schemes and hyperparameter settings as described in their original papers. The entry "FGD-best" refers to the highest PSNR obtained by FGD with small initialization, while "FGD-ES" corresponds to the PSNR achieved using early stopping based on validation. For both settings, the initialization scale is set to $\alpha = 10^{-5}$ and the step size is set to $\eta = 1e - 3$. The tubal-ranks of FGD-ES, FGD-best and UTF are set to 100 for all images. The max iteration number is 2000. We present in Figures 9-12 the PSNR and RE values of different methods on each image under various model parameters $((p, \sigma))$. In addition, Figure 13 shows the visual reconstruction results.

We observe that FGD-best and FGD-ES achieve the best recovery performance in most cases. Moreover, when the noise level increases, the performance of other algorithms degrades significantly, whereas FGD with small initialization is much less affected, highlighting the benefit of small initialization.

### J.2.2 VIDEO COMPLETION EXPERIMENTS

Beyond image completion, we also performed video completion experiments with Gaussian noise. We randomly selected four videos from the YUV Video Sequences dataset [2], extracted the first 30 frames of each to form tensors of size $176 \times 144 \times 30$, added Gaussian noise drawn from $\mathcal{N}(0, \sigma^2)$, and again applied sampling rate $p$. Since TCTF performs bad in the low sampling rate case of video completion, we replace it with GTNN-HOP$_{0.6}$, a non-convex method with a sparsity-inducing regularizer. For FGD-ES and FGD-best, the initialization scale is set to $\alpha = 10^{-5}$ and the step size is set to $\eta = 2e-4$. The tubal-ranks of FGD-ES, FGD-best and UTF are set to 50 for all images. The max iteration number is 4000. Tables 3-7 report the PSNR and RE values of all methods on the four videos, and Figure 14 shows the reconstruction results of the first frame of the akiyo video for each method. As can be seen, our method achieves the smallest relative recovery error and the highest PSNR values. In addition, we evaluated the robustness of FGD-best and FGD-ES with respect to the choice of the tubal rank $R$. The results, shown in the Figure 15, demonstrate that both methods are highly robust to the selection of $R$ across all four videos.

One potential issue is that gradient-based methods are sensitive to the condition number of the underlying matrix or tensor, leading to slower convergence when the condition number is large. Thus, developing methods that accelerate FGD while controlling the amplification of noise remains an interesting direction for future research.

---

[2]https://www.cnets.io/traces.cnets.io/trace.eas.asu.edu/yuv/index.html

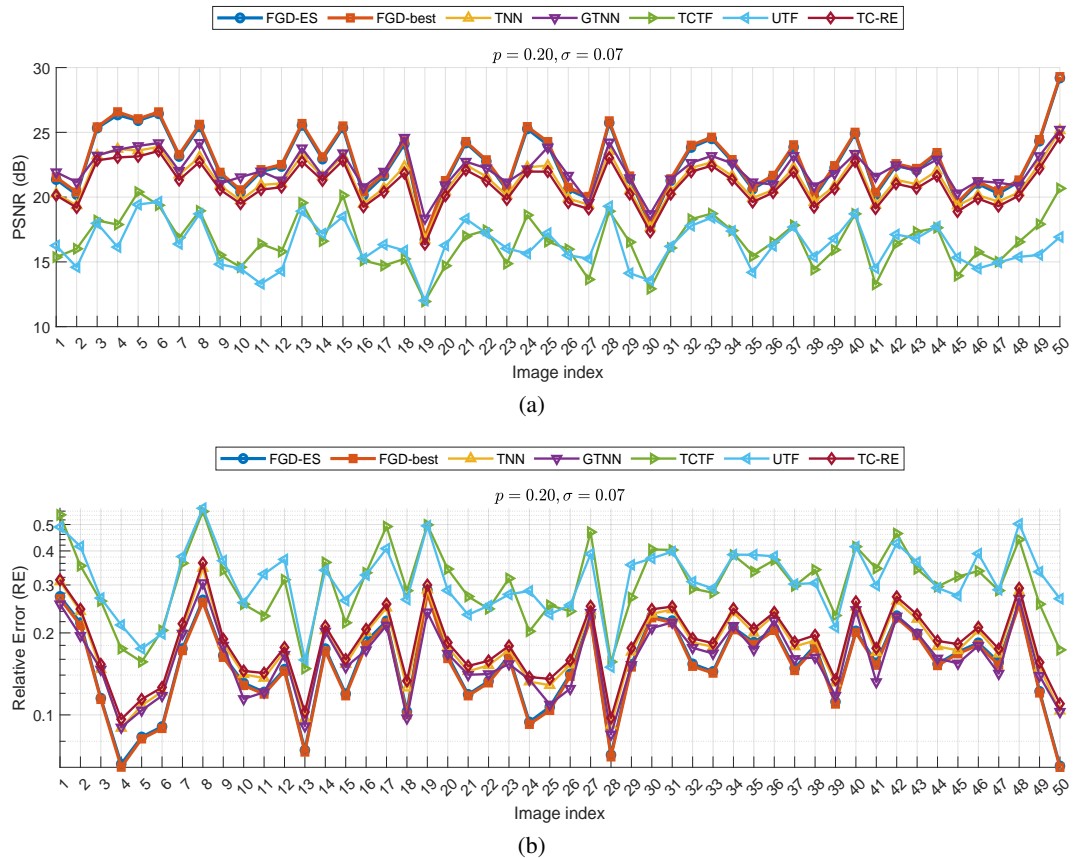

Figure 9: Comparison of PSNR values and Relative Error across 50 images for different methods, with sampling rate $p = 0.2$ and noise standard deviation $\sigma = 0.07$.

Table 4: Comparison of different methods in terms of average Peak Signal-to-Noise Ratio (PSNR) and average Relative Error (RE) under various sampling rates and noise levels. A higher PSNR and smaller RE indicates better reconstruction quality. "FGD-ES" denotes FGD with early stopping, while "FGD-best" refers to the minimum error achieved by FGD over all iterations.

| Methods | $p = 0.3$ | | | | $p = 0.4$ | | | |
|---|---|---|---|---|---|---|---|---|
| | $\sigma = 0.03$ | | $\sigma = 0.05$ | | $\sigma = 0.03$ | | $\sigma = 0.05$ | |
| | PSNR ↑ | RSE ↓ | PSNR ↑ | RSE ↓ | PSNR ↑ | RSE ↓ | PSNR ↑ | RSE ↓ |
| TNN | 20.7258 | 0.0613 | 17.4371 | 0.0895 | 21.0157 | 0.0592 | 17.3267 | 0.0906 |
| TC-RE | 19.928 | 0.0671 | 17.1895 | 0.0920 | 20.2059 | 0.0650 | 16.5464 | 0.0991 |
| UTF | 6.3467 | 0.3207 | 6.4038 | 0.3186 | 5.9414 | 0.3360 | 4.7523 | 0.3853 |
| GTNN-HOP$_{0.3}$ | 19.9438 | 0.0670 | 16.1328 | 0.1039 | 20.2975 | 0.0643 | 16.2845 | 0.1021 |
| GTNN-HOP$_{0.6}$ | 20.5461 | 0.0625 | 16.8126 | 0.0961 | 20.8648 | 0.0603 | 16.9083 | 0.0951 |
| FGD-ES | 23.2899 | 0.0456 | 21.6321 | 0.0552 | 23.8244 | 0.0429 | 22.3928 | 0.0501 |
| FGD-best | **23.4511** | **0.0448** | **21.8002** | **0.0541** | **23.8539** | **0.0427** | **22.5611** | **0.0496** |

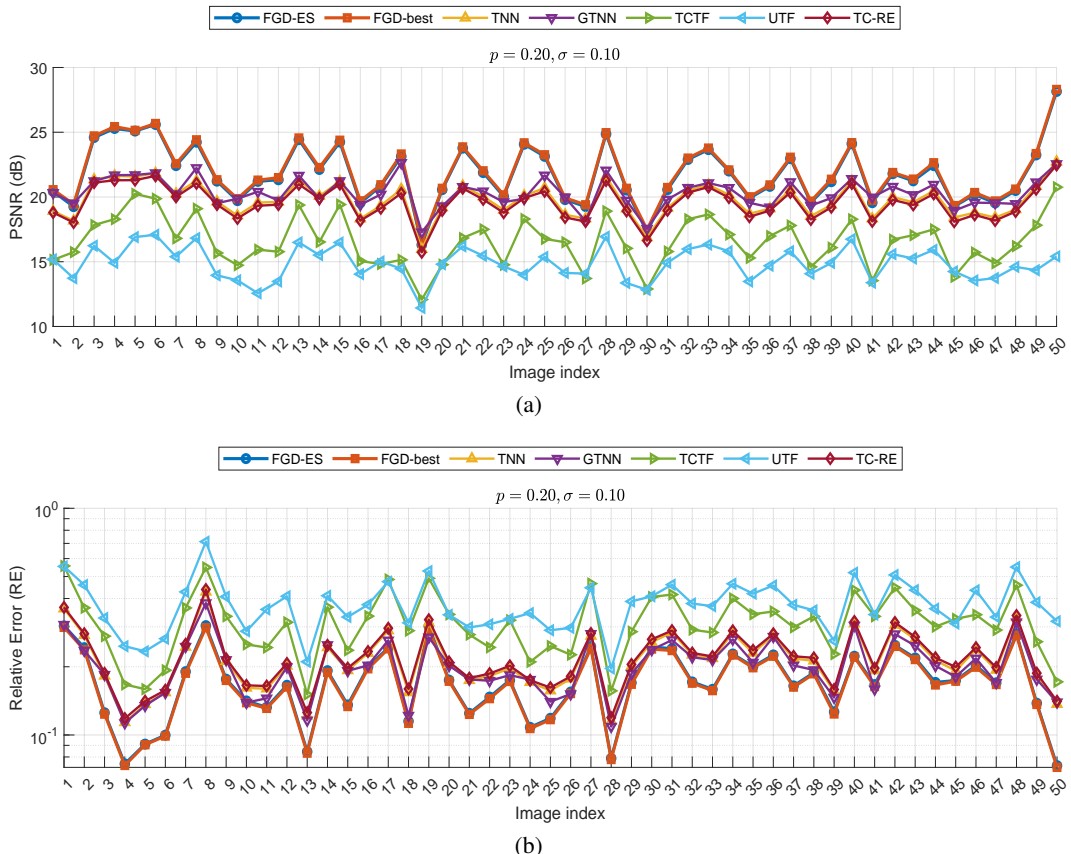

Figure 10: Comparison of PSNR values and Relative Error across 50 images for different methods, with sampling rate $p = 0.2$ and noise standard deviation $\sigma = 0.1$.

Table 5: Comparison of different methods in terms of average Peak Signal-to-Noise Ratio (PSNR) and average Relative Error (RE) under various sampling rates and noise levels for the "highway" video. A higher PSNR and smaller RE indicates better reconstruction quality. "FGD-ES" denotes FGD with early stopping, while "FGD-best" refers to the minimum error achieved by FGD over all iterations.

| Methods | $p = 0.3$ | | | | $p = 0.4$ | | | |
| | $\sigma = 0.03$ | | $\sigma = 0.05$ | | $\sigma = 0.03$ | | $\sigma = 0.05$ | |
| | PSNR ↑ | RSE ↓ | PSNR ↑ | RSE ↓ | PSNR ↑ | RSE ↓ | PSNR ↑ | RSE ↓ |
|---|---|---|---|---|---|---|---|---|
| TNN | 19.4802 | 0.0474 | 16.9433 | 0.0635 | 19.8369 | 0.0455 | 16.8423 | 0.0642 |
| TC-RE | 19.0227 | 0.0499 | 16.6549 | 0.0656 | 19.1568 | 0.0492 | 16.1336 | 0.0697 |
| UTF | 4.0060 | 0.2814 | 3.7117 | 0.2911 | 1.1904 | 0.3891 | 0.7714 | 0.4084 |
| GTNN-HOP$_{0.3}$ | 19.3115 | 0.0483 | 16.1687 | 0.0694 | 19.7269 | 0.0461 | 16.3390 | 0.0680 |
| GTNN-HOP$_{0.6}$ | 19.8894 | 0.0452 | 16.8541 | 0.0641 | 20.2802 | 0.0432 | 16.9568 | 0.0634 |
| FGD-ES | 20.6667 | 0.0413 | 20.0041 | 0.0446 | 20.7364 | 0.0410 | 20.3462 | 0.0429 |
| FGD-best | **20.7000** | **0.0412** | **20.1063** | **0.0441** | **20.7278** | **0.0410** | **20.4994** | **0.0421** |

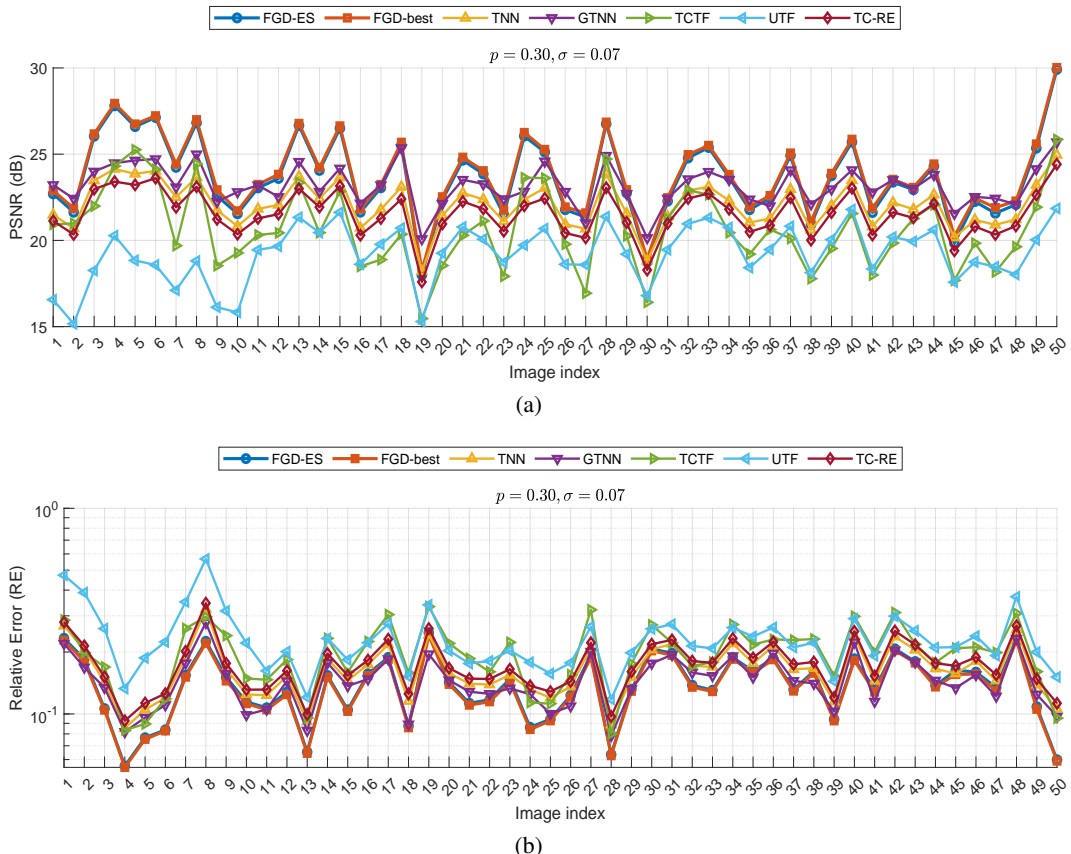

Figure 11: Comparison of PSNR values and Relative Error across 50 images for different methods, with sampling rate $p = 0.3$ and noise standard deviation $\sigma = 0.07$.

Table 6: Comparison of different methods in terms of average Peak Signal-to-Noise Ratio (PSNR) and average Relative Error (RE) under various sampling rates and noise levels for the "suzie" video. A higher PSNR and smaller RE indicates better reconstruction quality. "FGD-ES" denotes FGD with early stopping, while "FGD-best" refers to the minimum error achieved by FGD over all iterations.

| Methods | $p = 0.3$ | | | | $p = 0.4$ | | | |
|---|---|---|---|---|---|---|---|---|
| | $\sigma = 0.03$ | | $\sigma = 0.05$ | | $\sigma = 0.03$ | | $\sigma = 0.05$ | |
| | PSNR ↑ | RSE ↓ | PSNR ↑ | RSE ↓ | PSNR ↑ | RSE ↓ | PSNR ↑ | RSE ↓ |
| TNN | 19.3458 | 0.0717 | 16.6844 | 0.0974 | 19.8125 | 0.0679 | 16.7441 | 0.0967 |
| TC-RE | 19.2177 | 0.0728 | 16.5983 | 0.0984 | 19.2186 | 0.0728 | 16.0643 | 0.1046 |
| UTF | 11.4879 | 0.1772 | 10.2100 | 0.2053 | 10.8456 | 0.1908 | 8.6211 | 0.2465 |
| GTNN-HOP$_{0.3}$ | 19.0898 | 0.0738 | 15.7370 | 0.1086 | 19.5613 | 0.0699 | 15.9544 | 0.1059 |
| GTNN-HOP$_{0.6}$ | 19.6531 | 0.0692 | 16.4167 | 0.1005 | 20.0997 | 0.0657 | 16.5620 | 0.0988 |
| FGD-ES | 20.5670 | 0.0623 | 19.4874 | 0.0705 | 20.7175 | 0.0612 | 20.1540 | 0.0653 |
| FGD-best | **20.5947** | **0.0621** | **19.6263** | **0.0694** | **20.7445** | **0.0610** | **20.2629** | **0.0645** |

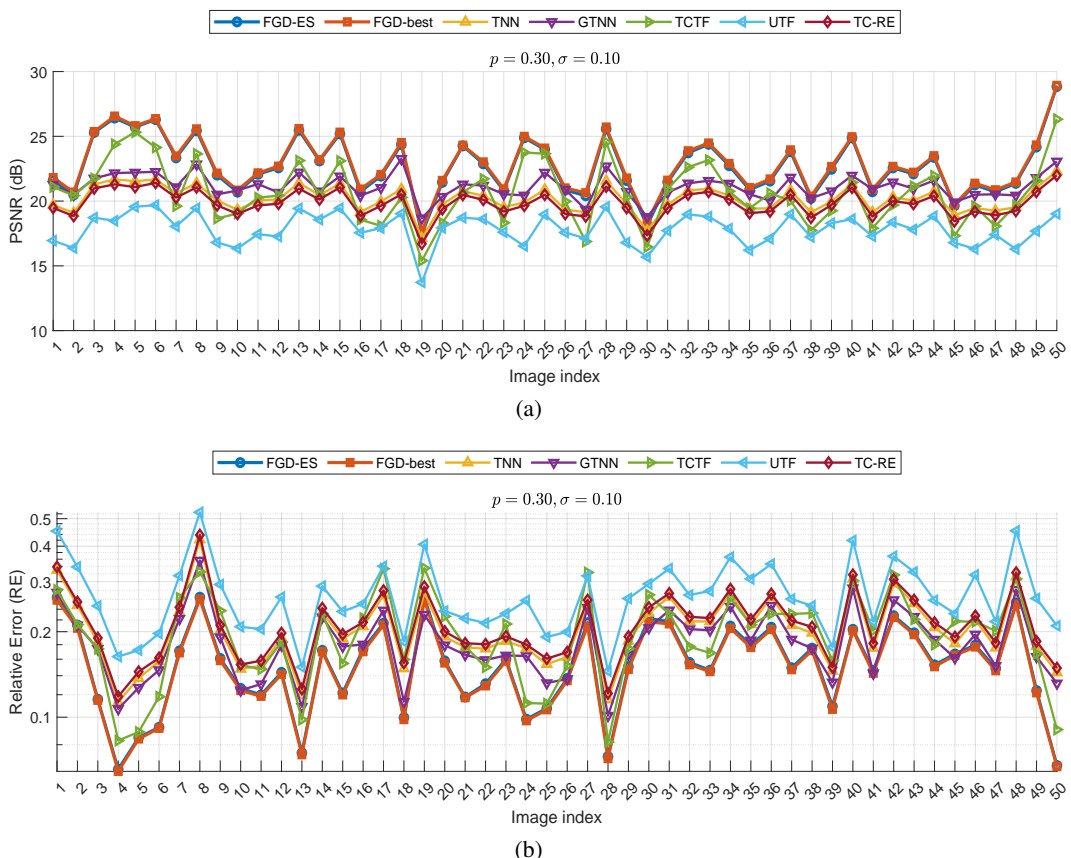

Figure 12: Comparison of PSNR values and Relative Error across 50 images for different methods, with sampling rate $p = 0.3$ and noise standard deviation $\sigma = 0.1$.

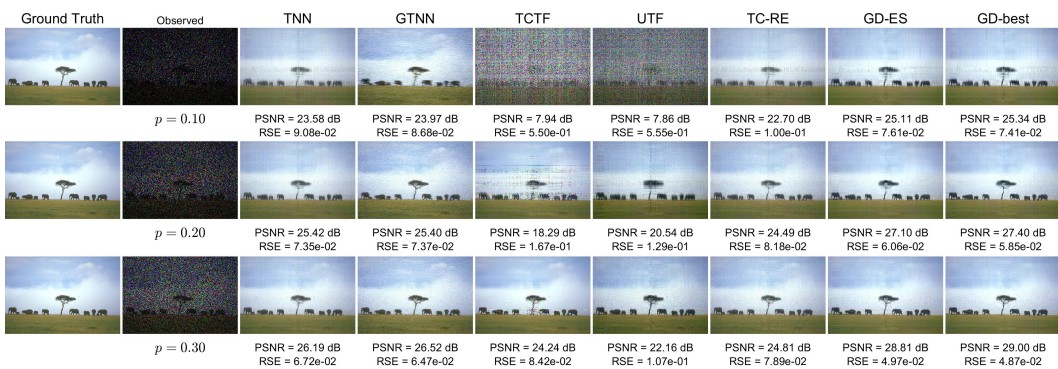

Figure 13: Comparison of the image recovery performance of different methods under varying sampling rate $p$. The noise standard deviation $\sigma = 0.05$. And 5% of the observed entries are used for validation.

Table 7: Comparison of different methods in terms of average Peak Signal-to-Noise Ratio (PSNR) and average Relative Error (RE) under various sampling rates and noise levels for the "miss-america" video. A higher PSNR and smaller RE indicates better reconstruction quality. "FGD-ES" denotes FGD with early stopping, while "FGD-best" refers to the minimum error achieved by FGD over all iterations.

| Methods | $p = 0.3$ | | | | $p = 0.4$ | | | |
|---|---|---|---|---|---|---|---|---|
| | $\sigma = 0.03$ | | $\sigma = 0.05$ | | $\sigma = 0.03$ | | $\sigma = 0.05$ | |
| | PSNR ↑ | RSE ↓ | PSNR ↑ | RSE ↓ | PSNR ↑ | RSE ↓ | PSNR ↑ | RSE ↓ |
| TNN | 21.2922 | 0.0831 | 17.8503 | 0.1235 | 21.3210 | 0.0828 | 17.5991 | 0.1271 |
| TC-RE | 20.6724 | 0.0892 | 17.5455 | 0.1279 | 20.3926 | 0.0921 | 16.8711 | 0.1382 |
| UTF | 9.4560 | 0.3245 | 9.0886 | 0.3386 | 9.3371 | 0.3290 | 8.0055 | 0.3835 |
| GTNN-HOP$_{0.3}$ | 20.2760 | 0.0934 | 16.3891 | 0.1461 | 20.4505 | 0.0915 | 16.4938 | 0.1443 |
| GTNN-HOP$_{0.3}$ | 20.9446 | 0.0865 | 17.1388 | 0.1340 | 21.0611 | 0.0853 | 17.1462 | 0.1339 |
| FGD-ES | 23.8085 | 0.0622 | 22.9563 | 0.0686 | 23.8721 | 0.0617 | 23.4395 | 0.0649 |
| FGD-best | **23.8186** | **0.0621** | **23.0738** | **0.0677** | **23.8743** | **0.0617** | **23.5618** | **0.0640** |

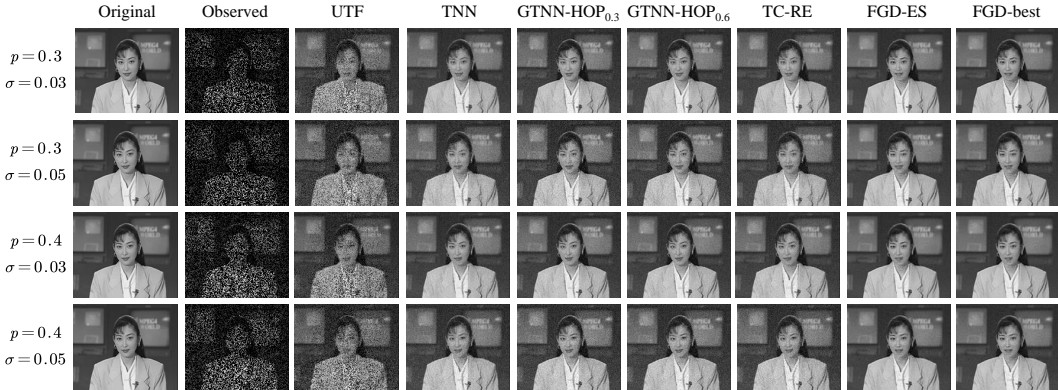

Figure 14: Comparison of the video recovery performance of different methods under varying sampling rate $p$ and noise standard deviation $\sigma$ for video "akiyo". For FGD-ES, 5% of the observed entries are used for validation.

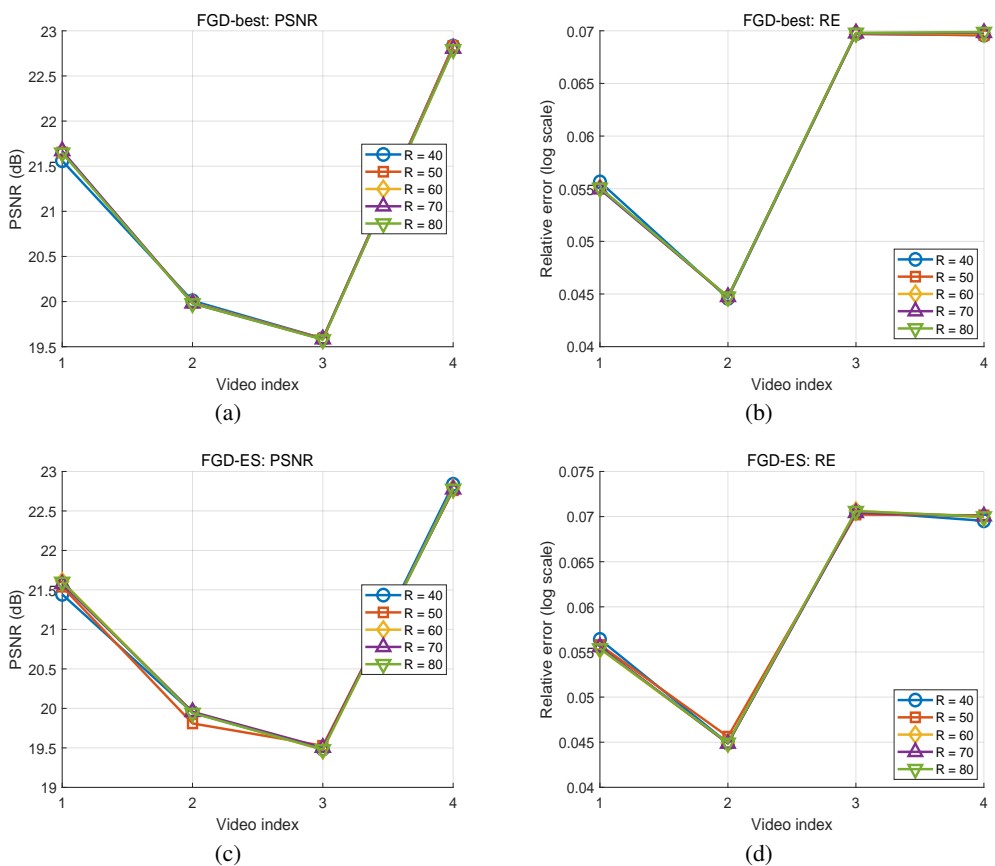

Figure 15: Evaluate the effect of different tubal-rank $R$ on the performance of video completion. Subfigure (a) shows the PSNR values of FGD-best on the four videos, and subfigure (b) shows the corresponding RE values. Subfigure (c) reports the PSNR values of FGD-ES on the four videos, while subfigure (d) presents the associated RE values.

