# OpenReview forum: "The Power of Small Initialization in Noisy Low-Tubal-Rank Tensor Recovery"
_ICLR.cc/2026/Conference — ICLR 2026 Poster_

### Official Review · Reviewer_SVNR · 2025-10-29

**Soundness:** 3
**Presentation:** 2
**Contribution:** 3
**Rating:** 6
**Confidence:** 3

**Summary:**

This paper investigates the problem of recovering a noisy low-tubal-rank tensor model. Specifically, it studies the theoretical guarantees of the gradient descent method with small initialization for solving a low-rank recovery problem formulated via the Burer–Monteiro factorization under the t-product framework. The authors prove that when the dimensions of the optimization variables match the ground-truth rank, the algorithm enjoys a linear convergence rate. They further provide error estimates under different levels of over-parameterization, showing that the final recovery error depends only on the noise scale as the iteration proceeds. Furthermore, the paper proposes a practical stopping criterion and establishes the corresponding recovery error bound under this termination condition. Numerical experiments demonstrate that the small-initialization gradient descent method achieves superior recovery performance compared with those initialized randomly or by spectral methods.

**Strengths:**

The paper is well structured and clearly written, particularly on the mathematical side. The main results are rigorously proved and thoughtfully discussed. The authors also provide useful comparisons with existing works, which helps position their contributions within the current literature. The topic—low-tubal-rank tensor recovery—is timely and important, as tensor models have recently attracted significant attention due to their complex structures and broad applications. Moreover, the analysis of over-parameterization is of great interest, and the finding that the recovery error depends only on the noise scale is both elegant and encouraging.

**Weaknesses:**

Although this paper studies the tensor case, the introduction and background on tensor are relatively limited, which reduces the readability for a broader audience. To improve accessibility, it would be helpful to include more definitions and explanations—such as the t-product and the corresponding notion of tensor rank—in the appendix, given the space limitations of the main text.

Another concern is that, as mentioned in Remark 7, the results of this paper can be viewed as a direct extension of the matrix case studied in [Ding et.al., 2025]. If one interprets the tensor in this paper as a matrix and the t-product as standard matrix multiplication, it is not immediately clear how the theoretical analysis differs from the matrix setting. The paper would benefit from a more explicit discussion highlighting the distinctive challenges or techniques specific to tensors, which would not only improve readability but also strengthen the contribution of the work.

Finally, Remark 7 might be placed earlier in the text, since both Theorem 2 and its proof sketch are quite similar to those in [Ding et.al., 2025], not only the proposed stopping strategy.

**Questions:**

- In the matrix case, the model obtained via the Burer–Monteiro (BM) factorization is nonconvex and may contain spurious local minima. For the tensor case studied in this paper, do similar spurious local minima exist? If so, what guarantees that the proposed gradient descent iterates will not converge to such undesirable local minima?

- In the proof sketch of Theorem 2, the authors mention that the iterates undergo four different stages, during which the singular values of the iterates exhibit distinct behaviors. It would be interesting and instructive to design numerical experiments that illustrate these four phases and verify their correspondence with the theoretical analysis.

---

> ### Author Response · Authors · 2025-11-21
> **Response to Reviewer SVNR (1/2)**
>
> We sincerely thank the reviewer for carefully evaluating our paper and providing many valuable and insightful comments. We are also pleased to hear the reviewer describe our results as “elegant and encouraging.” Below, we address the concerns.
>
> **Weakness 1: include more definitions and explanations**
>
> We appreciate the reviewer's suggestion. In the revised version, we have expanded **Appendix D** to include additional background and definitions, specifically the t-product, t-SVD, and the tubal-rank.
>
> **Weakness 2: highlight the distinctive challenges or techniques specific to tensors**
>
> Thank you for this suggestion; it is indeed very helpful for improving the quality of our paper. Following your advice, we have added a detailed discussion of the tensor-specific challenges to **Section 3.3** of the revised manuscript.
>
> Extending the matrix setting to the tensor setting is nontrivial, as one must address several challenges unique to tensors:
> 1. **Non-complementarity of range and kernel.**
>       In the matrix case, the range and the kernel are complementary subspaces. This property no longer holds for third-order tubal tensors. If the tensor contains non-invertible tubes in its t-SVD—equivalently, if some frequency slices vanish in the Fourier domain—then the range and kernel share common generators. As a result, the classical decomposition of gradient updates into a "signal term" and an "over-parameterization term" fails on these non-invertible tubes. This necessitates introducing a more refined notion of tensor condition number to track the identifiable and unidentifiable components separately.
>
> 2. **Inter-slice coupling induced by the measurement operator.** For the power method, each frequency slice of a tubal tensor behaves like an independent matrix power iteration, a known fact in the literature [1]. However, in gradient descent for tensor recovery, the measurement operator and its adjoint couple information across all frequency slices. Specifically, for the observation model ${y}=\mathfrak{{M}}(\mathcal{{X}}\_\star)+{s}\in\mathbb{R}^m,\ \mathcal{X}\_\star\in\mathbb{R}^{n\times n\times k}$, we have $$y\_i=\langle \mathcal{{A}}\_i, \mathcal{{X}}\_\star\rangle+s\_i=\frac{1}{k}\langle\overline{ \mathcal{{A}}\_i}, \overline{\mathcal{{X}}\_\star}\rangle+s\_i =\frac{1}{k}\sum_{q=1}^k\langle \overline{A}\_i^{(q)} , \overline{X\_\star}^{(q)}\rangle+s\_i,i=1,...,m.$$ Then we have $$\overline{\mathfrak{M}^*(\mathfrak{M}(\mathcal{X}\_\star))}^{(j)}=\sum_{i=1}^m\sum_{q=1}^k \overline{\mathcal{A}\_i}^{(j)}\langle \overline{A}\_i^{(q)} , \overline{X\_\star}^{(q)}\rangle.$$ Consequently, the update of any single slice depends on all other slices, making it impossible to analyze the slices independently, as in the power method.
>
> 3. **Minimax error for noisy low-tubal-rank tensor recovery.** In the matrix setting, Reference [2] has already established the minimax error for noisy matrix sensing. To the best of our knowledge, however, no such minimax error analysis exists for the tubal tensor setting. Therefore, we need to establish the minimax error bound for the noisy low-tubal-rank tensor recovery problem to show that our method matches the fundamental information-theoretic limit of the problem.
>
>
> [1] Gleich, D. F., Greif, C., & Varah, J. M. (2013). The power and Arnoldi methods in an algebra of circulants. _Numerical Linear Algebra with Applications_, 20(5), 809-831.
>
> [2] Candes, E. J., & Plan, Y. (2011). Tight oracle inequalities for low-rank matrix recovery from a minimal number of noisy random measurements. _IEEE Transactions on Information Theory_, 57(4), 2342-2359.

---

> ### Author Response · Authors · 2025-11-21
> **Response to Reviewer SVNR (2/2)**
>
> **Weakness 3: placed Remark 7 earlier in the text**
>
> Thank you for this helpful suggestion. In the revised version we have moved Remark 7 to **Section 3.3** and clarified the differences between our Theorem 2 and  [Ding et.al., 2025].
>
> **Question 1: spurious local minima in tensor case**
>
> We thank the reviewer for this insightful question. In the tensor setting, the BM factorization is also nonconvex, so spurious local minima may exist. At a high level, there are two complementary lines of analysis in nonconvex low rank recovery. One is a global landscape analysis, which aims to show that the objective has no spurious local minima. The other is a two-stage analysis, where one first assumes an appropriate initialization and then applies a local algorithm to obtain the solution. Our analysis follows the second approach. We do not claim a global landscape result for tensors under the t-SVD model, and we are not aware of prior work that establishes such a landscape. We guarantee that starting from a small random initialization, gradient descent achieves a recovery error close to the minimax rate, whereas these guarantees do not rule out the existence of spurious minima when initialized arbitrarily. By analogy with the matrix literature [3-5], we conjecture that under suitable sampling conditions, the low tubal rank tensor recovery problem enjoys a benign optimization landscape, in the sense that every local minimum is statistically close to the global solution. A rigorous landscape analysis for the t-SVD tensor model is an important and challenging direction for future work.
>
> [3] Zhu, Z., Li, Q., Tang, G., & Wakin, M. B. (2018). Global optimality in low-rank matrix optimization. _IEEE Transactions on Signal Processing_, 66(13), 3614-3628.
>
> [4] Li, Q., Zhu, Z., & Tang, G. (2019). The non-convex geometry of low-rank matrix optimization. _Information and Inference: A Journal of the IMA_, 8(1), 51-96.
>
> [5] Zhu, Z., Li, Q., Tang, G., & Wakin, M. B. (2021). The global optimization geometry of low-rank matrix optimization. _IEEE Transactions on Information Theory_, _67_(2), 1308-1331.
>
> **Question 2: design numerical experiments to illustrate the four phases in the proof sketch of Theorem 2**
>
> We thank the reviewer for this insightful suggestion. In the revised version, we have added a numerical experiment in the **Appendix E.2** to illustrate the four stages described in the proof sketch of Theorem 2 and included corresponding explanations that confirm their consistency with the theoretical analysis.

---

> > ### Comment · Reviewer_SVNR · 2025-11-24
> > **Response to Rebuttal**
> >
> > Dear Authors:
> >
> > Thank you for your detailed explanation, and for outlining some of the technical difficulties surrounding the use of tensors. I personally have also done research in the field of tensor optimization, so I could understand the intricacies involved.
> >
> > However, I still believe that the overall statements are too similar to the matrix case, for example the Stoger 2021 Neurips paper, and it is more or less a technical extension at core.
> >
> > Therefore I would still maintain my score of 6, which leans more towards accept given the non-trivial contribution in technicality.

---

> > > ### Author Response · Authors · 2025-11-24
> > >
> > > We sincerely thank the reviewer for the thoughtful follow-up. We agree that our results are close in spirit to related matrix based works. However, in the tensor setting there are many **non-trivial technical challenges** that do not arise in the matrix case. By overcoming these difficulties, we obtain, to the best of our knowledge, the first nearly minimax tubal tensor recovery error bound.
> > >
> > > We are grateful for the reviewer’s understanding and recognition of our technical contribution. If the reviewer has any further questions, we would be very happy to discuss them.

---

### Official Review · Reviewer_svxt · 2025-10-29

**Soundness:** 2
**Presentation:** 2
**Contribution:** 2
**Rating:** 2
**Confidence:** 4

**Summary:**

This paper investigates low-tubal-rank tensor recovery from noisy linear measurements using the t-product framework and analyzes the behavior of factorized gradient descent (FGD) under over-parameterization. The authors show that small initialization and early stopping can significantly reduce recovery error independent of the over-estimated tubal rank, with both theoretical guarantees and empirical validation.

**Strengths:**

The main contribution of this paper lies in providing an error bound for low-tubal-rank tensor recovery when the tubal rank is overestimated in noisy settings, showing that FGD can still achieve reliable recovery.

**Weaknesses:**

1. The contribution of this work is incremental. Factorized Gradient Descent (FGD) is not an original contribution of this work (see [1]). Similarly, the use of early stopping to avoid overfitting is a standard practice and cannot be regarded as an innovation here.
[1] Z. Liu, Z. Han, Y. Tang, X. -L. Zhao and Y. Wang, "Low-Tubal-Rank Tensor Recovery via Factorized Gradient Descent," in IEEE Transactions on Signal Processing, vol. 72, pp. 5470-5483, 2024

2. The relationship with [1] should be discussed in detail rather than mentioned only briefly.

3. Please provide a careful comparison with `Implicit Regularization for Tubal Tensors via GD' and A Validation Approach to Over-parameterized Matrix and Image Recovery'', particularly focusing on the technical tools employed in the proofs.

4. The main motivation of this work is that traditional algorithms often use a higher rank than the true rank. However, in most practical applications (e.g., image and video recovery), the true rank is unknown. What really matters is which estimated rank leads to better recovery performance—and in many cases, such rank estimates are not unique. A thorough discussion and empirical comparison in real-world scenarios are therefore necessary. At present, the paper includes only one example on a single color image and lacks broader discussion of practical implications.

5.Over the past five years, many tensor decomposition methods with rank estimation strategies have been proposed. Since this work is motivated by the problem of overestimated rank, it should include comparisons and discussion of rank-estimation-based approaches, such as the following:

[2] Q. Shi, Y.-M. Cheung, and J. Lou, “Robust tensor svd and recovery with rank estimation,” IEEE Transactions on Cybernetics, 2021.

[3] Zheng, J., Wang, W., Zhang, X., & Jiang, X. (2023). A Novel Tensor Factorization-Based Method with Robustness to Inaccurate Rank Estimation. arXiv preprint arXiv:2305.11458.

[4] Zhu Q, Wu S, Fang S, et al. Fast tensor robust principal component analysis with estimated multi-rank and Riemannian optimization[J]. Applied Intelligence, 2025, 55(1): 52.

6. Beyond Implicit Regularization for Tubal Tensors via GD, no other recent tensor decomposition methods (from the last five years) are compared.

**Questions:**

See the Weaknesses.

---

> ### Author Response · Authors · 2025-11-21
> **Response to Reviewer svxt (1/7)**
>
> We sincerely thank the reviewer for carefully evaluating our paper and providing valuable comments that help improve the quality of our work. Below, we address the concerns.
>
> **Weakness 1: the contribution is incremental, FGD and early stopping are not new.**
>
> We agree that FGD and early stopping are not new. However, we respectfully disagree with the assessment that our contributions are incremental. Our work makes **three substantive contributions**:
>
> 1.  **We are the first to prove that, under small initialization, the recovery error for the noisy and overparameterized low-tubal-rank tensor recovery problem depends _only_ on the true tubal rank and does _not_ grow with the overparameterized rank.** Moreover, when the noise is sub-Gaussian, we show that this recovery error is **minimax-optimal**. To the best of our knowledge, this is the **first minimax-optimal error bound** established for low-tubal-rank tensor recovery in the literature. A minimax optimal bound means that no other estimator can uniformly achieve a strictly smaller error order over the same model class, so our rates match the fundamental information theoretic limit of the problem. Such a result is clearly not incremental.
>
> 2.  **We further show that this minimax-optimal error can be achieved by a simple combination of small initialization and early stopping.** Unlike standard early-stopping heuristics, we provide **rigorous theoretical justification**, proving that once the sample size of validation set reaches a certain scale, the stopped iterate is guaranteed to be nearly optimal. This theoretical guarantee for tensor recovery under overparameterization is far from incremental.
>
> 3.  **Our experiments on both synthetic and real datasets demonstrate that FGD with small initialization achieves the smallest recovery error, and FGD with early stopping achieves slightly higher but still small recovery error.** Both variants significantly outperform convex methods, tensor factorization methods, and nonconvex regularization approaches. These strong empirical results further confirm the novelty and practical impact of our findings.
>
> Taken together, these contributions go beyond incremental improvements and provide **new theoretical understanding, new analytical tools, and new empirical insights** into noisy and overparameterized tensor recovery.
>
>
> It is worth noting that the other three reviewers strongly endorsed our theoretical contributions. Reviewer Zuao stated that our findings are "**definitely interesting.**" Reviewer wQhQ commented that “**the technical strength of analysis in this paper seems impressive**” and "**this is a quality submission.**" Reviewer SVNR remarked that our finding "**is both elegant and encouraging.**" and "**the paper is well structured and clearly written, particularly on the mathematical side.**"
>
> **Weakness 2: discuss the relationship with [1] in detail**
>
> Both this work and [1] employ factorized gradient descent algorithms to solve the low-tubal-rank tensor recovery problem. Reference [1] was the first to apply FGD to this problem and provided convergence and recovery error analyses. However, our work differs significantly from [1] in several key aspects:
>
> **(1) Initialization:** [1] relies on spectral initialization to obtain a sufficiently good starting point for its theoretical analysis. In contrast, our method requires only a small random initialization to guarantee convergence. These two initialization strategies lead to entirely different analytical frameworks and theoretical results.
>
> **(2) Convergence rate:** In [1], the convergence rate under over-parameterization is sublinear, whereas our analysis shows that the rate improves to linear in the over-parameterized regime, which is the fastest convergence rate achievable for gradient-based methods.
>
> **(3) Recovery error:** The recovery error in [1] depends on the over-parameterized tubal rank $R$, while ours depends only on the true tubal rank $r$.  When $R$ becomes very large, the recovery error in [1] increases significantly, as shown in **Figure 2 of Section 4**. In contrast, our recovery error remains unaffected.
>
> **(4) Sampling complexity:** In [1], the sample complexity is proportional to $16R^2$, whereas in our result, it is proportional to $4r^2$. Hence, the sample complexity in [1] grows significantly as the overparameterized rank $R$ increases, while our sample complexity does not depend on $R$.
>
> [1] Z. Liu, Z. Han, Y. Tang, X. -L. Zhao and Y. Wang, "Low-Tubal-Rank Tensor Recovery via Factorized Gradient Descent," in _IEEE Transactions on Signal Processing_, vol. 72, pp. 5470-5483, 2024

---

> ### Author Response · Authors · 2025-11-21
> **Response to Reviewer svxt (2/7)**
>
> **Weakness 3: provide a careful comparison with  "Implicit Regularization for Tubal Tensors via GD" [5] and "A Validation Approach to Over-parameterized Matrix and Image Recovery" [10]**
>
> **Comparison with [5]**
>
> Our work differs from [5] in several key aspects:
>
> 1. **Different research goals.**
>    While [5] focuses on the implicit regularization effect of small initialization, our goal is to provide theoretical guarantees for noisy low-tubal-rank tensor recovery under small initialization. In the overparameterized setting, the noisy and noiseless cases differ significantly. In the noiseless case, it is sufficient to show that the in-subspace error contracts while the overparameterization term remains small. After noise is introduced, however, the gradient includes random components arising from the adjoint of the measurement operator acting on the noise. These errors accumulate across iterations, and because of overparameterization, the extra degrees of freedom tend to absorb and amplify the noise. The nontrivial nature of handling noise under overparameterization has also been emphasized in the matrix recovery literature [6-8].
>
> 2. **Different technical tools.**
> First, their analysis splits the FGD trajectory into only two stages—the spectral stage and the convergence stage, which does not allow a precise characterization of the noise evolution. In contrast, we introduce a four-phase decomposition that provides a much finer description of the trajectory, enabling us to track the effect of noise throughout all stages. Second, we use the tensor tubal nuclear norm induced by the average rank, whereas [5] uses the one induced by the tubal rank. Since the average rank better captures the tensor's low-rank structure (see [9] for a detailed comparison).  Consequently, directly extending the results of [5] to the noisy tensor setting does not yield minimax-optimal recovery guarantees. In addition, because of the presence of noise, the overparameterization term grows faster than in the noiseless case. As a result, it becomes necessary to carefully control the initialization scale, step size, and number of iterations to ensure that there is sufficient room for effective early stopping.
>
> 3. **Different theoretical results.**
>    Our analysis requires less restrictive bounds on parameters. For example, the upper bound on the initialization scale $\alpha$ in our Theorem 2 is significantly more relaxed than that in [5], Theorem 3.1. Moreover, the results of [5], Theorem 3.1 are restricted to overparameterized settings with $R \ge 3r$, while our analysis covers both the exactly parameterized case $R = r$ and the overparameterized case $r < R<3r$， making our guarantees more comprehensive.
>
> [5] Karnik, S., Veselovska, A., Iwen, M., & Krahmer, F. Implicit Regularization for Tubal Tensor Factorizations via Gradient Descent. In Forty-second _International Conference on Machine Learning_.
>
> [6] Ma, J., & Fattahi, S. (2023). Global convergence of sub-gradient method for robust matrix recovery: Small initialization, noisy measurements, and over-parameterization. _Journal of Machine Learning Research_, _24_(96), 1-84.
>
> [7] Zhuo, J., Kwon, J., Ho, N., & Caramanis, C. (2024). On the computational and statistical complexity of over-parameterized matrix sensing. _Journal of Machine Learning Research_, _25_(169), 1-47.
>
> [8] Zhang, J., Zhang, R. Y., & Chiu, H. M. (2024, April). Fast and accurate estimation of low-rank matrices from noisy measurements via preconditioned non-convex gradient descent. In _International Conference on Artificial Intelligence and Statistics_ (pp. 3772-3780). PMLR.
>
> [9] Lu, C., Feng, J., Chen, Y., Liu, W., Lin, Z., & Yan, S. (2019). Tensor robust principal component analysis with a new tensor nuclear norm. _IEEE transactions on pattern analysis and machine intelligence_, _42_(4), 925-938.

---

> ### Author Response · Authors · 2025-11-21
> **Response to Reviewer svxt (3/7)**
>
> **Weakness 3: provide a careful comparison with  "Implicit Regularization for Tubal Tensors via GD " [5] and "A Validation Approach to Over-parameterized Matrix and Image Recovery" [10]**
>
>  **Comparison with [10]**
>
> Relative to [10], our differences are as follows:
>
>
> 1. **Different problem settings.**
>    [10] studies *noisy low-rank matrix recovery*, whereas we address the more complex *noisy low-tubal-rank tensor recovery* problem, which presents several nontrivial challenges.
>
> 2. **Different technical challenges and methodology.**
>    Extending matrix analysis to the tensor case is nontrivial:
>    * **Non-complementarity of range and kernel.**
>       In the matrix case, the range and the kernel are complementary subspaces. This property no longer holds for third-order tubal tensors. If the tensor contains non-invertible tubes in its t-SVD—equivalently, if some frequency slices vanish in the Fourier domain—then the range and kernel share common generators. As a result, the classical decomposition of gradient updates into a "signal term" and an "over-parameterization term" fails on these non-invertible tubes. This necessitates introducing a more refined notion of tensor condition number to track the identifiable and unidentifiable components separately.
>
>    * **Inter-slice coupling induced by the measurement operator.** For the power method, each frequency slice of a tubal tensor behaves like an independent matrix power iteration, a known fact in the literature [11]. However, in gradient descent for tensor recovery, the measurement operator and its adjoint couple information across all frequency slices. Specifically, for the observation model ${y}=\mathfrak{{M}}(\mathcal{{X}}\_\star)+{s}\in\mathbb{R}^m,\ \mathcal{X}\_\star\in\mathbb{R}^{n\times n\times k}$, we have $$y\_i=\langle \mathcal{{A}}\_i, \mathcal{{X}}\_\star\rangle+s\_i=\frac{1}{k}\langle\overline{ \mathcal{{A}}\_i}, \overline{\mathcal{{X}}\_\star}\rangle+s\_i =\frac{1}{k}\sum_{q=1}^k\langle \overline{A}\_i^{(q)} , \overline{X\_\star}^{(q)}\rangle+s\_i,i=1,...,m.$$ Then we have $$\overline{\mathfrak{M}^*(\mathfrak{M}(\mathcal{X}\_\star))}^{(j)}=\sum_{i=1}^m\sum_{q=1}^k \overline{\mathcal{A}\_i}^{(j)}\langle \overline{A}\_i^{(q)} , \overline{X\_\star}^{(q)}\rangle.$$ Consequently, the update of any single slice depends on all other slices, making it impossible to analyze the slices independently, as in the power method.
>
>    * **Minimax error for noisy low-tubal-rank tensor recovery.** In the matrix setting, Reference [12] has already established the minimax error for noisy matrix sensing. To the best of our knowledge, however, no such minimax error analysis exists for the tubal tensor setting. Therefore, we need to establish the minimax error bound for the noisy low-tubal-rank tensor recovery problem to show that our method matches the fundamental information-theoretic limit of the problem.
>
>
>
> [10] Ding, L., Qin, Z., Jiang, L., Zhou, J., & Zhu, Z. (2025, June). A validation approach to over-parameterized matrix and image recovery. In _Conference on Parsimony and Learning_ (pp. 323-350). PMLR.
>
> [11] Gleich, D. F., Greif, C., & Varah, J. M. (2013). The power and Arnoldi methods in an algebra of circulants. _Numerical Linear Algebra with Applications_, 20(5), 809-831.
>
> [12] Candes, E. J., & Plan, Y. (2011). Tight oracle inequalities for low-rank matrix recovery from a minimal number of noisy random measurements. _IEEE Transactions on Information Theory_, 57(4), 2342-2359.

---

> ### Author Response · Authors · 2025-11-21
> **Response to Reviewer svxt (4/7)**
>
> **Weakness 4: A thorough discussion and empirical comparison in real-world scenarios are necessary**
>
> The goal of this work is not to propose a new rank estimator, but to provide a theoretical and practical framework that is insensitive to rank overestimation under an over-parameterized setting. Our core contribution lies in **establishing unified convergence guarantees and near-minimax error bounds for gradient descent with small random initialization and data-driven early stopping, where the recovery error does not deteriorate as the chosen rank increases.** We agree with the reviewer that in practical applications, what truly matters is which estimated rank leads to better recovery performance, and such ranks are often non-unique.
>
> Therefore, we conducted two real-data experiments to validate the effectiveness of our method. First, we conducted color image completion experiments with Gaussian noise. We randomly selected 50 color images of size $481\times 321\times 3$ from the Berkeley Segmentation Dataset, added Gaussian noise drawn from $\mathcal{N}(0,\sigma^2)$, and applied random sampling with rate $p$. We compared our method with the classical convex approach TNN [13] and several nonconvex baselines [14-17]. Under different sampling rates and noise variances, our method consistently achieved the highest PSNR and the lowest relative recovery error among all approaches, as shown in **Table 1** below. In addition, we performed experiments to evaluate how different choices of the over-parameterized rank $R$ affect the final recovery results on 50 color images. The results show that when $R = [50, 75, 100, 125, 150]$, the PSNR and relative recovery errors remain nearly identical, demonstrating that our method is **highly robust to the choice of rank.** The results are shown in **Figure 4** of the main text.
>
> Beyond image completion, we also performed video completion experiments with Gaussian noise. We randomly selected four videos from the YUV Video Sequences dataset, extracted the first 30 frames of each to form tensors of size $176 \times 144 \times 30$, added Gaussian noise drawn from $\mathcal{N}(0,\sigma^2)$, and again applied sampling rate $p$. As shown in **Table 2**, our method achieves the smallest relative recovery error and the highest PSNR values. In addition, we conducted experiments to examine how choosing different over-parameterized ranks $R$ affects the final recovery results across four videos. The results show that when $R = [40,50,60,70,80]$, the PSNR and relative recovery errors are nearly identical, demonstrating that **our method is highly robust to the choice of rank.**  Additional experimental results are provided in **Section J.2**.
>
> Experiments on these two real-world datasets show that our method does not rely on knowing the exact tensor rank. Instead, one can simply choose a slightly overestimated rank and still obtain excellent recovery performance.

---

> ### Author Response · Authors · 2025-11-21
> **Response to Reviewer svxt (5/7): experimental results**
>
> **Table 1: Average PSNR and average Relative Error for color image completion experiments on 50 color images using different methods.  “FGD-ES” denotes FGD with early stopping, while “FGD-best” refers to the minimum error achieved by FGD over all iterations.**
>
> | Methods   | p=0.2,σ=0.07 PSNR ↑ | p=0.2,σ=0.07 RE ↓ | p=0.2,σ=0.1 PSNR ↑ | p=0.2,σ=0.1 RE ↓ | p=0.3,σ=0.07 PSNR ↑ | p=0.3,σ=0.07 RE ↓ | p=0.3,σ=0.1 PSNR ↑ | p=0.3,σ=0.1 RE ↓ |
> |-----------|----------------------|----------------------|---------------------|---------------------|----------------------|----------------------|----------------------|----------------------|
> | TCTF [14]      | 16.5892              | 0.3175               | 16.5484             | 0.3191              | 20.6744              | 0.2008               | 20.6335              | 0.2024               |
> | TNN [13]      | 21.2692              | 0.1851               | 19.7672             | 0.2188              | 22.0592              | 0.1681               | 20.1682              | 0.2082               |
> | TC-RE [15]    | 20.9288              | 0.1921               | 19.5480             | 0.2242              | 21.5387              | 0.1782               | 19.8376              | 0.2161               |
> | UTF  [16]     | 16.3227              | 0.3243               | 14.8770             | 0.3802              | 19.2245              | 0.2355               | 17.8283              | 0.2734               |
> | GTNN-HOP$_{0.3}$ [17]     | 22.1092              | 0.1675               | 20.3132             | 0.2051              | 23.1542              | 0.1481               | 21.1111              | 0.1867               |
> | FGD-ES    | *22.5912*           | *0.1616*             | *21.7977*           | *0.1765*            | *23.6579*            | *0.1426*             | *22.7157*           | *0.1585*            |
> | FGD-best  | **22.7438**          | **0.1587**           | **21.9268**         | **0.1739**          | **23.8422**          | **0.1395**           | **22.8550**          | **0.1559**           |
>
>
> **Table 2: Average PSNR and average Relative Error for video completion experiments on four videos using different methods.**
>
> | Methods   | p=0.3, σ=0.03 PSNR ↑ | p=0.3, σ=0.03 RE ↓ | p=0.3, σ=0.05 PSNR ↑ | p=0.3, σ=0.05 RE ↓ | p=0.4, σ=0.03 PSNR ↑ | p=0.4, σ=0.03 RE ↓ | p=0.4, σ=0.05 PSNR ↑ | p=0.4, σ=0.05 RE ↓ |
> |-----------|----------------------|----------------------|----------------------|----------------------|----------------------|----------------------|----------------------|----------------------|
> | UTF [16]       | 7.8242               | 0.2760               | 7.3535               | 0.2884               | 6.8286               | 0.3112               | 5.5376               | 0.3559               |
> | TNN [13]      | 20.2110              | 0.0659               | 17.2288              | 0.0934               | 20.4965              | 0.0639               | 17.1281              | 0.0947               |
> | TC-RE [15]    | 19.7102              | 0.0698               | 16.9971              | 0.0960               | 19.7435              | 0.0698               | 16.4039              | 0.1029               |
> | GTNN-HOP$_{0.3}$ [17]      | 19.6553              | 0.0706               | 16.1069              | 0.1070               | 20.0091              | 0.0680               | 16.2680              | 0.1051               |
> | GTNN-HOP$_{0.6}$ [17]      | 20.2583              | 0.0659               | 16.8056              | 0.0987               | 20.5764              | 0.0636               | 16.8933              | 0.0978               |
> | FGD-ES    | *22.0830*       | *0.0529*        | *21.0200*       | *0.0597*        | *22.2876*       | *0.0517*       | *21.5831*      | *0.0559*        |
> | FGD-best  | **22.1411**          | **0.0525**           | **21.1517**          | **0.0588**           | **22.3001**          | **0.0516**           | **21.7213**          | **0.0551**           |
>
> [13] Lu, C., Feng, J., Lin, Z., & Yan, S. (2018). Exact Low Tubal Rank Tensor Recovery from Gaussian Measurements. In _Proceedings of the Twenty-Seventh International Joint Conference on Artificial Intelligence_ (pp. 2504-2510). International Joint Conferences on Artificial Intelligence Organization.
>
> [14] Zhou, P., Lu, C., Lin, Z., & Zhang, C. (2017). Tensor factorization for low-rank tensor completion. _IEEE Transactions on Image Processing_, _27_(3), 1152-1163.
>
> [15] Shi, Q., Cheung, Y. M., & Lou, J. (2021). Robust tensor SVD and recovery with rank estimation. _IEEE Transactions on Cybernetics_, _52_(10), 10667-10682.
>
> [16] Du, S., Xiao, Q., Shi, Y., Cucchiara, R., & Ma, Y. (2021). Unifying tensor factorization and tensor nuclear norm approaches for low-rank tensor completion. _Neurocomputing_, _458_, 204-218.
>
> [17] Wang, Z. Y., So, H. C., & Zoubir, A. M. (2024). Low-rank tensor completion via novel sparsity-inducing regularizers. _IEEE Transactions on Signal Processing_.

---

> ### Author Response · Authors · 2025-11-21
> **Response to Reviewer svxt (6/7)**
>
> **Weakness 5: comparisons and discussion of rank-estimation-based approaches**
>
> Thank you for your suggestion. In the revised version, we have added a discussion in **Section 3.2** on rank estimation–based methods and compared them with the setting of our work, covering References [2–4].
>
> The motivation of our work is to address the case of overestimated rank. We do not aim to explicitly estimate the true rank; instead, we start from a small-scale random initialization and provide error guarantees that are insensitive to over-parameterization. In contrast, References [2–4] focus on estimating or adaptively adjusting the rank during iteration.
>
> To improve clarity, we explicitly highlight the following differences in **Section 3.2**:
>
> 1. **Different research goals.** Our goal is to achieve stable recovery even when the specified rank upper bound exceeds the true rank, ensuring that the error does not deteriorate as the upper bound increases. In contrast, rank estimation methods aim to identify or approximate the true rank.
>
> 2. **Noise models.** References [2] and [4] consider rank estimation in the presence of sparse noise, while reference [3] focuses on fast and robust rank estimation in the noiseless setting. Our results, on the other hand, cover sub-Gaussian noise and provide rigorous theoretical guarantees.
>
> 3. **Theoretical properties.** To the best of our knowledge, the above works do not provide rank-independent error bounds under the t-SVD and tubal-rank setting. Our main contribution is to establish such rank-independent guarantees and demonstrate near-minimax statistical accuracy.
>
> Moreover, we compared representative rank-estimation method [1] on both image completion and video completion tasks. The corresponding results are presented in **Section 4** of the main text and in **Appendix J.2**, and some of the results are shown in the tables below.
>
> **Table 1: Average PSNR and average Relative Error for color image completion experiments on 50 color images using different methods.**
>
> | Methods   | p=0.2,σ=0.07 PSNR ↑ | p=0.2,σ=0.07 RE ↓ | p=0.2,σ=0.1 PSNR ↑ | p=0.2,σ=0.1 RE ↓ | p=0.3,σ=0.07 PSNR ↑ | p=0.3,σ=0.07 RE ↓ | p=0.3,σ=0.1 PSNR ↑ | p=0.3,σ=0.1 RE ↓ |
> |-----------|----------------------|----------------------|---------------------|---------------------|----------------------|----------------------|----------------------|----------------------|
> | TC-RE [15]    | 20.9288              | 0.1921               | 19.5480             | 0.2242              | 21.5387              | 0.1782               | 19.8376              | 0.2161               |
> | FGD-ES    | *22.5912*           | *0.1616*             | *21.7977*           | *0.1765*            | *23.6579*            | *0.1426*             | *22.7157*           | *0.1585*            |
> | FGD-best  | **22.7438**          | **0.1587**           | **21.9268**         | **0.1739**          | **23.8422**          | **0.1395**           | **22.8550**          | **0.1559**           |
>
>
> **Table 2: Average PSNR and average Relative Error for video completion experiments on four videos using different methods.**
>
> | Methods   | p=0.3,σ=0.03 PSNR ↑ | p=0.3,σ=0.03 RE ↓ | p=0.3,σ=0.05 PSNR ↑ | p=0.3,σ=0.05 RE ↓ | p=0.4,σ=0.03 PSNR ↑ | p=0.4,σ=0.03 RE ↓ | p=0.4,σ=0.05 PSNR ↑ | p=0.4,σ=0.05 RE ↓ |
> |-----------|----------------------|----------------------|----------------------|----------------------|----------------------|----------------------|----------------------|----------------------|
> | TC-RE [15]    | 19.7102              | 0.0698               | 16.9971              | 0.0960               | 19.7435              | 0.0698               | 16.4039              | 0.1029               |
> | FGD-ES    | *22.0830*       | *0.0529*        | *21.0200*       | *0.0597*        | *22.2876*       | *0.0517*       | *21.5831*      | *0.0559*        |
> | FGD-best  | **22.1411**          | **0.0525**           | **21.1517**          | **0.0588**           | **22.3001**          | **0.0516**           | **21.7213**          | **0.0551**           |
>
> [15] Shi, Q., Cheung, Y. M., & Lou, J. (2021). Robust tensor SVD and recovery with rank estimation. IEEE Transactions on Cybernetics, 52(10), 10667-10682.

---

> ### Author Response · Authors · 2025-11-21
> **Response to Reviewer svxt (7/7)**
>
> **Weakness 7: no other recent tensor decomposition methods (from the last five years) are compared**
>
> Thank you for your suggestion. In the revised manuscript, we have added an additional tensor-factorization baseline and a non-convex regularization baseline  (from the last five years), along with other convex and non-convex methods, to the real-data experiments. The corresponding results are presented in **Section 4** and **Appendix J.2**, and some of the results are shown in the tables below. The results show that our approach consistently achieves the lowest recovery error among all compared methods.
>
>
> **Table 1: Average PSNR and average Relative Error for color image completion experiments on 50 color images using different methods.**
>
> | Methods   | p=0.2,σ=0.07 PSNR ↑ | p=0.2,σ=0.07 RE ↓ | p=0.2,σ=0.1 PSNR ↑ | p=0.2,σ=0.1 RE ↓ | p=0.3,σ=0.07 PSNR ↑ | p=0.3,σ=0.07 RE ↓ | p=0.3,σ=0.1 PSNR ↑ | p=0.3,σ=0.1 RE ↓ |
> |-----------|----------------------|----------------------|---------------------|---------------------|----------------------|----------------------|----------------------|----------------------|
> | UTF  [16]     | 16.3227              | 0.3243               | 14.8770             | 0.3802              | 19.2245              | 0.2355               | 17.8283              | 0.2734               |
> | GTNN-HOP$_{0.3}$ [17]     | 22.1092              | 0.1675               | 20.3132             | 0.2051              | 23.1542              | 0.1481               | 21.1111              | 0.1867               |
> | FGD-ES    | *22.5912*           | *0.1616*             | *21.7977*           | *0.1765*            | *23.6579*            | *0.1426*             | *22.7157*           | *0.1585*            |
> | FGD-best  | **22.7438**          | **0.1587**           | **21.9268**         | **0.1739**          | **23.8422**          | **0.1395**           | **22.8550**          | **0.1559**           |
>
>
> **Table 2: Average PSNR and average Relative Error for video completion experiments on four videos using different methods.**
>
> | Methods   | p=0.3,σ=0.03 PSNR ↑ | p=0.3,σ=0.03 RE ↓ | p=0.3,σ=0.05 PSNR ↑ | p=0.3,σ=0.05 RE ↓ | p=0.4,σ=0.03 PSNR ↑ | p=0.4,σ=0.03 RE ↓ | p=0.4,σ=0.05 PSNR ↑ | p=0.4,σ=0.05 RE ↓ |
> |-----------|----------------------|----------------------|----------------------|----------------------|----------------------|----------------------|----------------------|----------------------|
> | UTF [16]       | 7.8242               | 0.2760               | 7.3535               | 0.2884               | 6.8286               | 0.3112               | 5.5376               | 0.3559               |
> | GTNN-HOP$_{0.3}$ [17]      | 19.6553              | 0.0706               | 16.1069              | 0.1070               | 20.0091              | 0.0680               | 16.2680              | 0.1051               |
> | GTNN-HOP$_{0.6}$ [17]      | 20.2583              | 0.0659               | 16.8056              | 0.0987               | 20.5764              | 0.0636               | 16.8933              | 0.0978               |
> | FGD-ES    | *22.0830*       | *0.0529*        | *21.0200*       | *0.0597*        | *22.2876*       | *0.0517*       | *21.5831*      | *0.0559*        |
> | FGD-best  | **22.1411**          | **0.0525**           | **21.1517**          | **0.0588**           | **22.3001**          | **0.0516**           | **21.7213**          | **0.0551**           |
>
>
> [16] Du, S., Xiao, Q., Shi, Y., Cucchiara, R., & Ma, Y. (2021). Unifying tensor factorization and tensor nuclear norm approaches for low-rank tensor completion. _Neurocomputing_, _458_, 204-218.
>
> [17] Wang, Z. Y., So, H. C., & Zoubir, A. M. (2024). Low-rank tensor completion via novel sparsity-inducing regularizers. _IEEE Transactions on Signal Processing_.

---

> > ### Comment · Reviewer_svxt · 2025-11-21
> >
> > I sincerely appreciate the author's detailed response. My concern has been fully resolved, and I will update my score.

---

> > > ### Author Response · Authors · 2025-11-23
> > >
> > > We sincerely appreciate the reviewer for the encouraging feedback and for updating the score. We are glad that our response has fully addressed the concerns. Please feel free to reach out if the reviewer has any further concerns, and we would be glad to discuss them with the reviewer.

---

### Official Review · Reviewer_wQhQ · 2025-11-02

**Soundness:** 3
**Presentation:** 3
**Contribution:** 3
**Rating:** 6
**Confidence:** 4

**Summary:**

The paper analyzes the optimization dynamics of factorized gradient descent (FGD) for low tubal rank tensor recovery, which is based on the quite recent t-SVD for tensors.  The authors prove that using a small random initialization for FGD together with suitable early stopping of FGD leads to a minimax optimal estimator, under an RIP-type condition.  The authors include synthetic experiments that validate the theory.  Overall, this is a quality submission.  I would be open to *raising* my score if the authors addressed a couple of my concerns in the Weaknesses section below with their rebuttal.

**Strengths:**

- The optimization dynamics are interesting and not so intuitive.  In particular, the authors find that small random initialization is better than smart spectral initialization for this problem.  Further, FGD exhibits non-monotonicity with respect to error with the ground-truth; therefore, early stopping is needed.

- The technical strength of analysis in this paper seems impressive.

- The authors confirm their results also through numerical simulations.  Random small initialization and early stopping indeed improve the estimated tensor in experiments.

**Weaknesses:**

- It is unclear whether the factored form of the tensor used by the authors imposes symmetry and/or psd constraints.  The authors use $\mathcal{U} \ast \mathcal{U}^{\top}$ as their ansatz for the low tubal rank tensor.

- The relation to low rank matrix recovery (i.e., with matrices and matrix SVD, rather than with tensors and t-SVD) is not well-specified as far as I can tell.  The authors should comment on this.

- There are no real data experiments.  Real data would be nice, because there are certain RIP assumptions in this work, and one wonders how well they capture real data situations.

**Questions:**

Line 60-61: The authors write "Under the t-SVD framework, since problem (2) is NP-hard, a common approach is to relax the tubal-rank constraint to the tensor nuclear norm."  Is it actually known that the low tubal rank recovery problem (2) is NP-hard?  What is a reference?  This wouldn't be in Hillar-Lim's paper, for instance.

Line 60-61: The authors should perhaps also say "tubal tensor nuclear norm", and provide a reference for its definition.  My understanding is that the authors mean the sum of singular values from the t-SVD, rather than the tensor nuclear norm as related to the CP decomposition.  That latter tensor nuclear norm is NP-hard to compute.

Lines 70-72: Would using the ansatz $\mathcal{A} = \mathcal{U} \ast \mathcal{U}^{\top}$ imply symmetry and/or some sort of positive semi-definiteness in $\mathcal{A}$?  If so, the authors should be explicit about these assumptions on $\mathcal{A}$.  To me the natural low-rank ansatz would be $\mathcal{A} = \mathcal{U} \ast \mathcal{V}^{\top}$ where $\mathcal{U}$ need not equal $\mathcal{V}$.

Line 131: In the caption of Table 1, start the second sentence as "The noise vector...", i.e. please drop the leading "And".

Line 202: Explain better what the block diagonal matrix $\bar{Y}$ is.

Lines 240-269: In Theorem 2, item 1, is this bound true for all $t \geq t_1$?  If not, and the error can increase later on, how about formulating item 1 with $\hat{t}$ as you do with items 2 and 3 in Theorem 2?

Section 4: It would be appreciated if the authors included a real data experiment.

General question: How do the results in this work relate to results for factorized gradient descent with just matrices and matrix SVD, rather than with tensors and t-SVD?  Are there parallel results in the matrix case?  How do the results here compare?

---

> ### Author Response · Authors · 2025-11-21
> **Response to Reviewer wQhQ (1/5)**
>
> We sincerely thank the reviewer for carefully reading our paper and providing highly professional feedback. We are also pleased to hear the reviewer noted that the optimization dynamics are interesting. Below, we address the concerns.
>
>  **Weakness 1 & Question 3: symmetry and PSD constraints**
>
> In the main text, we adopt the parameterization $\mathcal{A} = \mathcal{U} * \mathcal{U}^\top$, which indeed implies two properties:
> (1) tensor symmetry, i.e., $\mathcal{A} = \mathcal{A}^\top$; and
> (2) tensor PSD, meaning that each FFT slice $\overline{\mathcal{A}}^{(i)} = \bar{U}^{(i)} {\bar{U}^{(i)\top}}$ is positive semidefinite.
>
> We have made this assumption explicit and unified the notation in the revised version. In prior matrix and tensor factorization works [1-5], it is common to begin the analysis in the symmetric and PSD setting. This avoids the rotation and scaling ambiguities introduced by two factors, which in turn makes our core phenomenon more transparent: **under over-parameterization with noise, small initialization combined with early stopping yields an error that is independent of the overestimated rank R and is nearly minimax.** We agree with the reviewer that the form $\mathcal{A} = \mathcal{U} * \mathcal{V}^\top$ is more natural and practical. Our real-data experiments demonstrate that our conclusions also hold in the general setting. From a theoretical perspective, our four-phase analytical framework can be extended to the general case as well, although additional effort is required to handle the imbalance between the two factor tensors. We are currently working on this extension.
>
> [1] Liu, Z., Han, Z., Tang, Y., Zhao, X. L., & Wang, Y. (2024). Low-Tubal-Rank Tensor Recovery via Factorized Gradient Descent. _IEEE Transactions on Signal Processing_.
>
> [2] Karnik, S., Veselovska, A., Iwen, M., & Krahmer, F. Implicit Regularization for Tubal Tensor Factorizations via Gradient Descent. In Forty-second International Conference on Machine Learning.
>
> [3] Stöger, D., & Soltanolkotabi, M. (2021). Small random initialization is akin to spectral learning: Optimization and generalization guarantees for overparameterized low-rank matrix reconstruction. _Advances in Neural Information Processing Systems_, 34, 23831-23843.
>
> [4] Zhuo, J., Kwon, J., Ho, N., & Caramanis, C. (2024). On the computational and statistical complexity of over-parameterized matrix sensing. _Journal of Machine Learning Research_, _25_(169), 1-47.
>
> [5] Ma, J., & Fattahi, S. (2023). Global convergence of sub-gradient method for robust matrix recovery: Small initialization, noisy measurements, and over-parameterization. _Journal of Machine Learning Research_, _24_(96), 1-84

---

> ### Author Response · Authors · 2025-11-21
> **Response to Reviewer wQhQ (2/5)**
>
> **Weakness 2 & Question 8: relation to low rank matrix recovery**
>
> Our framework reduces to the matrix setting when $n_3=1$: the t-product becomes matrix multiplication, tubal rank becomes matrix rank, and $\mathcal X=\mathcal{U}*\mathcal{U}^{\top}$ reduces to $X=UU^\top$. In this special case, Theorem 2 recovers the same qualitative phenomenon reported for matrix FGD: small initialization and early stopping yield error bounds that do not deteriorate with the over-specified rank, as shown in the literature [6].
>
> However, extending the matrix setting to the tensor setting is nontrivial, as one must address several challenges unique to tensors:
> 1. **Non-complementarity of range and kernel.**
>       In the matrix case, the range and the kernel are complementary subspaces. This property no longer holds for third-order tubal tensors. If the tensor contains non-invertible tubes in its t-SVD—equivalently, if some frequency slices vanish in the Fourier domain—then the range and kernel share common generators. As a result, the classical decomposition of gradient updates into a "signal term" and an "over-parameterization term" fails on these non-invertible tubes. This necessitates introducing a more refined notion of tensor condition number to track the identifiable and unidentifiable components separately.
>
> 2. **Inter-slice coupling induced by the measurement operator.** For the power method, each frequency slice of a tubal tensor behaves like an independent matrix power iteration, a known fact in the literature [7]. However, in gradient descent for tensor recovery, the measurement operator and its adjoint couple information across all frequency slices. Specifically, for the observation model ${y}=\mathfrak{{M}}(\mathcal{{X}}\_\star)+{s}\in\mathbb{R}^m,\ \mathcal{X}\_\star\in\mathbb{R}^{n\times n\times k}$, we have $$y\_i=\langle \mathcal{{A}}\_i, \mathcal{{X}}\_\star\rangle+s\_i=\frac{1}{k}\langle\overline{ \mathcal{{A}}\_i}, \overline{\mathcal{{X}}\_\star}\rangle+s\_i =\frac{1}{k}\sum_{q=1}^k\langle \overline{A}\_i^{(q)} , \overline{X\_\star}^{(q)}\rangle+s\_i,i=1,...,m.$$ Then we have $$\overline{\mathfrak{M}^*(\mathfrak{M}(\mathcal{X}\_\star))}^{(j)}=\sum_{i=1}^m\sum_{q=1}^k \overline{\mathcal{A}\_i}^{(j)}\langle \overline{A}\_i^{(q)} , \overline{X\_\star}^{(q)}\rangle.$$ Consequently, the update of any single slice depends on all other slices, making it impossible to analyze the slices independently, as in the power method.
>
> 3. **Minimax error for noisy low-tubal-rank tensor recovery.** In the matrix setting, Reference [8] has already established the minimax error for noisy matrix sensing. To the best of our knowledge, however, no such minimax error analysis exists for the tubal tensor setting. Therefore, we need to establish the minimax error bound for the noisy low-tubal-rank tensor recovery problem to show that our method matches the fundamental information-theoretic limit of the problem.
>
> [6] Ding, L., Qin, Z., Jiang, L., Zhou, J., & Zhu, Z. (2025, June). A validation approach to over-parameterized matrix and image recovery. In _Conference on Parsimony and Learning_ (pp. 323-350). PMLR.
>
> [7] Gleich, D. F., Greif, C., & Varah, J. M. (2013). The power and Arnoldi methods in an algebra of circulants. _Numerical Linear Algebra with Applications_, 20(5), 809-831.
>
> [8] Candes, E. J., & Plan, Y. (2011). Tight oracle inequalities for low-rank matrix recovery from a minimal number of noisy random measurements. _IEEE Transactions on Information Theory_, 57(4), 2342-2359.
>
>  **Weakness 3 & Question 7: real data experiments**
>
> Thank you for the suggestion. In the revised version, we have added real-data experiments in Section 4, with additional results  provided in **Appendix J.2**. First, we conducted color image completion experiments with Gaussian noise. We randomly selected 50 color images of size $481\times 321\times 3$ from the Berkeley Segmentation Dataset, added Gaussian noise drawn from $\mathcal{N}(0,\sigma^2)$, and applied random sampling with rate $p$. We compared our method with the classical convex approach TNN and several nonconvex baselines. Under different sampling rates and noise variances, our method consistently achieved the highest PSNR and the lowest relative recovery error among all approaches, as shown in **Table 1** below. In addition, we verified that GD with small initialization is highly robust to the choice of the over-parameterized rank $R$. The results are shown in **Figure 4** of the main text.
>
> Beyond image completion, we also performed video completion experiments. We selected four videos from the YUV Video Sequences dataset, extracted the first 30 frames of each to form tensors of size $176 \times 144 \times 30$. As shown in **Table 2**, our method achieves the smallest relative recovery error and the highest PSNR. Additional experimental results are provided in **Appendix J.2**.  These experiments show that our method is broadly applicable and performs well on real data.

---

> ### Author Response · Authors · 2025-11-21
> **Response to Reviewer wQhQ (3/5): experimental results**
>
> **Table 1: Average PSNR and average Relative Error for color image completion experiments on 50 color images using different methods. “FGD-ES” denotes FGD with early stopping, while “FGD-best” refers to the minimum error achieved by FGD over all iterations.**
>
> | Methods   | p=0.2,σ=0.07 PSNR ↑ | p=0.2,σ=0.07 RE ↓ | p=0.2,σ=0.1 PSNR ↑ | p=0.2,σ=0.1 RE ↓ | p=0.3,σ=0.07 PSNR ↑ | p=0.3,σ=0.07 RE ↓ | p=0.3,σ=0.1 PSNR ↑ | p=0.3,σ=0.1 RE ↓ |
> |-----------|----------------------|----------------------|---------------------|---------------------|----------------------|----------------------|----------------------|----------------------|
> | TCTF [9]      | 16.5892              | 0.3175               | 16.5484             | 0.3191              | 20.6744              | 0.2008               | 20.6335              | 0.2024               |
> | TNN [10]      | 21.2692              | 0.1851               | 19.7672             | 0.2188              | 22.0592              | 0.1681               | 20.1682              | 0.2082               |
> | TC-RE [11]    | 20.9288              | 0.1921               | 19.5480             | 0.2242              | 21.5387              | 0.1782               | 19.8376              | 0.2161               |
> | UTF  [12]     | 16.3227              | 0.3243               | 14.8770             | 0.3802              | 19.2245              | 0.2355               | 17.8283              | 0.2734               |
> | GTNN-HOP$_{0.3}$ [13]     | 22.1092              | 0.1675               | 20.3132             | 0.2051              | 23.1542              | 0.1481               | 21.1111              | 0.1867               |
> | FGD-ES    | *22.5912*           | *0.1616*             | *21.7977*           | *0.1765*            | *23.6579*            | *0.1426*             | *22.7157*           | *0.1585*            |
> | FGD-best  | **22.7438**          | **0.1587**           | **21.9268**         | **0.1739**          | **23.8422**          | **0.1395**           | **22.8550**          | **0.1559**           |
>
>
> **Table 2: Average PSNR and average Relative Error for video completion experiments on four videos using different methods.**
>
> | Methods   | p=0.3,σ=0.03  PSNR ↑ | p=0.3,σ=0.03  RE ↓ | p=0.3,σ=0.05  PSNR ↑ | p=0.3,σ=0.05  RE ↓ | p=0.4,σ=0.03  PSNR ↑ | p=0.4,σ=0.03  RE ↓ | p=0.4,σ=0.05  PSNR ↑ | p=0.4,σ=0.05  RE ↓ |
> |-----------|----------------------|----------------------|----------------------|----------------------|----------------------|----------------------|----------------------|----------------------|
> | UTF [12]       | 7.8242               | 0.2760               | 7.3535               | 0.2884               | 6.8286               | 0.3112               | 5.5376               | 0.3559               |
> | TNN [10]      | 20.2110              | 0.0659               | 17.2288              | 0.0934               | 20.4965              | 0.0639               | 17.1281              | 0.0947               |
> | TC-RE [11]    | 19.7102              | 0.0698               | 16.9971              | 0.0960               | 19.7435              | 0.0698               | 16.4039              | 0.1029               |
> | GTNN-HOP$_{0.3}$ [13]      | 19.6553              | 0.0706               | 16.1069              | 0.1070               | 20.0091              | 0.0680               | 16.2680              | 0.1051               |
> | GTNN-HOP$_{0.6}$ [13]      | 20.2583              | 0.0659               | 16.8056              | 0.0987               | 20.5764              | 0.0636               | 16.8933              | 0.0978               |
> | FGD-ES    | *22.0830*       | *0.0529*        | *21.0200*       | *0.0597*        | *22.2876*       | *0.0517*       | *21.5831*      | *0.0559*        |
> | FGD-best  | **22.1411**          | **0.0525**           | **21.1517**          | **0.0588**           | **22.3001**          | **0.0516**           | **21.7213**          | **0.0551**           |
>
>
> [9] Zhou, P., Lu, C., Lin, Z., & Zhang, C. (2017). Tensor factorization for low-rank tensor completion. _IEEE Transactions on Image Processing_, _27_(3), 1152-1163.
>
> [10] Lu, C., Feng, J., Lin, Z., & Yan, S. (2018). Exact Low Tubal Rank Tensor Recovery from Gaussian Measurements. In _Proceedings of the Twenty-Seventh International Joint Conference on Artificial Intelligence_ (pp. 2504-2510). International Joint Conferences on Artificial Intelligence Organization.
>
> [11] Shi, Q., Cheung, Y. M., & Lou, J. (2021). Robust tensor SVD and recovery with rank estimation. _IEEE Transactions on Cybernetics_, _52_(10), 10667-10682.
>
> [12] Du, S., Xiao, Q., Shi, Y., Cucchiara, R., & Ma, Y. (2021). Unifying tensor factorization and tensor nuclear norm approaches for low-rank tensor completion. _Neurocomputing_, _458_, 204-218.
>
> [13] Wang, Z. Y., So, H. C., & Zoubir, A. M. (2024). Low-rank tensor completion via novel sparsity-inducing regularizers. _IEEE Transactions on Signal Processing_.

---

> ### Author Response · Authors · 2025-11-21
> **Response to Reviewer wQhQ (4/5)**
>
> **Question 1: Is it actually known that the low tubal rank recovery problem (2) is NP-hard?**
>
> To the best of our knowledge, there is no paper that explicitly proves the low-tubal-rank recovery problem to be NP-hard, although many works state this fact without providing a full proof. We now include a self-contained proof.
> First, we define the affine rank minimization problem (ARM):
> $$
> \text{minimize}\ \operatorname{rank}(M)\ \operatorname{subject}\ \operatorname{to}\ \mathcal{A}(M)=b,
> $$
> where $M\in\mathbb{R}^{m\times n}$ is the optimization variable, and the linear map $\mathcal{A}:\mathbb{R}^{m\times n}\to \mathbb{R}^p$ and vector $b\in\mathbb{R}^{p}$ are given [14]. When the matrix variable is constrained to be diagonal, the affine rank minimization problem reduces to finding the sparsest vector in an affine subspace. This problem is commonly referred to as cardinality minimization, since one seeks the vector whose support has the smallest cardinality, and is known to be NP hard [15].
>
> Then we define the tubal-rank minimization problem (TRM):
> $$
> \operatorname{minimize}\ \operatorname{tubal-}\operatorname{rank}(\mathcal{X})\ \text{subject to}\ \mathfrak{M}(\mathcal{X})=d,
> $$
>
> where $\mathcal{X}\in\mathbb{R}^{m\times n\times k}$ is the optimization variable, and the linear map $\mathfrak{M}:\mathbb{R}^{m\times n\times k}\to \mathbb{R}^q$ and vector $d\in\mathbb{R}^{q}$ are given. Note that for any $\mathcal{X}\in\mathbb{R}^{m\times n\times k}$, we have tubal-rank$(\mathcal{X})=\max_{1\le i\le k} \operatorname{rank}(\bar{\mathcal{X}}^{(i)}).$
>
> Given an arbitrary ARM instance $(\mathcal{A},b)$, construct an instance variable $\mathcal{X}\in\mathbb{R}^{m\times n\times k},\ k\ge 2$ of TRM, then let
> $$
> \mathfrak{M}(\mathcal{X})= \begin{bmatrix}
> \text{vec}(\mathcal{X}(:,:,2)-\mathcal{X}(:,:,1))^\top;
> \cdots;
> \text{vec}(\mathcal{X}(:,:,k)-\mathcal{X}(:,:,1))^\top ;
> \mathcal{A}(\mathcal{X}(:,:,1))^\top
> \end{bmatrix}^\top= d=\begin{bmatrix} 0; \cdots; 0; b^\top \end{bmatrix}^\top,
> $$
> where $q=(k-1)mn+p.$ Clearly, this construction can be carried out in polynomial time.
> For any feasible ARM solution $M$ with $\mathcal{A}(M)=b$, consider the tensor $\mathcal{X}$ defined by $\mathcal{X}(:,:,j)=M$ for all $j=1,\dots,k$. Then $\mathcal{X}(:,:,j)-\mathcal{X}(:,:,1)=0$ for all $j\ge 2$ and $\mathcal{A}(\mathcal{X}(:,:,1))=\mathcal{A}(M)=b$, hence $\mathfrak{M}(\mathcal{X})=d$ and $\mathcal{X}$ is feasible for TRM. Moreover, since all frontal slices of $\mathcal{X}$ are identical, after applying the FFT along the third mode, only the first frequency slice is nonzero and equals a nonzero scalar multiple of $M$. Therefore, $\operatorname{tubal-rank}(\mathcal{X}) = \operatorname{rank}(M).$
>
>
> Conversely, any feasible TRM tensor $\mathcal{X}$ satisfies $\mathcal{X}(:,:,j)=\mathcal{X}(:,:,1)\ ,  j=2,...,k$; letting $M=\mathcal{X}(:,:,1)$, we have $\mathcal A(M)=b$ and $\operatorname{rank}(M)=\operatorname{tubal-rank}(\mathcal X)$.
> Therefore, the two problems have the same optimal value.
>
> If the TRM  could be solved in polynomial time, then by the equality above, we could obtain the optimal rank of ARM in a single call and thus solve ARM in polynomial time; however, since ARM is NP-hard in general, it follows that TRM  is NP-hard as well.
>
> [14] Recht, B., Fazel, M., & Parrilo, P. A. (2010). Guaranteed minimum-rank solutions of linear matrix equations via nuclear norm minimization. _SIAM review_, 52(3), 471-501.
>
> [15] Natarajan, B. K. (1995). Sparse approximate solutions to linear systems. _SIAM journal on computing_, 24(2), 227-234.

---

> ### Author Response · Authors · 2025-11-21
> **Response to Reviewer wQhQ (5/5)**
>
> **Question 2: definition of the tubal tensor nuclear norm**
>
> Thank you for the suggestion. We indeed mean the tubal tensor nuclear norm.
> The tubal tensor nuclear norm of a tensor $\mathcal{Y}\in\mathbb{R}^{n\_1\times n\_2\times n\_3}$ is defined as $ ||\mathcal{Y}||\_* := \frac{1}{n\_3}||\bar{Y}||_*,$
>  where $\bar{Y}\in\mathbb{C}^{n\_1 n\_3 \times n\_2 n\_3}$ denotes the block-diagonal matrix whose $i$-th diagonal block is the $i$-th frontal slice $\bar{Y}^{(i)}$ of $\bar{\mathcal{Y}}$. That is, $\bar{Y} = \mathtt{bdiag}(\bar{\mathcal{Y}}) =
>  \mathtt{diag}\big( \bar{Y}^{(1)}, \bar{Y}^{(2)}, \ldots, \bar{Y}^{(n_3)} \big)
> $[16].
> We have revised the text to state “tubal tensor nuclear norm” and added reference in the revised manuscript.
>
> [16] Lu, C., Feng, J., Chen, Y., Liu, W., Lin, Z., & Yan, S. (2019). Tensor robust principal component analysis with a new tensor nuclear norm. _IEEE transactions on pattern analysis and machine intelligence_, _42_(4), 925-938.
>
> **Question 3: symmetry and positive semi-definiteness assumptions**
>
> Please refer to our response to weakness 1.
>
> **Question 4: grammatical issue**
>
> Thank you for your careful review. We have corrected this issue in the revised version.
>
> **Question 5: definition of block diagonal matrix $\bar{Y}$**
>
> For any tensor $\mathcal{Y}\in\mathbb{R}^{m\times n\times k}$, we denote $\bar{Y}\in\mathbb{C}^{mk\times nk}$ as a block diagonal matrix with it's $i$-th block on the diagonal as the $i$-th frontal slice $\bar{Y}^{(i)}$ of $\bar{\mathcal{Y}}$, i.e.,
> $$
> \bar{Y}=\mathtt{bdiag}(\bar{\mathcal{Y}}) = \mathtt{diag}(\bar{Y}^{(1)};\bar{Y}^{(2)};...;\bar{Y}^{(k)}).
> $$Due to space limitations in the main text, we have added the corresponding explanation in **Appendix D**.
>
> **Question 6: explain the bound in Theorem 2, item 1**
>
> Yes, the bound in Theorem 2, item 1, holds uniformly for all $t\ge t_1$ under the stated step size and small-initialization conditions. For consistency with items 2 and 3, we have formulated item 1 with $\hat{t}$ as items 2 and 3 in the revised manuscript.
>
> **Question 7: add real data experiments**
>
> We have conducted real data experiments; please refer to our response to weakness 3.
>
> **Question 8: **relation to low rank matrix recovery****
>
> Please refer to our response to weakness 2.

---

### Official Review · Reviewer_Zuao · 2025-11-02

**Soundness:** 3
**Presentation:** 3
**Contribution:** 3
**Rating:** 4
**Confidence:** 3

**Summary:**

This paper studies the problem of recovering a low-tubal rank tensor $X_*$ from its noisy linear measurements, $y = M (X_*) + s$. Focusing on the commonly used Burer-Monteiro factorization framework, it assumes that $X_*$ can be decomposed as $U * U^\top$, and aims to minimize the quadratic loss function to recover $U$. The authors consider the fractorized gradient descent (FGD) method and attempts to derive upper bounds for the estimator error. A key challenge here is that the true tubal rank of $X_*$, denoted as $r$, is typically unknown, and error bounds in past work often depends on the over-estimated tubal-rank $R$.

The main contribution of this paper is to show that, both theoretically and empirically, using a small random initialization for FGD achieves a nearly minimax optimal estimation error, which only relies on the true tubal rank $r$, not the over-estimated tubal-rank $R$. To be specific: (i) Theorem 2 establishes estimator error bounds for early stopped FGD, under three different regimes of $r$ and $R$; (ii) Theorem 3 shows that the error bound of Theorem 2 is minimax optimal in the case of Gaussian noise; (iii) Finally, Theorem 4 shows that the early stopping time $t$ can be reliably chosen using sample spitting, thus giving a practical algorithm based on the theoretical results in Theorem 2 and 3.

**Strengths:**

The key findings of this paper, that the estimation error of FGD only depends on the true tubal-rank $r$, and that small initialization overcomes the dependency on the estimated rank $R$, are definitely interesting. The authors also compare their work with earlier results on the same topic to highlight their contributions, and include a proof sketch section to illustrate why FGD with small initialization works, which I really appreciate.

**Weaknesses:**

(i) The entire analysis replies on the T-PSD assumption on $X_*$, i.e., $X_*$ has the decomposition $X_* = U * U_*$. This is a significant simplification. The authors also acknowledge this limitation in Remark 5 and discussed a bit about extensions to general asymmetric case in Appendix I.
(ii) Although the error bounds in Theorem 2 does not depend on $R$, the initialization scale $\alpha$ and early stopping time $t$ depends on $R$. Further, in case 3 it seems to me that we can take $R$ to be so large that $\alpha$ and $t$ are close to $0$, which is quite counterintuitive. Could the authors clarify more on this? This is very important since the error bounds in this paper improves on previous results only in the case $R \geq 3r$.

**Questions:**

A few minor comments/suggestions for the authors:
(i) "CONDECOMP" should be "CANDECOMP" in line 051.
(ii) What is $R_n$ in Theorem 2, is it just $R$? Is it possible to state the choices of $\alpha$ and $t$ only in terms of $R$, since $r$ is unknown?
(iii) In line 216, perhaps it is better to replace $s$ as some other notation since $s$ denotes the noise in $y$.

---

> ### Author Response · Authors · 2025-11-21
> **Response to Reviewer Zuao**
>
> We sincerely thank the reviewer for carefully reading our paper and providing valuable feedback. We are delighted to hear the reviewer describe our findings as “definitely interesting.” Below, we address the concerns.
>
> **Weakness 1: T-PSD assumption**
>
> We agree that assuming T-PSD is a simplifying assumption. In prior matrix and tensor factorization works [1-5], it is common to begin the analysis in the symmetric and PSD setting. This avoids the rotation and scaling ambiguities introduced by two factors, which in turn makes our core phenomenon more transparent: **under over-parameterization with noise, small initialization combined with early stopping yields an error that is independent of the overestimated rank R and is nearly minimax.**  It is worth noting that in **Appendix I**, we have added additional experiments showing that the same phenomenon also holds in the non-symmetric and non-PSD case. Our four-phase analytical framework can likewise be extended to the general setting, although additional work is needed to handle the imbalance between the two factor tensors. We are currently pursuing this extension.
>
>
> [1] Liu, Z., Han, Z., Tang, Y., Zhao, X. L., & Wang, Y. (2024). Low-Tubal-Rank Tensor Recovery via Factorized Gradient Descent. _IEEE Transactions on Signal Processing_.
>
> [2] Karnik, S., Veselovska, A., Iwen, M., & Krahmer, F. Implicit Regularization for Tubal Tensor Factorizations via Gradient Descent. In Forty-second _International Conference on Machine Learning_.
>
> [3] Stöger, D., & Soltanolkotabi, M. (2021). Small random initialization is akin to spectral learning: Optimization and generalization guarantees for overparameterized low-rank matrix reconstruction. _Advances in Neural Information Processing Systems_, 34, 23831-23843.
>
> [4] Zhuo, J., Kwon, J., Ho, N., & Caramanis, C. (2024). On the computational and statistical complexity of over-parameterized matrix sensing. _Journal of Machine Learning Research_, _25_(169), 1-47.
>
> [5] Ma, J., & Fattahi, S. (2023). Global convergence of sub-gradient method for robust matrix recovery: Small initialization, noisy measurements, and over-parameterization. _Journal of Machine Learning Research_, _24_(96), 1-84.
>
> **Weakness 2: the initialization scale $\alpha$ and early stopping time $\hat{t}$ depend on $R$**
>
> We first clarify that the parameter in Theorem 2 is defined as **$R\_n = \min(R, n)$**, not $R$. We have emphasized this in the revision. Consequently, the initialization scale $\alpha$ and the early stopping time $\hat{t}$ depend on $R\_n$, which is bounded above by the tensor dimension $n$. Moreover, even if we let $R \to \infty$, neither $\alpha$ nor $\hat{t}$ approaches zero.
>
> **Questions 1 & 3: typos**
>
>  Thank you for your careful review. We have corrected these issues in the revised version.
>
>  **Question 2: definition of $R\_n$ and the choices of $\alpha$ and $\hat{t}$**
>
>  In Theorem 2 we define $R\_n := \min(R, n)$, and we have clarified this in the revision. With this definition, the choice of $\alpha$ depends only on $R_n$ and is independent of $r$. This is because we can simply choose $\alpha$ small enough to satisfy the condition in Theorem 2 for all $1 \le r \le R_n$, without requiring knowledge of the true rank. In practice, we also select $\alpha$ to be as small as possible.
>
> In contrast, we cannot guarantee theoretically that the choice of $\hat{t}$ is independent of $r$. The stopping time must be carefully controlled; otherwise, FGD may not have fully converged, or the error may have already begun to diverge. This is precisely why we rely on validation and early stopping to achieve near-optimal recovery error in practice.

---

> > ### Comment · Reviewer_Zuao · 2025-11-24
> >
> > I appreciate the authors' efforts to address my comments. I have raised my score to 6.

---

> > > ### Author Response · Authors · 2025-11-24
> > >
> > > We sincerely thank the reviewer for the response and for the recognition of our work. If the reviewer has any further questions, we would be very happy to discuss them.

---

### Author Response · Authors · 2025-11-21
**Response to All**

Dear reviewers and area chairs:

We thank all reviewers and area chairs for their careful review and constructive comments, which have been extremely helpful in improving the quality of our paper. We are especially grateful for the positive recognition of our work:
1. Reviewer Zuao stated that our findings are "**definitely interesting**"  and "**really appreciate**" that we provide a proof sketch.
2. Reviewer wQhQ commented that "**the technical strength of analysis in this paper seems impressive**" and "**this is a quality submission.**"
3. Reviewer svxt stated that we **"provided an error bound for low-tubal-rank tensor recovery when the tubal rank is overestimated in noisy settings**", **"with both theoretical guarantees and empirical validation.**"
3. Reviewer SVNR remarked that our finding "**is both elegant and encouraging**" and that "**the paper is well structured and clearly written, particularly on the mathematical side.**"

We have addressed all reviewer comments and revised the manuscript accordingly. The main changes in the revised version are as follows:

1.  Following the suggestions of reviewers wQhQ, sxvt, and SVNR, we have included a detailed comparison with matrix-based work in **Section 3.3**, highlighting the technical challenges unique to the tensor setting.

2.  Following the suggestions of reviewers wQhQ and sxvt, we have added real-data experiments on color image completion and video completion, with results presented in **Section 4** and **Appendix J.2**.

3. Following the suggestions of reviewer sxvt, we have clarified the distinctions between our work and prior approaches on tensor recovery in **Section 3.2**.

4.  Following the suggestions of reviewer SVNR, we have added an introduction to the tensor–tensor product framework in **Appendix D**, and included a numerical experiment in **Appendix E.2** to illustrate the four stages described in the proof sketch of Theorem 2.

All modifications in the main text are marked in blue, and the code of real data experiments is included in the supplementary materials.

We thank all reviewers and area chairs again!

Best,

Authors of Paper #5985

---

### Author Response · Authors · 2025-12-02
**Summary of Reviewer Discussions**

Dear reviewers and area chairs,

We are very grateful to the area chairs for their time and for the additional effort in handling our submission under the current circumstances. We also sincerely thank all reviewers for their careful reading and thoughtful comments on our paper.

We would like to briefly summarize how the reviewers' opinions evolved after our rebuttal and discussion:

1.  **Reviewer Zuao (Theorem 2)**
     Initially, Reviewer Zuao misunderstood Theorem 2. After our rebuttal, Reviewer Zuao replied that the concerns about Theorem 2 had been resolved and that the score would be **increased to 6**.

2.  **Reviewer svxt (main contributions, related work, real-data experiments)**
     Reviewer svxt was initially uncertain about our main contributions, and requested (i) a clearer distinction from related work, and (ii) additional comparisons and real-data experiments.
     In the rebuttal, we:

    *   Restated and clarified the main contributions and their relationship to key prior work;

    *   Expanded our discussion of related work to more clearly highlight the differences from prior approaches;

    *   Added two additional real-data experiments to demonstrate the practical relevance of our method.

    Reviewer svxt replied that all of these concerns were **fully resolved** and that the score would be **increased**. This shows that the reviewer was very satisfied with our response and holds a **positive view of our work**.

3.  **Reviewer SVNR (technical contribution and overall assessment)**
Reviewer SVNR mentioned having prior research experience in tensor optimization and, therefore, being familiar with its complexity. From this perspective, the reviewer acknowledged the **non-trivial technical contributions** of our work, kept a positive score of **6**, and indicated a **leaning towards acceptance**.

4. **Reviewer wQhQ (weaknesses and potential score increase)**
Reviewer wQhQ raised three main concerns:

    * **Symmetry and PSD constraints**: We clarified that the tensor is indeed **symmetric and positive semidefinite**, which is a common approach in prior matrix and tensor factorization works. This assumption makes the main results of the paper more intuitive and easier to understand.

    * **Distinction from matrix case**: We emphasized the **non-trivial technical challenges** between matrix and tensor, which was also acknowledged by **Reviewer SVNR**.

    * **Real-data experiments**: We added **two real-data experiments** to demonstrate the practical relevance of our method.

Due to unforeseen circumstances, Reviewer wQhQ was unable to respond. However, in the initial review, the reviewer already assigned a score of **6** and explicitly mentioned that **raising the score would be considered if the authors addressed a couple of concerns in the Weaknesses section**, which demonstrates recognition of this work.


We hope this summary, together with our rebuttal and the existing discussion, helps convey how the reviewers' concerns have been resolved and how their evaluations evolved. Once again, we sincerely thank the area chairs for their time and efforts in managing this process.

Best regards,

Authors of Paper #5985

---

### Meta-Review · Area_Chair_yFYP · 2026-01-06

**Summary:**

The reviews converged toward a positive assessment after rebuttal, primarily due to the strength of the technical analysis and the clarified positioning of the contribution relative to prior tensor and matrix results. The remaining disagreement is mostly about perceived novelty (new phenomenon vs technically demanding but incremental extension of known matrix ideas).

**Reviewer Concerns:**

Novelty and positioning vs prior work. Is the contribution is incremental given existing FGD and early-stopping ideas? How distinct is it from recent tubal-tensor GD analyses and matrix analogs? Reviewers requested more explicit discussion of tensor-specific challenges and clearer differentiation from closely related papers.

Assumptions. Work relies on symmetry/T-PSD assumptions induced by the chosen factorization and concerns about generality beyond this setting.

Empirical validation. Lack of real-data experiments in the original submission and limited breadth of empirical evaluation.

Clarity + presentation. Need for more accessible background on the t-product/t-SVD framework and clearer definitions (e.g., tubal tensor nuclear norm, block-diagonal operator).

**Reviewer Scores:**

As the reviewers themselves stated: Zuao +2, svxt +2 or +4. Others unchanged.

---

### Decision · Program_Chairs · 2026-01-26

Accept (Poster)